# Identification of an alternative triglyceride biosynthesis pathway

Gian-Luca McLelland[1✉], Marta Lopez-Osias[1], Cristy R. C. Verzijl[2], Brecht D. Ellenbroek[1], Rafaela A. Oliveira[1], Nicolaas J. Boon[1], Marleen Dekker[1], Lisa G. van den Hengel[1], Rahmen Ali[3], Hans Janssen[4], Ji-Ying Song[5], Paul Krimpenfort[3], Tim van Zutphen[2,6], Johan W. Jonker[2] & Thijn R. Brummelkamp[1✉]

Triacylglycerols (TAGs) are the main source of stored energy in the body, providing an important substrate pool for mitochondrial beta-oxidation. Imbalances in the amount of TAGs are associated with obesity, cardiac disease and various other pathologies[1,2]. In humans, TAGs are synthesized from excess, coenzyme A-conjugated fatty acids by diacylglycerol *O*-acyltransferases (DGAT1 and DGAT2)[3]. In other organisms, this activity is complemented by additional enzymes[4], but whether such alternative pathways exist in humans remains unknown. Here we disrupt the DGAT pathway in haploid human cells and use iterative genetics to reveal an unrelated TAG-synthesizing system composed of a protein we called DIESL (also known as TMEM68, an acyltransferase of previously unknown function) and its regulator TMX1. Mechanistically, TMX1 binds to and controls DIESL at the endoplasmic reticulum, and loss of TMX1 leads to the unconstrained formation of DIESL-dependent lipid droplets. DIESL is an autonomous TAG synthase, and expression of human DIESL in *Escherichia coli* endows this organism with the ability to synthesize TAG. Although both DIESL and the DGATs function as diacylglycerol acyltransferases, they contribute to the cellular TAG pool under specific conditions. Functionally, DIESL synthesizes TAG at the expense of membrane phospholipids and maintains mitochondrial function during periods of extracellular lipid starvation. In mice, DIESL deficiency impedes rapid postnatal growth and affects energy homeostasis during changes in nutrient availability. We have therefore identified an alternative TAG biosynthetic pathway driven by DIESL under potent control by TMX1.

TAGs are neutral lipids composed of a glycerol backbone conjugated to three fatty acyl chains and serve as the main unit of stored energy in a range of organisms, including oleaginous bacteria, algae and mammals. In humans, most (possibly all) cell types are able to synthesize triglycerides. High levels of TAGs (hypertriglyceridaemia) are associated with obesity and metabolic syndrome, and the mobilization of TAGs from adipose tissue can contribute to cachexia, a multiorgan wasting disorder[1,2,5–7]. In cells, TAGs can be stored in dedicated organelles known as lipid droplets, which provide energy to mitochondria through direct interorganellar contact sites[8]. TAG synthesis in humans is carried out by DGAT enzymes, which catalyse the CoA-dependent acylation of diacylglycerol (DAG) produced by the Kennedy pathway[3,5,9–11]. DGAT1 and DGAT2 reside at the endoplasmic reticulum (ER) and are considered to be therapeutic targets for metabolic disease[12–15]. DGAT-dependent TAG formation is typically limited by the availability of free fatty acids[3,5,16]. Inspired by observations of alternative mechanisms of TAG synthesis in algae, yeast and mice, we used a haploid genetic approach in human cells to identify an unexpected route for catalysing TAG synthesis.

## TMX1 restricts TAG accumulation

We first characterized TAG synthesis in haploid human HAP1 cells. Acute inhibition of DGAT1 and DGAT2 (Extended Data Fig. 1a) in HAP1 cells severely decreased baseline TAG levels, as measured by using thin-layer chromatography (TLC) to separate lipid extracts from these cells (Extended Data Fig. 1b,c). We next deleted *DGAT1* and *DGAT2* (*DGAT* double-knockout (DKO); Extended Data Fig. 1d,e) in HAP1 cells, making them resistant to lipid droplet accumulation through the loading of free fatty acids (in the form of oleic acid; Extended Data Fig. 1f) as this is a DGAT-dependent process[5].

We designed a genetic screen in haploid human cells to identify regulators of alternative TAG accumulation, reasoning that carrying out this screen in cells lacking DGAT1 and DGAT2 would allow us to identify regulators of an alternative pathway. Applying gene-trap mutagenesis coupled with fluorescence-activated cell sorting (FACS) and deep sequencing[17], we used lipid droplets as a fluorescent surrogate for TAG levels (Fig. 1a). Notably, mutations in transmembrane thioredoxin 1

[1]Oncode Institute, Division of Biochemistry, The Netherlands Cancer Institute, Amsterdam, The Netherlands. [2]Department of Pediatrics, Section of Molecular Metabolism and Nutrition, University Medical Center Groningen, University of Groningen, Groningen, The Netherlands. [3]Animal Modeling Facility, The Netherlands Cancer Institute, Amsterdam, The Netherlands. [4]Electron Microscope Facility, The Netherlands Cancer Institute, Amsterdam, The Netherlands. [5]Animal Pathology, The Netherlands Cancer Institute, Amsterdam, The Netherlands. [6]Faculty Campus Fryslân, University of Groningen, Leeuwarden, The Netherlands. ✉e-mail: g.mclelland@nki.nl; t.brummelkamp@nki.nl

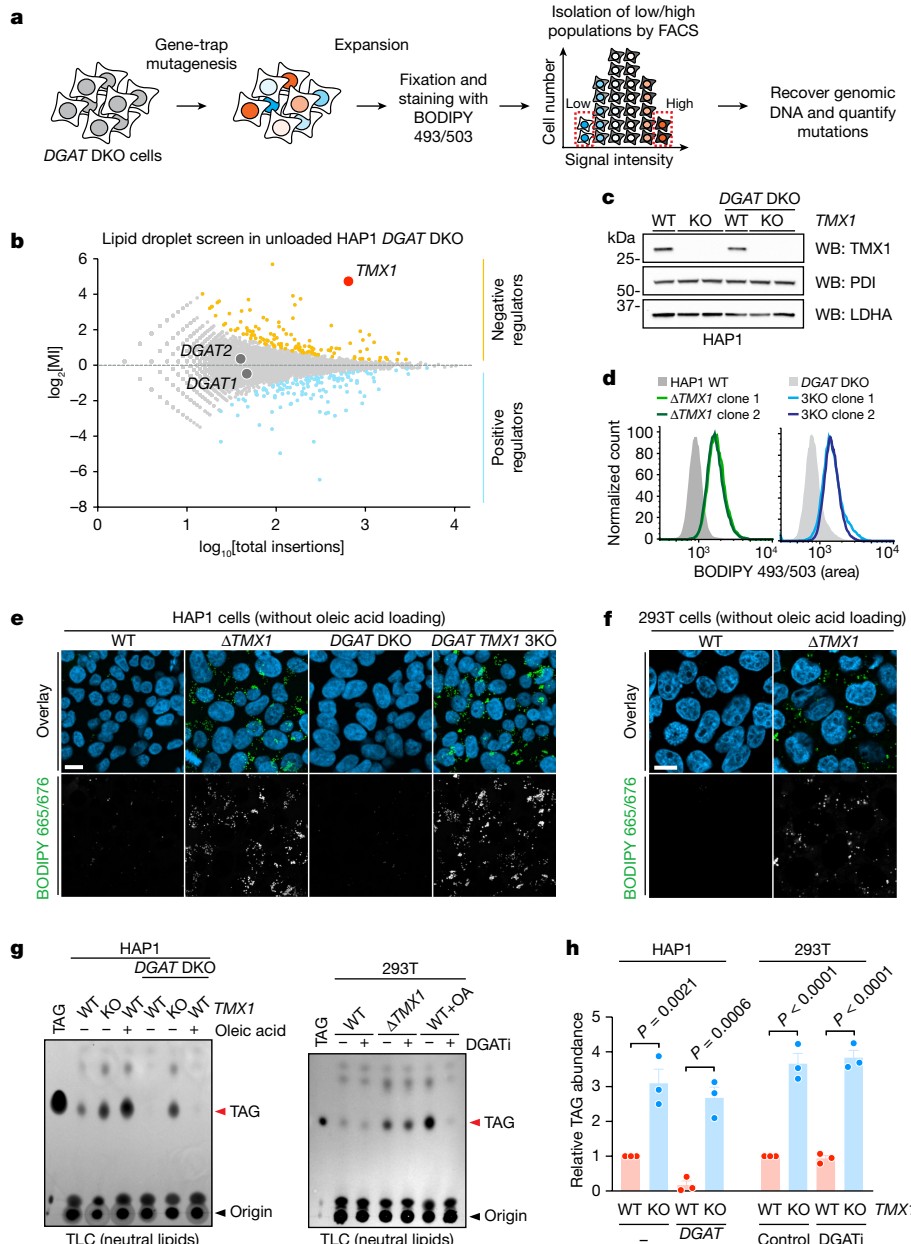

**Fig. 1 | TMX1 suppresses alternative TAG accumulation. a**, Schematic of a haploid genetic screen in *DGAT* DKO cells using the stain BODIPY 493/503. Low and high represent the 5% of cells with the lowest and highest fluorescent signal, respectively. **b**, Fishtail plot depicting genetic regulators of lipid droplets in a screen of *DGAT* DKO HAP1 cells. Significant positive and negative regulators are coloured light blue and orange, respectively. The mutational index (MI) represents the ratio of inactivating gene-trap mutations per gene recovered from each (high and low) population (see Methods for a complete description). **c**, Immunoblot of TMX1 levels in HAP1 cell lines. WB, western blot; PDI, protein disulfide isomerase; LDHA, lactate dehydrogenase A. **d**, Quantitative increase in lipid droplets (visualized by BODIPY 493/503) in *TMX1*-knockout (Δ*TMX1*)

HAP1 cells, as measured by flow cytometry. **e**, Lipid droplets, visualized by BODIPY 665/676 (green in the overlay), in HAP1 cell lines, including a double-*DGAT* and *TMX1* triple knockout (3KO). Blue, Hoechst 33342. Scale bar, 10 μm. **f**, Lipid droplets (visualized by BODIPY 665/676) in Δ*TMX1* 293T cells. Blue, Hoechst 33342. Scale bar, 10 μm. **g**, TLC of neutral lipids in HAP1 (left) and 293T (right) cell lines. Cells were pulsed with 50 μM oleic acid (OA) for 24 h where indicated. 293T cells were additionally treated with 10 μM (each) DGAT inhibitor (DGATi) where indicated. **h**, Quantification of the increase in TAG induced by *TMX1* deletion in HAP1 and 293T cell lines, normalized to WT cells. Data are mean ± s.e.m. of *n* = 3 independent experiments (two-way ANOVA, Bonferroni correction; each cell line was analysed separately).

(*TMX1*) led to the accumulation of lipid droplets independently of DGAT molecules and also in the absence of free fatty acid loading (Fig. 1b). This result was further validated by the genetic disruption of *TMX1* in either a wild-type (WT) or *DGAT* DKO background in HAP1 cells (Fig. 1c–e and Extended Data Fig. 2a), as well as in 293T cells (Fig. 1f). TLC analysis of *TMX1*-null HAP1 and 293T cells revealed robust accumulation of TAG in the absence of DGAT activity, with an amount similar to that of oleic acid-induced TAG in DGAT-competent cells (Fig. 1g,h).

We observed a similar induction of DGAT-independent TAG accumulation after *TMX1* disruption in A549 and U2OS cells (Extended Data Fig. 2b). *TMX1* encodes a transmembrane, ER-resident oxidoreductase (Extended Data Fig. 2c,d) that has a cysteine-containing thioredoxin domain located in the lumen of the ER[18]. Mutation of these redox cysteines correlated with a substantial reduction in TMX1 protein abundance (Extended Data Fig. 2e). All TMX family members are expressed in HAP1 cells[17], but *TMX1* transcripts are the most abundant (Extended

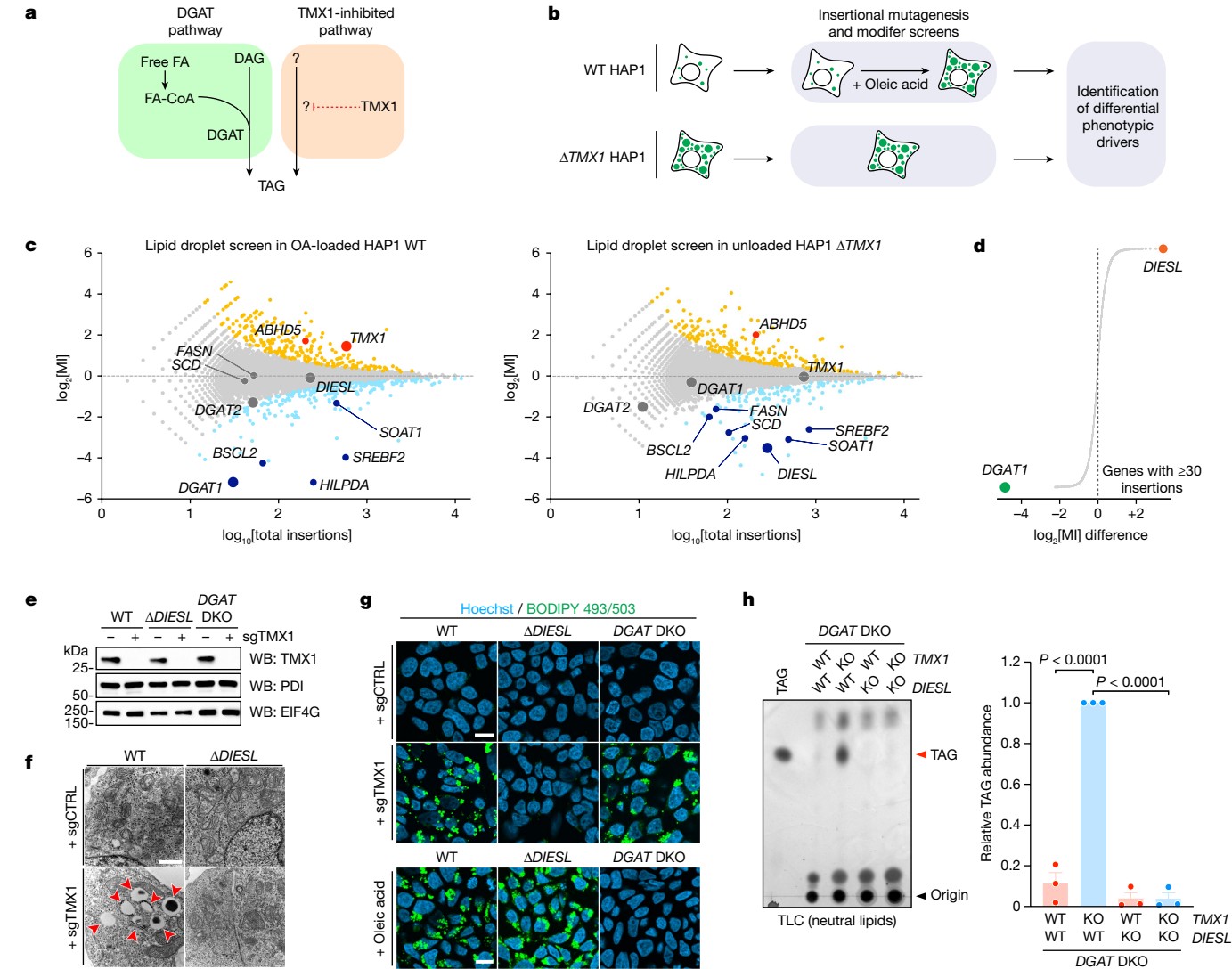

**Fig. 2 | DIESL drives TAG accumulation in the absence of TMX1. a**, Schematic representation of the DGAT pathway (green box) and the putative TMX1-inhibited pathway (orange box). FA, fatty acid or fatty acyl. **b**, Set-up of the modifier screens used to identify the regulators of each pathway. **c**, Fishtail plots of lipid droplet screens in WT HAP1 cells treated with oleic acid (OA) (left) and in ΔTMX1 HAP1 cells (right). Significant positive and negative regulators are coloured light blue and orange, respectively; larger dots indicate the genes of interest. **d**, Difference in mutational index (log₂-transformed) between the two screens for every gene with at least 30 insertions in each screen.

**e**, Immunoblot of TMX1 in WT, ΔDIESL and DGAT DKO HAP1 cells transduced with a synthetic guide RNA targeting TMX1 (sgTMX1). **f**, Ultrastructural analysis of lipid droplets (red arrowheads) in WT and ΔDIESL cells after loss of TMX1. Scale bar, 1 μm. **g**, Lipid droplets, visualized by BODIPY 665/676, in WT, ΔDIESL and DGAT DKO cells transduced with the indicated sgRNA or treated with 200 μM oleic acid for 24 h. sgCTRL is a control sgRNA. Scale bars, 10 μm. **h**, TLC analysis of neutral lipids (left) and quantification of TAG (right) in DGAT DKO HAP1 cells also lacking DIESL and/or TMX1. Bars represent mean ± s.e.m. of n = 3 independent experiments (two-way ANOVA, Bonferroni correction).

Data Fig. 2f). We therefore introduced other TMX family members into cells that lacked *TMX1* (Extended Data Fig. 2g,h). Unlike TMX1, TMX3 and the evolutionarily similar TMX4 failed to suppress TAG and lipid droplet accumulation in these cells (Extended Data Fig. 2i–k); TMX2 was expressed at lower levels. The loss of *TMX1* therefore activates TAG accumulation in human cells by a mechanism that is independent of the known TAG biosynthetic process.

## Alternative TAG accumulation by DIESL

To find out how *TMX1*-deficient cells accumulate TAG (Fig. 2a), we carried out a genetic suppressor screen in this condition to identify the underlying mechanism, again using lipid droplet accumulation as a readout. We then compared the results of this screen with a screen in WT cells loaded with oleic acid (Fig. 2b). Taken together, these two genetic maps (Fig. 2c) revealed lipogenesis to be a major driver

of lipid droplet accumulation common to both backgrounds, with sphingolipid metabolism and mitochondrial electron transport as minor positive and negative regulators, respectively (Extended Data Fig. 3). Moreover, both screens identified *SOAT1*, which encodes sterol O-acyltransferase, as a positive regulator of lipid droplet biogenesis (Extended Data Fig. 3). Cholesteryl esters co-accumulate with TAG in lipid droplets[3], although SOAT1 activity by itself did not explain the accumulation of TAGs. Comparison of the mutational biases of regulators in both screens identified *DGAT1* as an important driver of lipid droplets in cells loaded with oleic acid[5] (Fig. 2c). Notably, our study revealed a functionally uncharacterized gene, *TMEM68*, as a selective driver of lipid droplet accumulation in the absence of *TMX1* (Fig. 2c). For reasons discussed below, we renamed this gene DGAT1/2-independent enzyme synthesizing storage lipids (*DIESL*). *DGAT1* and *DIESL* were the strongest selective drivers of lipid droplet accumulation in response to oleic acid loading and *TMX1* loss, respectively (Fig. 2d). *DIESL*-knockout

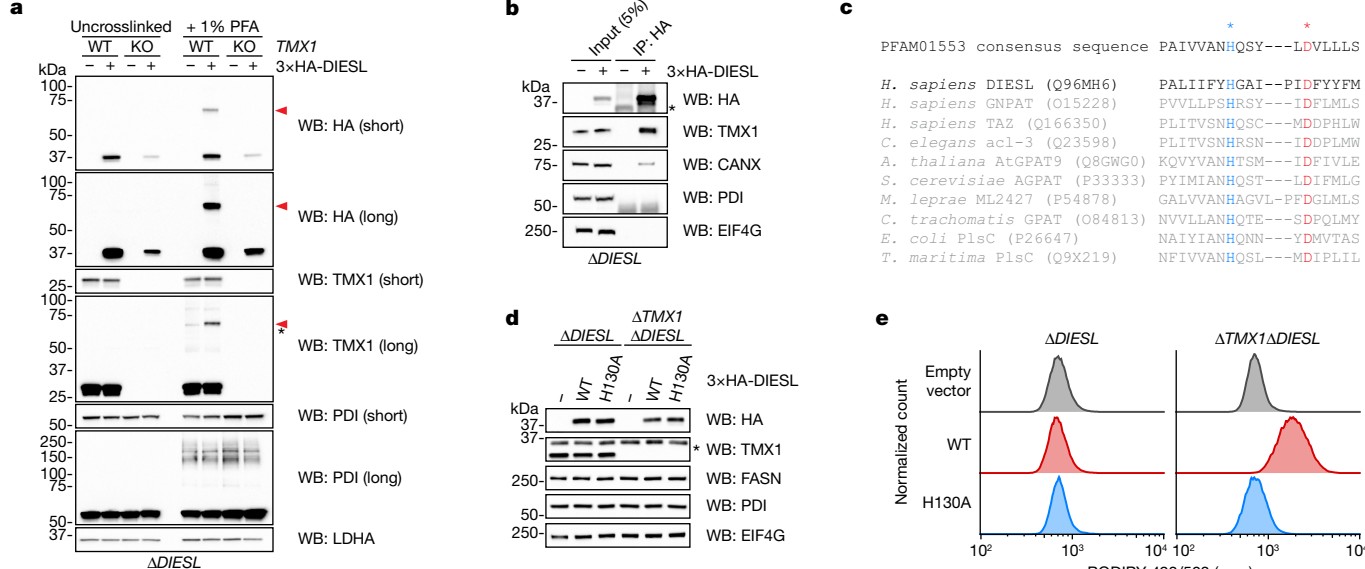

**Fig. 3 | The TMX1–DIESL enzymatic complex drives alternative triglyceride synthesis. a**, Immunoblot analysis of TMX1 and haemagglutinin (HA)-tagged DIESL (3×HA–DIESL) in HAP1 cells lacking endogenous DIESL (and TMX1), with or without crosslinking by 1% paraformaldehyde (PFA). Red arrowheads indicate the DIESL–TMX1 heterodimer, and the asterisk indicates a non-DIESL band. **b**, Co-immunoprecipitation of TMX1 with DIESL from rescued HAP1 cells (the asterisk indicates antibody chains). CANX, calnexin; PDI, protein disulfide isomerase; EIF4G, eukaryotic translation initiation factor 4G. **c**, Sequence conservation of catalytic dyads composed of a histidine (H, blue asterisk) and an aspartate (D, red asterisk) across acyltransferases (Uniprot accession numbers in parentheses). Species are *Homo sapiens, Caenorhabditis elegans, Arabidopsis thaliana, Saccharomyces cerevisiae, Mycobacterium leprae, Chlamydia trachomatis, Escherichia coli* and *Thermotoga maritima*. **d**, Western blot of HAP1 cells rescued with WT or catalytic-dead (H130A) DIESL (the asterisk indicates a non-specific band). FASN, fatty acid synthase. **e**, Analysis of lipid droplets (BODIPY 493/503 fluorescence intensity) in DIESL-rescued HAP1 cells by flow cytometry.

cells were resistant to lipid droplet accumulation when *TMX1* was disrupted (Fig. 2e–g) but were phenotypically normal when loaded with oleic acid (Fig. 2g), which confirmed the independence of the TMX1–DIESL genetic pathway from the canonical DGAT pathway[9]. In agreement with these observations, the loss of *TMX1* induced TAG accumulation that was carried out by DIESL, rather than by DGAT1 or DGAT2 (Fig. 2h). These genetic studies demonstrate that DIESL and the two DGAT enzymes promote TAG accumulation independently in different cellular contexts.

## DIESL is a TMX1-bound acyltransferase

*DIESL* encodes a protein that is localized to the ER membrane[19] (Extended Data Fig. 4a). We observed that genetic disruption of *TMX1* was associated with altered abundance of DIESL protein (Extended Data Fig. 4b). When protein synthesis was inhibited, TMX1 deficiency led to the increased turnover of DIESL (Extended Data Fig. 4c), which suggested the existence of a stable TMX1–DIESL complex. Analysing the binding partners of DIESL by chemical crosslinking, we found a single, crosslinked band at around 70 kDa by SDS polyacrylamide gel electrophoresis (SDS–PAGE) and subsequent immunoblotting (Fig. 3a). We determined that this 70-kDa band was a complex composed of both TMX1 and DIESL because it was detected by an antibody raised against TMX1 and was absent from *TMX1*-knockout cells (HAP1 cells in Fig. 3a and HeLa cells in Extended Data Fig. 4d). Moreover, we were able to co-immunoprecipitate TMX1 with DIESL (Fig. 3b and Extended Data Fig. 4e) in a detergent-dependent manner (Extended Data Fig. 4f). These data show that both TMX1 and DIESL are proximal (accessible by formaldehyde crosslinking) and can interact in a membrane-dependent manner.

DIESL has previously been characterized broadly as an ER-resident acyltransferase of otherwise-unknown function[19]. Homology modelling of DIESL revealed that its C terminus adopts an acyltransferase fold

similar to that of a bacterial acylglycerol-phosphate acyltransferase, which belongs to an enzyme family that can carry out diverse forms of acylation reactions[20]. The DIESL N terminus contains a conserved *N*-glycosylation site on Asn5 that could be validated experimentally (Extended Data Fig. 4g–i), implying that this extremity is exposed to the ER lumen. These glycans could be elongated by treating cells with brefeldin A (Extended Data Fig. 4j,k), which allows Golgi glycosylases to access ER-resident proteins by merging these two organelles[21], supporting the localization of DIESL to the ER. Loss of glycosylation did not affect the ability of DIESL to drive TAG accumulation in the absence of TMX1 (Extended Data Fig. 4l). AlphaFold[22] predicts an extended helical N terminus that would pass through the membrane (Extended Data Fig. 4m). The expression of truncated DIESL lacking this N-terminal luminal domain was not detected by immunoblot analysis (Extended Data Fig. 4n,o). By sequence alignment, we could identify the active site as a catalytic dyad composed of His130 and Asp136 (Fig. 3c) contained within a catalytic pocket facing the membrane (Extended Data Fig. 4p). Overexpressing DIESL induced the accumulation of lipid droplets only in the absence of TMX1, and this was abolished by mutation of the DIESL active site (H130A; Fig. 3d,e). These findings indicate that DIESL resides in the ER membrane with TMX1 and catalyses DGAT-independent TAG accumulation in an enzymatic manner.

## DIESL is a DGAT

We then studied the effects of the activated DIESL acyltransferase on the cellular lipidome. We began by generating 'TAG-null' HAP1 cells that lacked DIESL and the DGATs (Δ*DGAT1*Δ*DGAT2*Δ*DIESL*; Extended Data Fig. 5a) and reintroduced either catalytic active or inactive DIESL in the presence or absence of TMX1, observing robust TAG accumulation under the control of both the DIESL active site and its regulator, TMX1 (Extended Data Fig. 5b). We then analysed the lipidome in

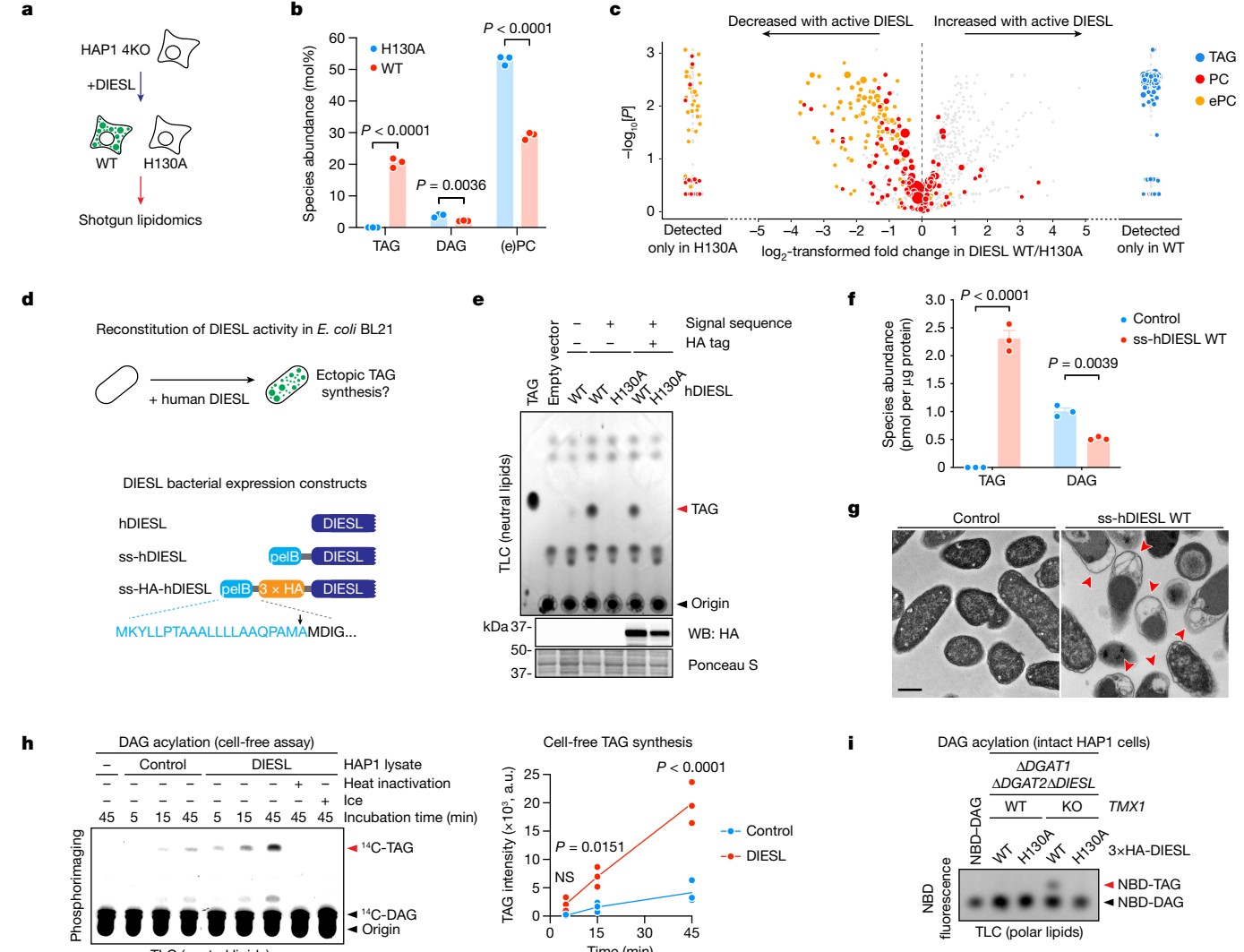

**Fig. 4 | TAG synthesis by DIESL. a**, Lipidomic analysis of 4KO ($\Delta DGAT1\Delta DGAT2$ $\Delta DIESL\Delta TMX1$) HAP1 cells. **b**, Relative abundance of TAG, DAG and (e)PC in 4KO HAP1 cells reconstituted with 3×HA–DIESL. Bars represent mean ± s.e.m. of $n = 3$ independent samples (two-way ANOVA, Bonferroni correction). **c**, Change in abundance of detected lipids (1,183 species) between 4KO HAP1 cells expressing WT or H130A DIESL. PC (red), ePC (orange) and TAG (blue) are indicated. Circle size depicts the relative, scaled abundance of the indicated molecule. **d**, Reconstitution of TAG synthesis in *E. coli* by DIESL (top) and schematic representation of human (h) DIESL constructs for expression in *E. coli* (bottom). Blue, pectate lysate B (pelB) signal sequence (ss); black arrowhead, cleavage site. **e**, TLC separation of neutral lipids and immunoblot analysis from *E. coli* expressing the indicated construct. **f**, Absolute abundance of TAG and DAG in *E. coli* expressing either ss-hDIESL or empty vector. Bars represent

mean ± s.e.m. of $n = 3$ independent samples (two-way ANOVA, Bonferroni correction). **g**, Ultrastructural analysis of *E. coli* expressing ss-hDIESL H130A (control) or WT. Red arrowheads indicate lipid-rich inclusions. Scale bar, 500 nm. **h**, Left, cell-free reconstitution of TAG synthesis by DIESL. Lysates from control 4KO HAP1 cells or those expressing 3×HA-DIESL were incubated with 50 μM $^{14}$C-DAG for the indicated time period at 37 °C unless otherwise indicated. Lipid extracts were separated by TLC and analysed by phosphorimaging. Right, intensities were quantified for $n = 3$ independent experiments. The line connects the mean of each time point for both conditions (two-way ANOVA, Bonferroni correction). NS, not significant. **i**, Acylation assay of NBD–DAG. HAP1 cell lines were treated with 25 μM NBD–DAG for 1 h before TLC analysis of polar lipids (to accommodate the charge on the NBD group). NBD-tagged lipids were identified by NBD fluorescence.

DIESL-reconstituted quadruple knockouts ($\Delta DGAT1\Delta DGAT2$ $\Delta DIESL\Delta TMX1$ (4KO)) and compared the effects of reintroducing catalytic active and inactive DIESL (Fig. 4a). The most pronounced change was observed for TAGs, which were not detected in control DIESL(H130A) cells but made up 20.5% of the lipidome in cells expressing active DIESL (Extended Data Fig. 5c). We also observed an increase in cholesteryl esters, which was not surprising because we had identified *SOAT1* in our genetic screens (Extended Data Fig. 3). DIESL activity reduced the fractional abundance of only two lipid species (Fig. 4b,c and Extended Data Fig. 5d–g): DAG and the major membrane phospholipid phosphatidylcholine (PC), as well as its ether-linked form (ePC). This unbiased lipidomic analysis demonstrates that DIESL primarily stimulates TAG

abundance, possibly by the acylation of DAG using a phospholipid (or a phospholipid precursor) as an acyl donor.

To demonstrate that DIESL is an autonomous TAG synthase, we reconstituted DIESL-dependent TAG production in *E. coli*, a non-oleaginous organism that lacks the genes required for TAG synthesis[23]. Expression of human DIESL in *E. coli* (preceded by a bacterial membrane-targeting sequence; Fig. 4d) conferred TAG synthesis to this organism, as determined by TLC (Fig. 4e) and mass spectrometry (MS) (Fig. 4f), which came at the expense of DAG levels. We additionally observed the formation of lipid inclusion bodies in these bacteria (Fig. 4g). We then examined DIESL-dependent DAG-to-TAG conversion using a cell-free assay. Incubation of HAP1 4KO ($\Delta DGAT1\Delta DGAT2\Delta DIESL\Delta TMX1$) lysates

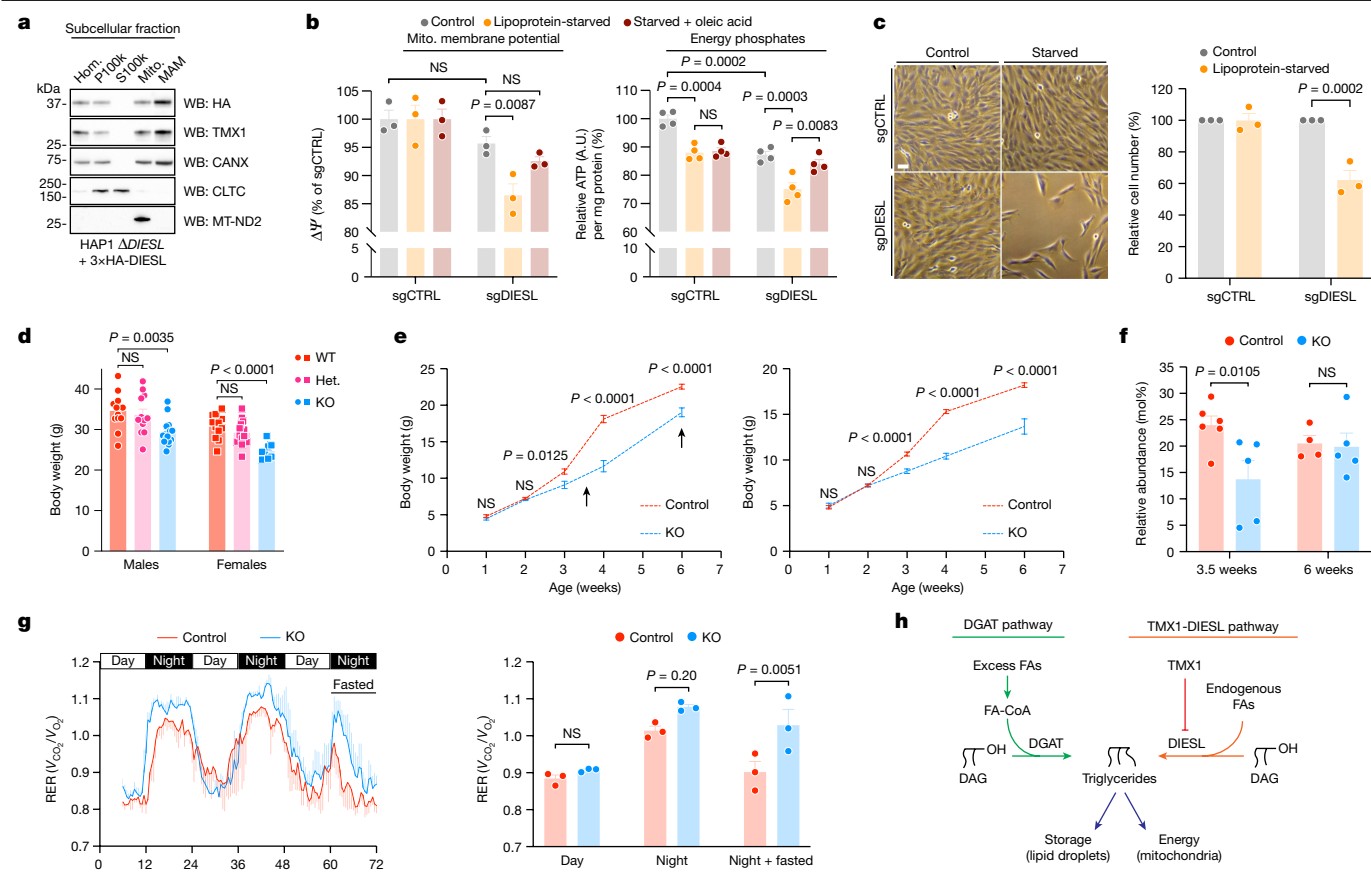

**Fig. 5 | DIESL deficiency in cells and mice. a**, Subcellular fractionation of HAP1 cells. Hom., homogenate; P100k, pellet obtained by ultracentrifugation at 100,000$g$; S100k, 100,000$g$ supernatant; Mito., mitochondria; MAM, mitochondria-associated membrane of the ER. Markers of light membranes and cytosol (CLTC), as well as mitochondria (MT-ND2), are included. **b**, Mitochondrial membrane potential ($\Delta\psi$, left) and ATP levels (right) in RPE1 cells, cultured in either complete medium (control), medium lacking lipoproteins or lipoprotein-deficient medium supplemented with 50 μM oleic acid for 24 h. Bars represent mean ± s.e.m. of $n = 3$ ($\Delta\psi$) or $n = 4$ (ATP) independent experiments, respectively (two-way ANOVA, Bonferroni correction). **c**, Bright-field images (left) and quantification of viability (right) of the RPE1 cells from **b**. Bars represent mean ± s.e.m. of $n = 3$ independent experiments (two-way ANOVA, Bonferroni correction). Scale bar, 50 μm. **d**, Body weight of adult (22–28-week-old) mice. Bars represent mean ± s.e.m. of $n = 8$–15 mice (one-way ANOVA, Bonferroni correction). Het., heterozygous mice. **e**, Postnatal growth curves of male (left) and female (right) mice (Control designates both WT and heterozygous mice). Bars represent mean ± s.e.m. of $n = 5$–17 mice (two-way ANOVA, Bonferroni correction); male and female mice were analysed separately. **f**, Hepatic TAG levels in male mice, quantified at 3.5 and 6 weeks (arrows in **e**), expressed as a percentage of total lipids quantified (Extended Data Fig. 9l). Bars represent mean ± s.e.m. of $n = 4$–6 livers per condition (two-way ANOVA performed on the entire dataset, Bonferroni correction). **g**, Respiration exchange ratio (RER). Left, an increased RER indicates preferential carbohydrate oxidation (instead of lipids). Right, quantification of RER in male mice. Bars represent mean ± s.e.m. of $n = 3$ mice (two-way ANOVA, Bonferroni correction). **h**, The DGAT pathway (green) acylates DAG using exogenously derived fatty acyl-CoA (FA-CoA) whereas the TMX1–DIESL pathway (orange) uses endogenous fatty acyl chains derived from membrane phospholipids or their precursors.

with isotope-labelled DAG demonstrated its time-dependent conversion to TAG, which was stimulated by the presence of DIESL and was sensitive to heat inactivation and low temperature (Fig. 4h). We also measured this conversion in intact cells using fluorescent DAG (nitrobenzoxadiazole-labelled DAG (NBD–DAG); Extended Data Fig. 6a). Whereas triple-knockout, TAG synthase-null cells showed no detectable conversion of DAG to TAG, DIESL was able to stimulate DAG acylation when TMX1 was inactivated in a manner strictly dependent on its catalytic acyltransferase site (Fig. 4i and Extended Data Fig. 6b–e). These data show that DIESL functions as a DAG acyltransferase and that its exogenous expression confers the ability to synthesize TAG to an organism inherently devoid of this capacity.

## DIESL affects energy homeostasis

We next addressed the role and importance of DIESL-synthesized TAGs. During conditions of nutrient abundance (in the form of extracellular free fatty acids), the DGAT enzymes are known to synthesize TAGs,

and we observed that this process was unaffected by the loss of DIESL (Fig. 2g). Accordingly, measuring the amount of TAG in several cell lines revealed that DIESL could noticeably affect the cellular TAG pool in the absence of DGAT activity (Extended Data Fig. 7). We therefore focused on the function of DIESL when the DGAT enzymes would be inactive. Because both TMX1 (as previously shown[24]) and DIESL reside at the mitochondria-associated membrane of the ER (Fig. 5a), we reasoned that TMX1–DIESL might support mitochondrial function during periods of lipid starvation, when decreased DGAT-dependent TAG formation would necessitate a reliance on an alternative means of TAG synthesis. To examine mitochondrial activity, we analysed the activation of AMP kinase (AMPK), a master metabolic sensor that directly detects the energy state of the cell[25,26]. Four *DIESL*-deficient cell lines (RPE1, U251, HT29 and 293T) were sensitized to AMPK phosphorylation, which occurred when extracellular lipids were depleted or the DGAT enzymes were inhibited (Extended Data Fig. 8a,b). Focusing on untransformed RPE1 cells, we measured increased mitochondrial reactive oxygen species in the absence of DIESL, a feature that was exacerbated during

lipoprotein starvation (Extended Data Fig. 8c). This starvation response did not cause widespread autophagy (Extended Data Fig. 8d,e). DIESL deficiency in RPE1 cells also decreased the mitochondrial membrane potential and ATP levels when cells were cultured in the absence of lipoproteins, but this could be rescued with oleic acid (Fig. 5b). This rescue was sensitive to the inhibition of carnitine palmitoyltransferase 1A (CPT1A) by etomoxir, demonstrating that it was dependent on fatty acid uptake by mitochondria (Extended Data Fig. 8f). RPE1 cells that lacked DIESL showed impaired cellular fitness when cultured under lipoprotein-depleted conditions (Fig. 5c). Culturing more resilient cell lines, such as 293T and U251 cells, under more severe starvation conditions also sensitized them to *DIESL* loss (Extended Data Fig. 8g). Thus, during various starvation stimuli across several cell types, DIESL was shown to contribute to mitochondrial function, energy levels and cell fitness.

To study the role of DIESL in vivo, we generated knockout mice (Extended Data Fig. 9a–c). *Diesl*-knockout mice were viable, born at the expected Mendelian ratio (Methods) and displayed a reduction in male (17%) and female (19%) body weight at adulthood (more than 5 months old) (Fig. 5d). The reduction in body weight was accompanied by a decrease in body length, a roughly 50% reduction in fat mass (Extended Data Fig. 9d,e) and a reduction in TAG levels in serum and brain (Extended Data Fig. 9f,g). The organs of *Diesl*-knockout mice were of normal size (when corrected for body weight) and morphology (Extended Data Fig. 9h,i). It thus follows that the TAG synthase DIESL is required to attain normal body weight and size in mice.

We studied the consequences of *Diesl* deficiency in mice in more detail. Notably, knockout mice were indistinguishable from their WT counterparts one week after birth but grew at a much slower rate between weeks two and four (Fig. 5e). The weight difference we found after four weeks was maintained throughout adulthood (Fig. 5d). Knockout mice displayed a small decrease in food intake that was absent when adjusted for body weight (Extended Data Fig. 9j). Thus, *Diesl* is required to support rapid growth at a specific time point after birth, which notably coincides with a dietary switch from milk to chow. By MS, we detected a roughly 40% reduction in liver TAG at 3.5 weeks of age, but this was no longer present several weeks later when WT and knockout mice grew at similar rates (Fig. 5f and Extended Data Fig. 9k,l). Because we measured low TAG levels, we reasoned that DIESL could provide the energy required for rapid growth at this stage of life because fuel sources were limiting. We used indirect calorimetry in fasted adult mice to measure energy homeostasis under conditions of nutrient shortage. We observed an increased respiratory exchange ratio in response to fasting in *Diesl*-deficient mice (significant by ANOVA, although the number of animals was too low for reliable regression analysis; Fig. 5g), indicating an attenuated switch from carbohydrate to lipid oxidation compared with WT mice. Taken together, our data show that TMX1–DIESL is essential to support the rapid organismal growth of young animals and regulates a response to dynamic changes in nutrient availability.

## Discussion

The enzymatic synthesis of triglyceride from fatty acyl-CoA precursors was described more than 60 years ago[9]. Using haploid genetics, we here identify a regulated protein module, composed of TMX1 and DIESL within the ER, that carries out cell-autonomous alternative TAG synthesis. In contrast to known TAG biosynthetic routes that are activated by high extracellular lipid levels, DIESL converts endogenous acyl chains into TAGs, altering the composition of cellular membranes. In cells, TMX1–DIESL may enable the supply of fatty acids to mitochondria during times of lipid or nutrient scarcity. TAG synthesis by TMX1–DIESL at the expense of membrane phospholipids during nutrient limitation is reminiscent of an autophagy-like mechanism and thus would require tight regulation by TMX1 (Fig. 5h). How the activity of the TMX1–DIESL

complex is precisely tuned to the needs of the mitochondrion (perhaps by oxidative signals, as TMX1 contains a thioredoxin domain) remains to be elucidated.

Loss of *Diesl* leads to metabolic phenotypes in mice, indicating the importance of alternative TAG synthesis in normal physiology. In mice two to four weeks old, we observed a significant delay in rapid postnatal growth that coincides with a dietary switch from milk (20% fat[27]) to chow (4% fat). Furthermore, fasting of adult mice lacking *Diesl* suggests an impaired switch from carbohydrates to lipids for nutrient oxidation. These observations point to a role for DIESL during fluctuations or limitations in nutrient supply.

Studies of DGAT-dependent TAG synthesis during conditions of high lipid accumulation have led to them being examined as drug targets in disease (DGAT1 in obesity[6] and DGAT2 in fatty liver disease[14]). Conversely, TAG synthesis during nutrient restriction has received less attention but may also have a prominent biological role. Here, we ascribe such a function to TMX1–DIESL. It is conceivable that organisms would have evolved such a pathway to mitigate the consequences of food scarcity or extreme physical endurance, and these activities may have important roles in pathologies that are seriously affected by cellular energy availability, such as cancer and neurodegeneration.

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

# Methods

## Cell culture, reagents and antibodies

HAP1 cells were maintained in IMDM medium supplemented with 10% FBS, L-glutamine, penicillin and streptomycin, and cultured at 37 °C in 5% $CO_2$. 293T, A549, HeLa, HT29, RPE1, U251 and U2OS cells (purchased from ATCC) were maintained in DMEM under the same conditions. Cell lines used in this study were routinely monitored for mycoplasma contamination. BODIPY 493/503 and 665/676 neutral lipid dyes, TMRM and MitoTracker Red CM-$H_2$XROS were from ThermoFisher. Brefeldin A, CCCP, cycloheximide, etomoxir, lipoprotein-deficient fetal calf serum, oleic acid, PF-06424439 (a DGAT2 inhibitor) and phosphatase-inhibitor cocktail were from Sigma-Aldrich. A-922500 (a DGAT1 inhibitor) was from Selleck. Protease inhibitor cocktail was from Roche. TAG (triolein) and DAG (diolein) were from Avanti Polar Lipids. $^{14}$C-DAG (1,2,-dioleoyl-rac-glycerol) was from American Radio-labeled Chemicals. Antibodies used in this study were anti-ACTB (Abcam, ab6276, WB 1:10,000), anti-α-tubulin (Santa Cruz Biotechnology, sc-32293, WB 1:1,000), anti-AMPK (Cell Signaling Technology, 2532, WB 1:1,000), anti-AMPK pT172 (Cell Signaling Technology, 2535, WB 1:1,000), anti-CANX (Abcam, ab22595, immunofluorescence (IF) 1:100, WB 1:1,000), anti-CLTC (Thermo Fisher, PA5-17347, WB 1:1,000), anti-EIF4G (Cell Signaling Technology, 2498, WB 1:1,000), anti-FASN (Santa Cruz Biotechnology, sc-55580, WB 1:1,000), anti-HA (Biolegend, 901503, IF 1:200, WB 1:1,000), anti-HSPA5 (Cell Signaling Technology, 3177, WB 1:1,000), anti-LAMP1 (Santa Cruz Biotechnology, sc-19992, WB 1:1,000), anti-LC3B (Cell Signaling Technology, 2775, WB 1:1,000), anti-LDHA (Cell Signaling Technology, 3582, WB 1:5,000), anti-PDI (Abcam, ab2792, IF 1:500, WB 1:20,000), anti-S6 (Cell Signaling Technology, 2137, WB 1:1,000), anti-S6 pS235/pS236 (Cell Signaling Technology, 4856, WB 1:1,000), anti-TMX1 (Atlas Antibodies, HPA003085, IF 1:100, WB 1:1,000 and Origene, TA507042, WB 1:1,000), anti-TOMM20 (Abcam, ab186735, WB 1:10,000) and anti-V5 (ThermoFisher, 14-6796-82, IF 1:500, WB 1:1,000).

## Plasmids and cloning

We purchased sgRNAs as short single-stranded DNA with sticky ends. These oligonucleotides were then annealed and cloned into pX330 and pLentiCRISPRv2 (with puromycin, blasticidin or mCherry selection markers) that had been cut with BbsI or BsmBI (New England BioLabs), respectively. As a non-targeting control sgRNA (sgCTRL), a sequence targeting the zebrafish *tia* gene was used[28]. A complete list of sgRNAs used in this study, generated using the Broad Institute's Genetic Perturbation Platform (portals.broadinstitute.org/gpp/public), is provided in Supplementary Table 1. 3×HA-tagged DIESL (Q96MH6-1, WT and N0) and V5-tagged TMX1 (Q9H3N1), TMX2 (Q9Y320-1), TMX3 (Q96JJ7-1) and TMX4 (Q9H1E5) were purchased as linear DNA flanked by NheI and AgeI restriction sites from Integrated DNA Technologies. These were then digested and cloned into pLEX305 backbones. DIESL(N5Q) and DIESL(H130A) mutants were generated by site-directed mutagenesis using the QuikChange II kit (Agilent). For recombinant expression of DIESL, human DIESL was codon-optimized for *E. coli* expression and pET24 constructs encoding DIESL, pelB−DIESL and pelB−3×HA−DIESL were purchased as plasmids from Twist Bioscience.

## Generation of clonal knockout cell lines

HAP1 and 293T cells were transfected with pX330 encoding the sgRNA of choice, along with a plasmid either carrying an integration cassette[28] or encoding a blasticidin or puromycin resistance gene, using Xfect (Takara), according to the manufacturer's instructions. Transfected cells were selected with 25 μg ml$^{-1}$ blasticidin (HAP1) or 1 μg ml$^{-1}$ puromycin (HAP1 and 293T). After selection, the medium was replaced with complete medium and clones were allowed to form colonies, which were eventually picked by micropipette and transferred to 24-well plates. Genetic modification of individual clones was detected by PCR, in some cases using a primer annealing to the sequence of the blasticidin (5′-CCGACATGGTGCTTGTTGTCCTC-3′) or puromycin (5′-GCAACCTCCCCTTCTACGAGC-3′) resistance gene. See Supplementary Table 2 for a list of primers used to amplify genomic loci. Disruption of the locus was confirmed by Sanger sequencing, either directly or, in the case of compound heterozygotes, using TIDE analysis[29] for 293T cell lines. The targeted genetic modifications in cell lines used in this study are summarized in Supplementary Table 3.

## Lentiviral transduction

Lentivirus was produced in 293T cells transfected with pΔVPR, pVSVg and either pLEX305 or pLentiCRISPRv2 lentiviral plasmids, as well as pAdVAntage. Then, 2 days after transfection, virus was collected from conditioned medium passed through a 40 μm filter. Viral supernatants, supplemented with 8 μg ml$^{-1}$ protamine sulfate, were applied directly to cells for 24 h. In some cases, a second collection and transduction was done to improve efficiency. Transduced cells were selected with puromycin (in HAP1, 1 μg ml$^{-1}$; A549, 0.5 μg ml$^{-1}$; HeLa, 2 μg ml$^{-1}$; HT29, 3 μg ml$^{-1}$; U251, 2 μg ml$^{-1}$; U2OS, 2 μg ml$^{-1}$) or transduction efficiency was assessed by expression of a fluorescent marker (mCherry). RPE1 cells were transduced with a large viral titre and were not selected. Transduced cells were analysed between 7 and 28 days after transduction.

## Mutagenesis screening

All genetic screens reported here used gene-trap mutagenesis in haploid HAP1 cells, as described previously[17]. Typically, 2–3 × 10$^9$ gene-trapped HAP1 cells of the indicated genotype were collected by trypsinization and fixed in Fix Buffer I (BD Biosciences) for 10 min at 37 °C. For the oleic acid-loaded screen, cells were first cultured for 24 h in complete medium supplemented with 200 μM oleic acid and then chased in medium lacking oleic acid for another 24 h before collection. Cells were treated with 1 mg ml$^{-1}$ RNase A (Qiagen) diluted in FACS buffer (10% FBS in PBS) at 37 °C for 30 min before staining with 1 μg ml$^{-1}$ BODIPY 493/503 and 10 μg ml$^{-1}$ propidium iodide (Sigma-Aldrich), diluted in FACS buffer, for 1 h at room temperature. Cells were washed twice in FACS buffer before being passed through a 40 μm cell strainer. Cell sorting was done using an S3 Sorter (Bio-Rad), collecting 10$^7$ cells (per population) representing the lowest and highest 5% of BODIPY signal from haploid cells in G1 phase. The isolation of genomic DNA, preparation of sequencing libraries and the analysis were done as described[17]. Reads were aligned to the reference genome (GRCh37 genome assembly), tolerating one mismatch, and sense insertions in the 'low' and 'high' populations were compared using the Fisher's exact test to determine significant differences ($P < 0.05$). The mutational index represents the ratio of the occurrence of unique, disruptive (that is, insertions (ins.) of the gene-trap in the sense orientation) mutations in the body of a given gene (5′ untranslated region, exon and intron) in the high compared with the low population, normalized by the total of other unique, disruptive mutations in each population[17]: (sense ins. of gene $X$ in high/(total sense ins. in high − sense ins. of gene $X$ in high))/(sense ins. of gene $X$ in low/(total sense ins. in low − sense ins. of gene $X$ in low)). The number of recovered mutations for each population in each screen is reported in Supplementary Table 4.

## SDS−PAGE and immunoblot analysis

Unless otherwise specified, cells were lysed in RIPA buffer (25 mM Tris-HCl pH 7.5, 150 mM NaCl, 1% NP-40, 1% sodium deoxycholate, 0.1% SDS and protease-inhibitor cocktail) on ice. Lysates were sonicated twice (40% amplitude for 4 s), cleared by centrifugation and protein concentrations were determined by BCA assay (Thermo Scientific). Equivalent amounts of protein were separated by SDS−PAGE over Bolt 4−12% Bis−Tris gels (Invitrogen) then transferred to nitrocellulose membranes. Membranes were blocked with 5% milk−TBST and

incubated overnight at 4 °C in primary antibody diluted in 3% BSA–TBST. The next day, membranes were washed three times in TBST and then incubated in HRP-conjugated secondary antibodies (Invitrogen 65-6120 and Bio-Rad 1706516), diluted 1:10,000 in blocking buffer for 1 h at room temperature. After three more washes in TBST, signal was developed using Clarity Western ECL substrate (Bio-Rad) and imaged on a Universal Hood II GelDoc system (Bio-Rad).

## Recombinant expression in *E. coli*

BL21 *E. coli* were transformed with pET24 or pET28a plasmids expressing the indicated construct. Single clones were cultured up to a 5 ml volume in LB broth at an optical density of 0.5 at 37 °C. Cultures were then transferred to 25 °C for 1 h and expression was induced with 0.2 mM IPTG for 16 h at 25 °C. For SDS–PAGE and immunoblot analysis, 100 μl of culture was pelleted for 5 min at 3,500*g* before lysis in SDS–PAGE sample buffer.

## TLC and DAG acylation assays

Equal numbers of human cells grown to confluence in six-well plates were washed with PBS. Cells were then incubated twice in 400 μl extraction buffer (hexane:isopropanol at 3:2) for 10 min[30]. *E. coli* cell pellets from 2 ml of the overnight culture were incubated in 500 μl extraction buffer (only once) for 30 min while being rotated. Extractions were pooled and dried under a stream of nitrogen to a volume of around 10 μl and then spotted on silica gel 60 TLC plates (Sigma-Aldrich). Plates were placed in a TLC tank containing a mobile phase to separate neutral lipids (hexane:diethyl ether:acetic acid at 80:20:1) or polar lipids (chloroform:methanol:water:acetic acid at 60:50:4:1). After separation, plates were air dried and stained with 0.2% Amido Black 10B (Sigma-Aldrich) dissolved in 1 M NaCl for 15–30 min (ref. 31). After staining, plates were rinsed with water and then washed several times with 1 M NaCl before drying overnight. Plates were imaged the next day on a Universal Hood II or EZ Imager GelDoc system (Bio-Rad).

**NBD–DAG acylation in HAP1 cells.** Unless otherwise specified, 25 μM NBD–DAG (Avanti Polar Lipids) was first conjugated to 0.125% fatty acid-free BSA in serum-free IMDM for 1 h at 37 °C. Cells were then incubated in this medium for 1 h before lipid extraction. TLC was done as described above. After drying, NBD fluorescence was monitored on a Universal Hood GelDoc system (Bio-Rad) using the fluorescein channel.

**Cell-free $^{14}$C-DAG acylation.** 4KO HAP1 cells (naive or expressing 3×HA-DIESL) were washed in ice-cold PBS, collected in ice-cold buffer (20 mM HEPES pH 7.4, 250 mM sucrose, 2 mM MgCl$_2$) and lysed by passing through a glass homogenizer 60 times on ice. Lysates were cleared by centrifugation at 600*g* for 10 min at 4 °C. Protein content was assayed and lysates were diluted to 2 mg ml$^{-1}$ and stored at −80 °C. Reconstitution assays were based on assays described previously[32] and were performed at the Radionuclides Centre of the Netherlands Cancer Institute according to its guidelines. First, 1.4 μl of 1.8 mM $^{14}$C-DAG, 3.6 μl buffer and 5.0 μl of 6.6 mg ml$^{-1}$ fatty acid-free BSA were incubated on a heat block for 1 h at 37 °C with shaking at 850 rpm. We then added 100 μl of the indicated lysate and assays were incubated at 37 °C with shaking at 850 rpm for the indicated time (the final $^{14}$C-DAG concentration was 50 μM and the total radioactivity per assay was 0.14 μCi). For some controls, the reaction was incubated instead on wet ice or the lysate was heat-inactivated for 20 min at 90 °C before being added to the reaction. Reactions were quenched by performing a modified Bligh and Dyer extraction[33]; 110 μl chloroform and 110 μl methanol were added to each reaction, mixed and centrifuged at 20,000*g* for 2 min at room temperature. The bottom organic layer was transferred to a new tube, dried and reconstituted in 8 μl ethanol and separated by neutral-lipid TLC as described above. Plates were

dried and imaged using a Typhoon FLA 9500 phosphorimager (General Electric, software v.1.1.0.187) after exposure to a BAS-TR2949 imaging plate (Fuji) for 3 days.

## Immunoprecipitation and crosslinking

**Formaldehyde crosslinking.** Δ*DIESL* and Δ*TMX1*Δ*DIESL* HAP1 cells reconstituted with 3×HA-DIESL or WT HeLa cells expressing 3×HA-DIESL were washed with PBS and incubated in 1% paraformaldehyde for 20 min at room temperature. Crosslinking reactions were quenched by incubation in 0.2 M glycine for 10 min. Cells were then lysed in RIPA buffer on ice as described above, sonicated, and protein concentrations were determined. Samples were separated by SDS–PAGE and immunoblotted as described above.

**Co-immunoprecipitation.** Δ*DIESL* HAP1 cells reconstituted with 3×HA-DIESL and grown to confluence in a six-well plate were lysed in 400 μl IP buffer (25 mM HEPES pH 7.0, 150 mM NaCl, 1% detergent (Tween-20 unless otherwise indicated) and protease- and phosphatase-inhibitor cocktails) on ice, sonicated and cleared by centrifugation. Then 250 μl of the lysate was incubated with 10 μl PBS-washed anti-HA magnetic beads (Pierce) overnight while being rotated at 4 °C. The next day, beads were washed three times in IP buffer before elution in SDS–PAGE sample buffer at 90 °C for 20 min. One-quarter of this eluate (along with the corresponding amount of input sample) were separated by SDS–PAGE and subjected to immunoblot analysis as described above.

## Deglycosylation assay

Δ*DIESL* HAP1 cells reconstituted with 3×HA-DIESL were lysed in buffer (25 mM Tris-HCl pH 7.5, 150 mM NaCl, 1% NP-40, protease-inhibitor cocktail) on ice, sonicated and protein levels were quantified as described above. *N*-glycans were removed using PNGase F (New England Bio-Labs) according to the manufacturer's instructions. Denatured lysates (1 μg μl$^{-1}$) were incubated for 15 min at 37 °C while being shaken at 850 rpm, in the presence or absence of 25 units μl$^{-1}$ PNGase F. Reactions were separated by SDS–PAGE and analysed by immunoblotting as described above.

## Subcellular fractionation and MAM purification

The MAM of the ER was purified from six confluent 15 cm plates of HAP1 cells, as described previously[34], with all steps performed on ice or at 4 °C. Cells were washed in ice-cold PBS, collected in 750 μl per plate ice-cold homogenization buffer (10 mM HEPES pH 7.4, 250 mM sucrose) and passed through a glass homogenizer 100 times. The homogenate was centrifuged for 5 min at 600*g*. The pellet was resuspended in 2 ml homogenization buffer, homogenized and centrifuged again. Both supernatants were pooled and centrifuged at 10,300*g* for 20 min. The supernatant was put aside and the pellet (crude mitochondria) was resuspended in 1 ml isolation buffer 1 (5 mM HEPES pH 7.4, 250 mM mannitol, 0.5 mM EGTA). Next, 500 μl of resuspended mitochondrial pellet was layered on top of 1.5 ml Percoll solution (25 mM HEPES pH 7.4, 225 mM mannitol, 1 mM EGTA, 30% (v/v) Percoll) in two ultracentrifuge tubes. Mitochondria were fractionated by centrifugation at 95,000*g* for 30 min using a TLS-55 swinging-bucket rotor (Beckman). Pure mitochondria and crude MAM layers were collected using a syringe and resuspended in four volumes of isolation buffer 1 and isolation buffer 2 (25 mM HEPES pH 7.4, 225 mM mannitol, 1 mM EGTA), respectively. Purified mitochondria were centrifuged at 10,500*g* for 10 min. The crude MAM was cleared by centrifugation at 6,300*g* for 10 min and then both the MAM and the postmitochondrial supernatant from earlier were centrifuged for 1 h at 100,000*g*. The pellets (microsome and pure MAM) were collected and the cytosolic supernatant was concentrated by centrifugal filtration (10 kDa filter, Amicon). Protein concentration was determined by BCA assay (Thermo Scientific) and fractions were stored at −80 °C before immunoblot analysis.

### Electron microscopy
Cells were prepared for transmission electron microscopy as previously described[35]. Grids were imaged on a Tecnai 12 G2 (ThermoFisher).

### Immunofluorescence and confocal microscopy
Cells were grown on glass coverslips and fixed in 4% formaldehyde in PBS for 15 min at room temperature and then washed three times in PBS. Cells were permeabilized with 0.1% Triton X-100 in PBS for 10 min and then washed three times in PBS and blocked in 1% BSA in PBS for 20 min. Cells were stained with primary antibodies, diluted in blocking buffer, for 1 h at room temperature. After three washes in blocking buffer, cells were stained with Alexa Fluor 488-, 568- and 647-conjugated secondary antibodies (Invitrogen A10042, A11011, A11004, A11008, A21245 and A31571) and/or neutral lipid dyes, diluted 1:500 in blocking buffer, for 1 h at room temperature. Cells were washed three times in PBS, counterstained with Hoechst 33342 (Invitrogen) diluted in PBS, and then washed three more times in PBS, before being mounted on glass slides using Aqua Poly/Mount (Polysciences). Cells were imaged by confocal laser-scanning microscopy using a Leica SP5 microscope with a 60× (1.4 NA) objective lens. Images were analysed using ImageJ (NIH).

### Flow cytometry
**BODIPY 493/503 measurements in fixed HAP1 cells.** HAP1 cells grown on 10 cm plates were collected by trypsinization and fixed in Fix Buffer I (BD Biosciences) for 10 min at 37 °C. Cells were pelleted, washed with FACS buffer (10% FBS in PBS), resuspended in FACS buffer and counted. Next, about 10 million cells were stained with 1 μg ml$^{-1}$ BODIPY 493/503 and 5 μg ml$^{-1}$ DAPI (Invitrogen), diluted in FACS buffer, for 1 h at room temperature. Cells were washed once in FACS buffer then passed through a 35 μm nylon mesh cell strainer into a FACS tube. Fluorescence was analysed on an LSR Fortessa (BD Biosciences) analytical flow cytometer, using lasers with wavelengths of 405 nm and 488 nm to detect DAPI and BODIPY 493/503, respectively. Data were analysed using FlowJo (BD Life Sciences). Fluorescence plots represent the fluorescent signal measured in single haploid G1 cells, as determined by DAPI fluorescence intensity.

**Mitochondrial measurements in live RPE1 cells.** RPE1 cells were cultured in the presence or absence of lipoproteins, supplemented with 50 μM oleic acid where indicated, and pulsed for 30 min with either 600 nM TMRM or 250 nM MitoTracker Red CM-H$_2$XROS in medium depleted of lipoproteins. In TMRM experiments, cells were then incubated in 150 mM TMRM for an additional 30 min, using 20 μM carbonyl cyanide m-chlorophenylhydrazone (CCCP) as a positive control for depolarization. Cells were collected by trypsinization and stored on ice in FACS buffer. Fluorescence (using a 561 nm laser) and data were analysed as described above. Fluorescence plots represent the fluorescent signal measured in single cells. Membrane potential was calculated by subtracting the fluorescence mean of depolarized (CCCP-treated) cells.

### ATP measurements
RPE1 cells were plated in 96-well plates and ATP levels were measured using CellTitre-Glo (Promega) according to the manufacturer's instructions. ATP levels were normalized by measuring protein content in a parallel plate by BCA assay (Thermo Scientific).

### Mice
**Generation and genotyping.** Animal experiments were performed with the approval of the National Ethics Committee for Animal Experiments of the Netherlands, in accordance with the relevant guidelines and regulations (including laboratory and biosafety regulations). Frozen embryos (morula stage, from C57/BL6J mice) carrying a single disruptive allele in *Diesl* (also known as *Tmem68*) tm1a(EUCOMM)Wtsi (herein referred to as *gt(lacZ-neo)*), were purchased from the European Conditional Mouse Mutagenesis Program (EUCOMM). After thawing, embryos were developed into blastocysts overnight in KSOM medium in an incubator at 37 °C, and were then implanted into C57/BL6N foster female mice and carried to term. Mice were genotyped by PCR, using primer combinations to detect the WT (5′-GCTCCCTTCCATTTACTCTG-3′ and 5′-CCGGTGAGATAGCTAACAAG-3′) and mutant (5′-CTTATCAT GTCTGGATCCGG-3′ and 5′-CCGGTGAGATAGCTAACAAG-3′) alleles. Because C57/BL6J mice have a deletion in the *Nnt* gene that influences DIESL protein levels[36], only mice with 6N alleles at the *Nnt* locus were used in this study, and this was monitored by PCR, using primer combinations to detect the 6N (5′-GGGCATAGGAAGCAAATACCAAGTTG-3′ and 5′-GTAGGGCCAACTGTTTCTGCATGA-3′) and 6J (5′-GTGGAATTCCGC TGAGAGAACTCTT-3′ and 5′-GTAGGGCCAACTGTTTCTGCATGA-3′) alleles. To generate mice for analysis, mice heterozygous for the mutant *Diesl* allele were intercrossed to generate knockout and control mice. Mice were born at the expected Mendelian ratio ($P = 0.4279$, chi-squared test, $n = 341$ mice) and maintained on a diet of RM3-SAFE (4.2% fat, Special Diet Services). Mice were maintained in a certified animal facility at 21 °C and 55% humidity in 12 h:12 h light:dark cycles. No sample-size calculations were performed. No randomization was performed because animals were assayed based on genotype. In experiments where subjectivity could be introduced, the experimenter was blinded to the details.

**PCR with reverse transcription.** RNA from mouse liver tissue was isolated using the Qiagen RNeasy Mini Kit, and cDNA libraries were synthesized with SuperScript III reverse transcriptase (Invitrogen) using random hexamers. The *DIESL* transcript was detected by PCR using a primer pair spanning exons 4 and 5 (5′-GAGCCATCCCCATAGACTTTT ACTACTTC-3′ and 5′-CCCGGTGAGATAGCTAACAAGTGAC-3′). In the knockout, the disruptive cassette was integrated between these two exons. The *ACTB* transcript was used as a control and was amplified using the following primer pair: 5′-ATCCTGACCCTGAAGTACCCCA-3′ and 5′-CCTCTCAGCTGTGGTGGTGAAGCTGTAGCCACGCT-3′. Although DIESL has previously been reported to be a brain-specific protein[19], we readily detected *DIESL* expression in the mouse liver, in agreement with proteomic data[36]. Furthermore, expression data from the Human Protein Atlas indicates that *DIESL* expression could be detected in all tissues that were tested[37].

**Histology.** Tissues and organs were collected and fixed in EAF fixative (ethanol:acetic acid:formaldehyde:saline at 40:5:10:45 v:v) and embedded in paraffin. Sections were prepared at 2 μm thickness from the paraffin blocks and stained with haematoxylin and eosin according to standard procedures. The sections were reviewed with a Zeiss Axioskop2 Plus microscope (Carl Zeiss Microscopy) and images were captured with a Zeiss AxioCam HRc digital camera and processed with AxioVision 4 software (both from Carl Zeiss Vision).

**Indirect calorimetry.** Indirect calorimetry was done using fully automated metabolic cages (LabMaster, TSE systems). Mice were housed individually in a 12 h:12 h light:dark cycle (light, 07:00–19:00 hours) with ad libitum access to water and a standard chow diet (10% fat, 23% protein and 67% carbohydrate; V1554-703, Ssniff Spezialdiäten). After at least 24 h of acclimatization, O$_2$ consumption ($V_{O_2}$), CO$_2$ production ($V_{CO_2}$) and caloric intake were measured for at least three consecutive days followed by 12 h fasting and refeeding. The RER was calculated as $V_{CO_2}/V_{O_2}$. Glucose oxidation was calculated using the formula $((4.585 \times V_{CO_2}) - (3.226 \times V_{O_2})) \times 4$ and fat oxidation was calculated as $((1.695 \times V_{O_2}) - (1.701 \times V_{CO_2})) \times 9$. Values were corrected for the lean mass of each individual mouse.

**Body composition.** Fat mass and lean tissue mass were assessed at several time points in non-anaesthetized mice using NMR (MiniSpec LF90 BCA-analyser).

## Shotgun lipidomics

**Sample preparation.** 4KO ($\Delta DGAT1\Delta DGAT2\Delta DIESL\Delta TMX1$) HAP1 cells, reconstituted with WT or H130A DIESL, were cultured in complete medium. Prior to collection, cells were cultured in serum-free IMDM for 30 min. Cells were then collected by trypsinization, washed twice with PBS and 3 million cells (in triplicate) were pelleted and stored at −80 °C. For *E. coli* samples, 2 ml of induced culture (as described above, in triplicate) was pelleted and incubated overnight at −80 °C. The next day, bacterial pellets were thawed and incubated in 25 μg ml$^{-1}$ lysozyme in PBS for 30 min. Cells were pelleted, washed in PBS and pelleted again before resuspension in distilled water. Cells were sonicated at 60% amplitude for 5 min, in continuous cycles of 5 s on and 25 s off. The insoluble fraction was pelleted by centrifugation and the lysate (supernatant) was quantified for protein and stored at −80 °C. For mouse tissue samples, tissues were collected from mice fed ad libitum and stored at −80 °C. Tissues were then thawed on ice, weighed and homogenized in PBS (50 mg ml$^{-1}$ tissue) using a glass homogenizer with 20 and 40 strokes for brain and liver tissue, respectively. Homogenates were diluted 10-fold further in PBS and homogenized again with 20 strokes. Homogenates (5 mg ml$^{-1}$) were stored at −80 °C. Lipid abundancies are expressed either in weight (where indicated, corrected for the amount of protein in the sample) or as a molar percentage of the total lipid species in the sample (mol%).

**Lipid extraction for MS lipidomics.** MS-based lipid analysis was performed by Lipotype as described[38]. Lipids were extracted using a two-step chloroform–methanol procedure[39]. Samples were spiked with internal lipid standard mixture containing: cardiolipin 14:0/14:0/14:0/14:0 (CL), ceramide 18:1;2/17:0 (Cer), DAG 17:0/17:0, hexosylceramide 18:1;2/12:0 (HexCer), lyso-phosphatidate 17:0 (LPA), lyso-phosphatidylcholine 12:0 (LPC), lyso-phosphatidylethanolamine 17:1 (LPE), lyso-phosphatidylglycerol 17:1 (LPG), lyso-phosphatidylinositol 17:1 (LPI), lyso-phosphatidylserine 17:1 (LPS), phosphatidate 17:0/17:0 (PA), phosphatidylcholine 17:0/17:0 (PC), phosphatidylethanolamine 17:0/17:0 (PE), phosphatidylglycerol 17:0/17:0 (PG), phosphatidylinositol 16:0/16:0 (PI), phosphatidylserine 17:0/17:0 (PS), cholesterol ester 20:0 (CE), sphingomyelin 18:1;2/12:0;0 (SM) and TAG 17:0/17:0/17:0. Chain composition is given as length:saturation;oxidation. After extraction, the organic phase was transferred to an infusion plate and dried in a speed vacuum concentrator. The first-step dry extract was resuspended in 7.5 mM ammonium acetate in chloroform:methanol:propanol at 1:2:4 (v:v:v) and the second-step dry extract in 33% ethanol solution of methylamine:chloroform:methanol at 0.003:5:1 (v:v:v). All liquid-handling steps were performed using the Hamilton Robotics STARlet robotic platform with the anti-droplet control feature for organic solvents pipetting.

**MS data acquisition.** Samples were analysed by direct infusion on a QExactive mass spectrometer (Thermo Scientific) equipped with a TriVersa NanoMate ion source (Advion Biosciences). Samples were analysed in both positive and negative ion modes with a mass-to-charge ratio resolution ($Rm/z$) of $Rm/z = 200 = 280,000$ for MS and $Rm/z = 200 = 17,500$ for tandem MS (MS/MS) experiments, in a single acquisition. MS/MS was triggered by an inclusion list encompassing corresponding MS mass ranges scanned in 1-Da increments[40]. Both MS and MS/MS data were combined to monitor CE, DAG and TAG ions as ammonium adducts; PC, PC O- as acetate adducts; and CL, PA, PE, PE O-, PG, PI and PS as deprotonated anions. MS alone was used to monitor LPA, LPE, LPE O-, LPI and LPS as deprotonated anions; Cer, HexCer, SM, LPC and LPC O- as acetate adducts.

**Data analysis and postprocessing.** Data were analysed by Lipotype with in-house lipid-identification software based on LipidXplorer[41,42].

Data postprocessing and normalization were performed using an in-house data management system. Only lipid identifications with a signal-to-noise ratio greater than 5 and a signal intensity 5-fold higher than in corresponding blank samples were considered for further data analysis.

## Homology modelling and data analysis

The DIESL AlphaFold[22] model was accessed via the EMBL–EBI portal (https://alphafold.ebi.ac.uk/) and visualized using PyMOL. Data wrangling, statistical analyses and plot generation were done using Prism (GraphPad Software) and RStudio (https://www.rstudio.com/).

## Statistics and reproducibility

The number of replicates is indicated in the figure legends. All attempts at replication were successful.

## Reporting summary

Further information on research design is available in the Nature Portfolio Reporting Summary linked to this article.

## Data availability

Sequencing data and screening data are available, respectively, at the NCBI Sequence Read Archive (accession numbers SAMN35570720, SAMN35570721 and SAMN35570722) and an interactive web application (https://phenosaurus.nki.nl/). Source data are provided with this paper.

## Code availability

Analysis pipelines for haploid screens are available at https://github.com/BrummelkampResearch.

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

**Acknowledgements** T.R.B. is an Oncode investigator (oncode.nl). G.L.M. is a Banting postdoctoral fellow (Canadian Institutes of Health Research). J.W.J. is supported by a grant from the Netherlands Organization for Scientific Research (VICI grant 016.176.640). Research at the Netherlands Cancer Institute is supported by institutional grants from the Dutch Cancer Society and the Dutch Ministry of Health, Welfare and Sport. We thank P. Borst, T. Sixma and T. Fon for reading the manuscript, A. Perrakis for advice concerning homology modelling, D. Verwoerd for help with radioactivity assays, N. Kloosterhuis for biotechnical assistance and members of the Brummelkamp group for discussions and reading the manuscript. Electron micrographs were acquired at the Electron Microscopy Centre Amsterdam (Amsterdam University Medical Center).

**Author contributions** G.-L.M. and T.R.B. conceived the project and analysed the data. G.-L.M., M.-L.O., C.R.C.V., B.D.E., R.A.O., N.J.B., M.D., L.G.v.d.H. and T.v.Z. performed experiments and collected data. G.-L.M. and T.R.B. performed radioactivity assays. H.J. and J.-Y.S. performed electron microscopy and mouse pathology, respectively. Mouse embryo implantation was done by R.A. and P.K. Phenotypic and metabolic characterization of *Diesl*-knockout mice was done by C.R.C.V., T.v.Z. and J.W.J. The manuscript was written by G.-L.M. and T.R.B.

**Competing interests** T.R.B. is a co-founder and scientific advisor of Scenic Biotech. G.-L.M. and T.R.B. are inventors on a patent application related to this work.

**Additional information**

**Correspondence and requests for materials** should be addressed to Gian-Luca McLelland or Thijn R. Brummelkamp.

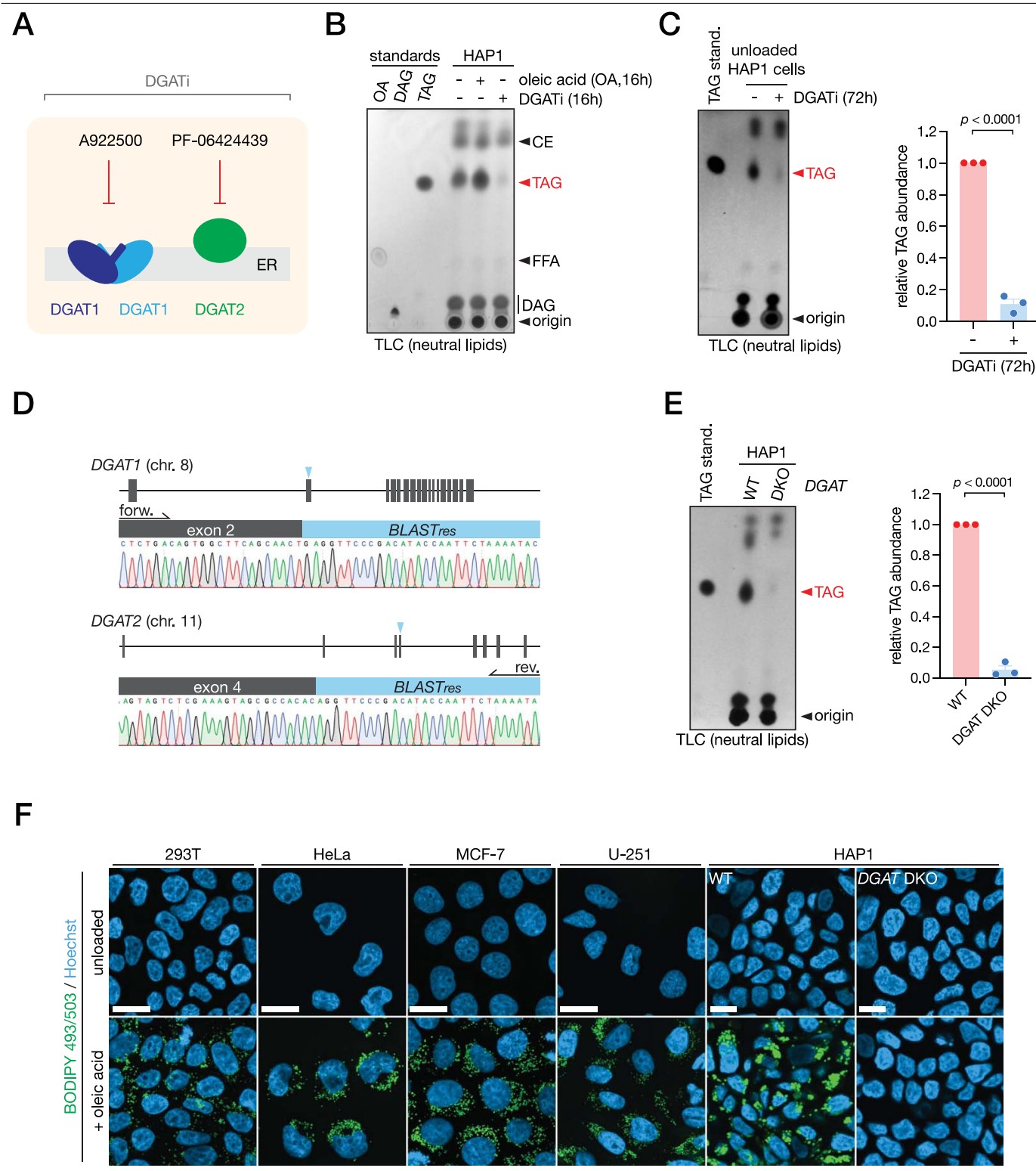

**Extended Data Fig. 1 | Characterization of TAG synthesis and lipid storage in HAP1 cells and other cell lines. a**, Schematic representation of DGAT1- and DGAT2-specific inhibitors. **b**, TLC separation of neutral lipids from HAP1 cells treated with 50 μM oleic acid or 10 μM (each) DGATi for 16 h. CE, cholesteryl ester; TAG, triacylglycerol; DAG, diacylglycerol; FFA, free fatty acid; OA, oleic acid. **c**, TLC analysis of neutral lipids (left) and quantification of TAG (right) from HAP1 cells treated with 10 μM (each) DGATi for 72 h. Bars represent mean ± SEM of $n$ = 3 independent experiments (Student's $t$ test, two-tailed). **d**, Sequencing peaks of the mutated *DGAT1* and *DGAT2* loci from a HAP1 *DGAT*

DKO clone where a blasticidin resistance (BLASTres) cassette was integrated at each locus. Arrows represent the direction of amplification. **e**, TLC analysis of neutral lipids (left) and TAG quantification (right) from HAP1 WT and *DGAT* DKO cells. Bars represent mean ± SEM of $n$ = 3 independent experiments (Student's $t$ test, two-tailed). **f**, Panel of cell lines treated with 200 μM oleic acid for 24 h. Neutral lipid loading is visualized by lipid droplets stained with BODIPY 493/503. Scale bars for HAP1 cells, 10 microns; scale bars for other cell lines, 20 microns.

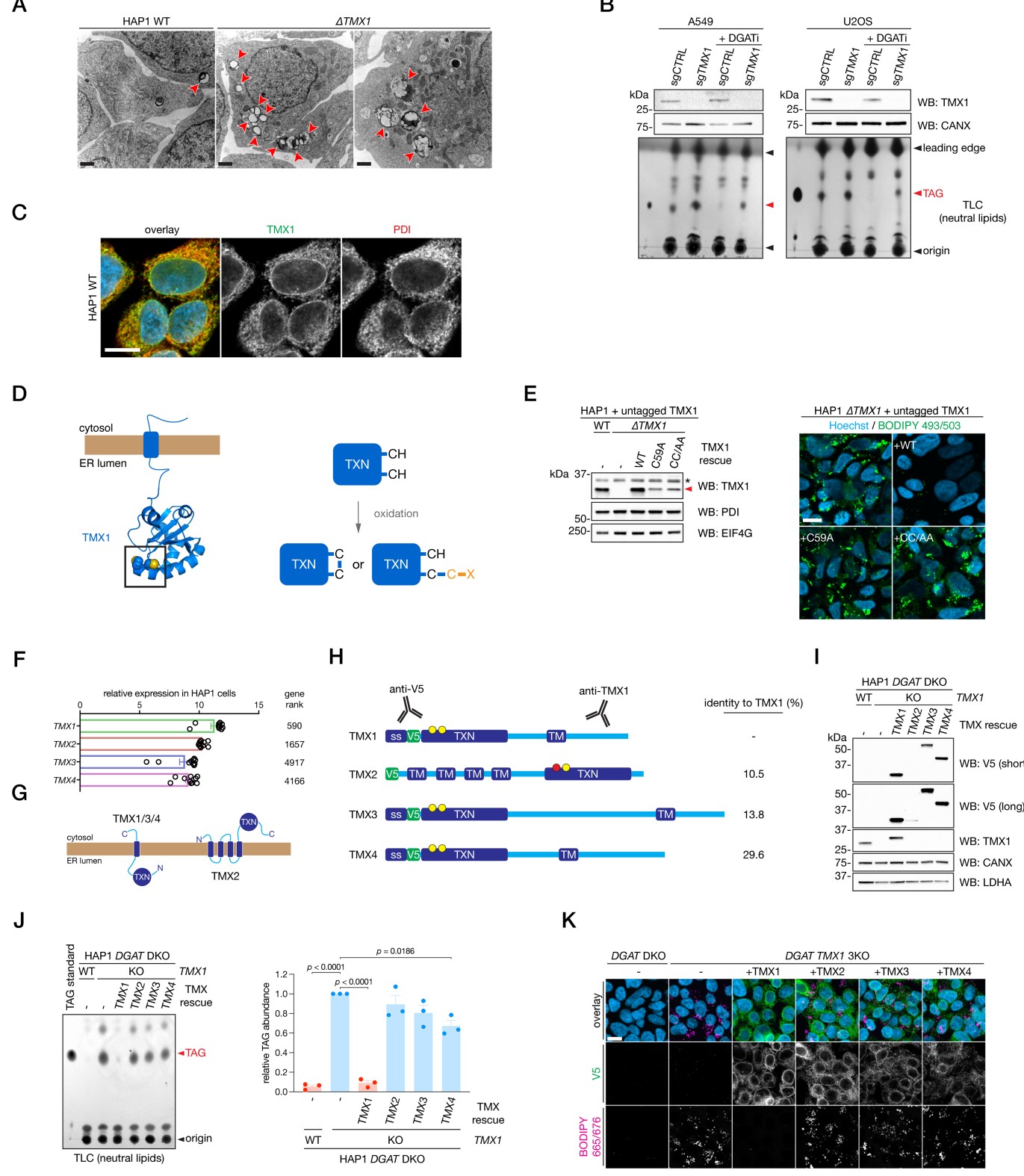

**Extended Data Fig. 2** | See next page for caption.

**Extended Data Fig. 2 | TMX1 uniquely controls TAG accumulation.**
**a**, Ultrastructural analysis of lipid droplets (red arrowheads) in HAP1 WT and *TMX1* KO cells. Scale bars, 1 micron. **b**, TLC analysis of neutral lipids and Western blot analysis of proteins extracted from A549 (left) and U2OS (right) cells, transduced with Cas9 and a control sgRNA (sgCTRL) or sgRNA targeting *TMX1* (sgTMX1), treated with 10 μM (each) DGATi for 72 h. **c**, Confocal imaging of TMX1 (green) and PDI (ER marker, red) in HAP1 cells. Hoechst 33342, blue; scale bar, 10 microns. **d**, Representation of TMX1 as the solution structure of its thioredoxin domain (PDB ID 1X5E) anchored to the membrane (left; the black box highlights the redox cysteines) as well as the TMX1 redox cycle (right). **e**, Immunoblot analysis from (left) and imaging of lipid droplets in (right) HAP1 *ΔTMX1* cells rescued with untagged TMX1 WT, C59A or C56A/C59A (CC/AA) constructs. * indicates a non-specific band. Scale bar, 10 microns. **f**, Relative expression of TMX family members in HAP1 cells as determined by RNAseq. **g**, Topological representation of TMX family members. **h**, Domain structure of TMX family members. Location of antibody binding and sequence homology to TMX1 are indicated. Yellow and red circles represent catalytic cysteine and serine residues, respectively. ss, signal sequence; V5, simian virus 5 epitope tag; TXN, thioredoxin domain; TM, transmembrane helix. **i**, Immunoblot analysis of HAP1 *DGAT TMX1* 3KO cells expressing V5-tagged TMX family members. **j**, TLC analysis of neutral lipids (left) and quantification (right) in cells from i. Bars represent mean ± SEM of *n* = 3 independent experiments (one-way ANOVA, Bonferroni correction). **k**, Confocal imaging of lipid droplets (stained with BODIPY 665/676) in cells from **i**. Scale bar, 10 microns.

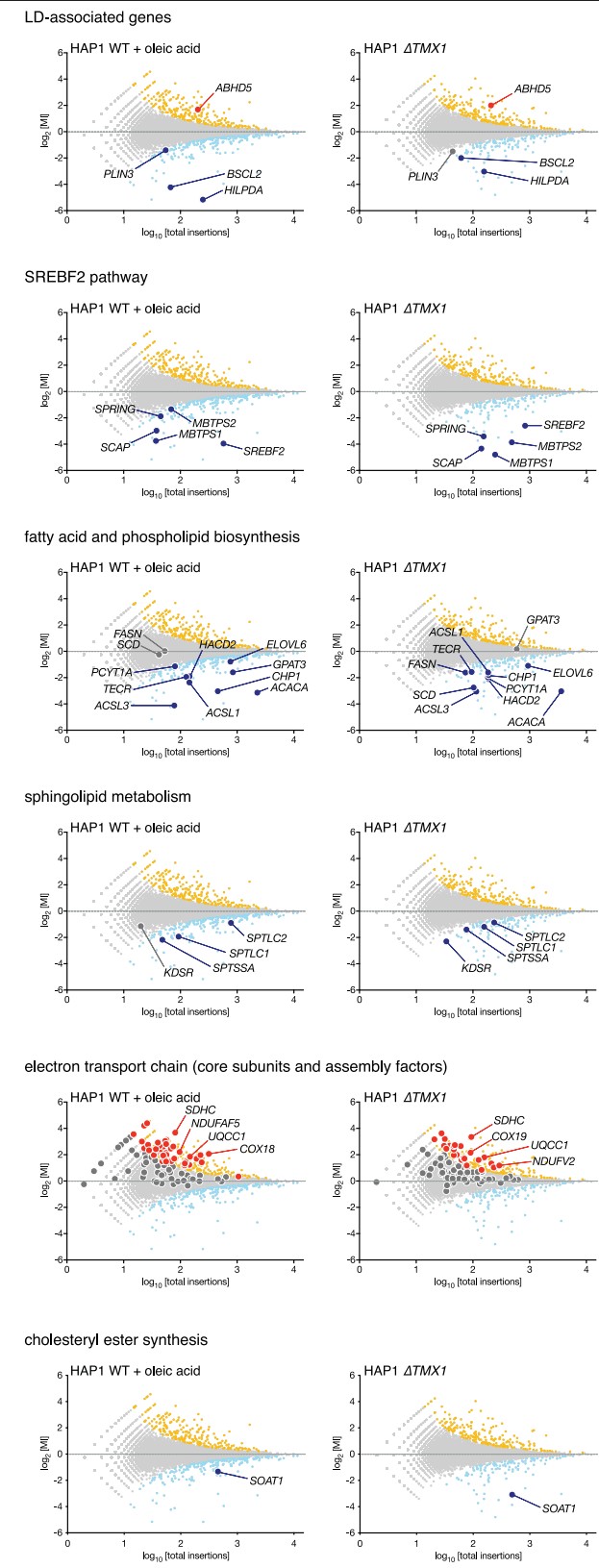

**Extended Data Fig. 3 | Genetics of lipid droplet regulation.** Genes related to the indicated cellular processes are labeled in the lipid droplet screens performed in oleic acid-loaded HAP1 WT cells (left column) and HAP1 *ΔTMX1* cells (right column). Significant (FDR-corrected *p* < 0.05) positive and negative regulators are coloured light blue and orange, respectively. Genes of interest are coloured dark blue, red and dark grey.

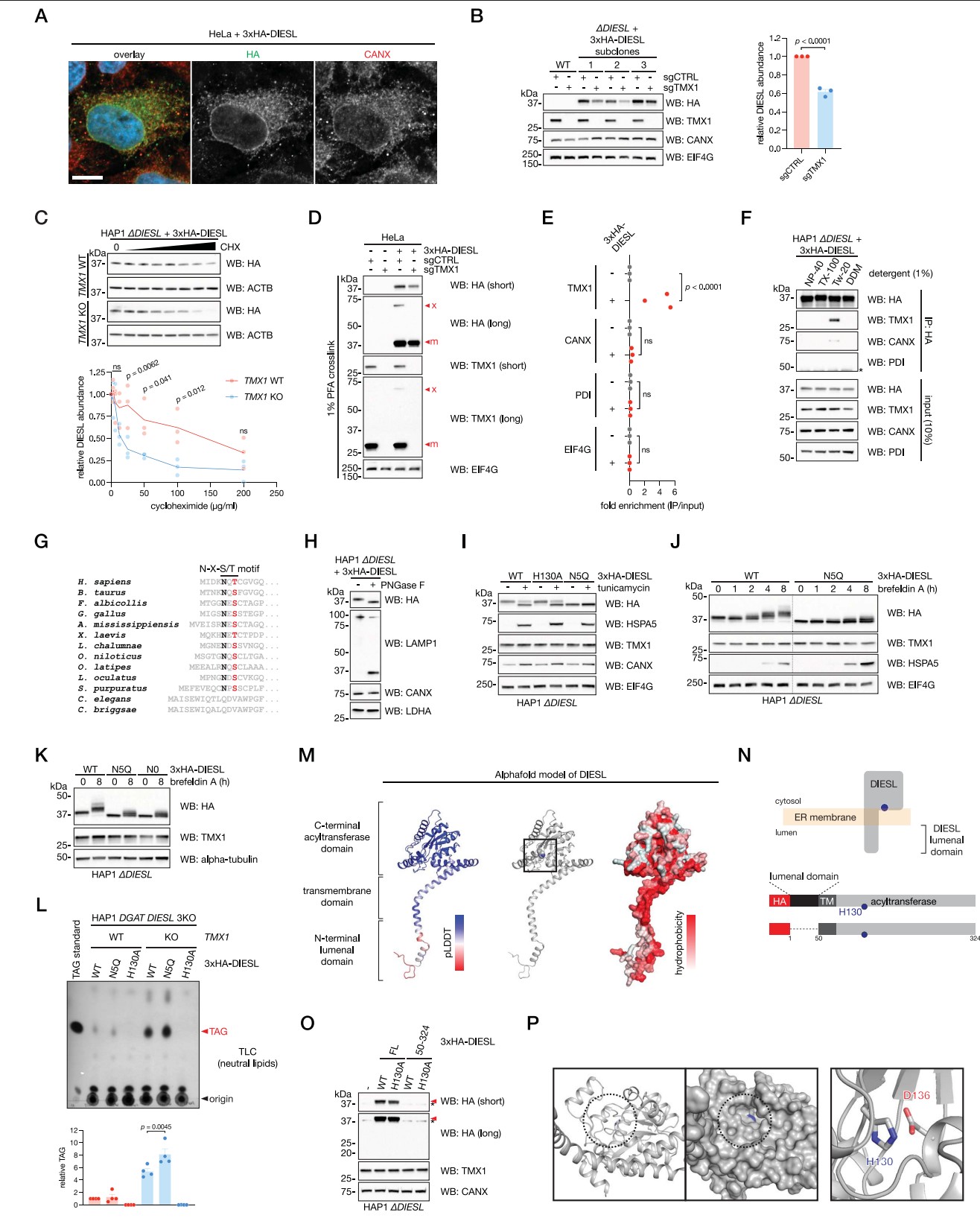

**Extended Data Fig. 4 | See next page for caption.**

**Extended Data Fig. 4 | DIESL is an ER membrane glycosylated acyltransferase that physically interacts with TMX1. a**, Localization of 3xHA-DIESL (green) in HeLa cells co-stained with the ER marker CANX (red). Hoechst 33342, blue; scale bar, 10 microns. **b**, HAP1 cells rescued with 3xHA-DIESL were subcloned, and three independent clones were then transduced with Cas9 and a control sgRNA (sgCTRL) or an sgRNA targeting *TMX1* (sgTMX1). DIESL levels were assessed by immunoblot (left) and quantified (right). Bars represent mean ± SEM (Student's t test, two-tailed) of $n$ = 3 independent cell lines from one representative experiment. **c**, TMX1-dependent DIESL half-life was analyzed by a 24 hour-long cycloheximide chase in HAP1 cells rescued with 3xHA-DIESL, with or without *TMX1* deletion. DIESL protein levels were analyzed by immunoblot (top) and quantified (bottom), normalized to ACTB. The cycloheximide concentration ranged from 200 to 3.125 μg/ml, using successive two-fold dilutions. Bars represent mean ± SEM of $n$ = 3 independent experiments (two-way ANOVA, Bonferroni correction; ns, not significant). **d**, HeLa cells expressing 3xHA-DIESL were transduced with Cas9 and a control sgRNA (sgCTRL) or an sgRNA targeting *TMX1* (sgTMX1). Cells were crosslinked with PFA prior to lysis and immunoblot analysis. x and m indicate the cross-linked TMX1-DIESL complex and corresponding monomer, respectively. **e**, Quantification of the 3xHA-DIESL coimmunoprecipitation of TMX1 in HAP1 cells shown in Fig. 3b. Fold enrichment over input is shown for TMX1, as well as CANX, PDI an EIF4G for $n$ = 3 independent experiments (two-way ANOVA, Bonferroni correction; ns, not significant). **f**, TMX1 was co-immunoprecipitated from HAP1 cells rescued with DIESL in a buffer containing the indicated detergent (*indicates an antibody chain). Tw-20, Tween-20; DDM, n-dodecyl-beta-maltoside. **g**, An N-glycosylation motif (N-X-S/T) is conserved at the DIESL N-terminus. **h**, Deglycosylation of DIESL N-glycans by PNGaseF. Glycosylation was assessed by immunoblot, where LAMP1 served as a positive control. **i**, DIESL is glycosylated at N5 at the steady-state. HAP1 *ΔDIESL* cells reconstituted with the indicated 3xHA-DIESL construct were treated with 1 μg/ml tunicamycin (an inhibitor of N-linked glycosylation) for 16 h. The lower and higher HA bands represent the unglycosylated and N-glycosylated forms of 3xHA-DIESL, respectively. **j**, HAP1 *ΔDIESL* cells rescued with 3xHA-DIESL were treated with 5 μg/ml brefeldin A for the indicated period of time, and DIESL glycosylation status was assessed by immunoblot. The N5Q mutant is not glycosylated. **k**, N5Q is the single N-glycosylation site, as this construct and a mutant lacking arginines (N0, where all arginines have been mutated to glutamine) show an identical band pattern upon brefeldin A treatment. **l**, TLC analysis of neutral lipids (top) and quantification of TAG (bottom) in HAP1 *DGAT DIESL* 3KO cells, with or without additional ablation of *TMX1*, reconstituted with the indicated 3xHA-DIESL construct. Bars represent mean ± SEM of $n$ = 4 independent experiments (one-way ANOVA, Bonferroni correction). **m**, AlphaFold model of human DIESL (Q96MH6) colored by pLDDT (left), active site histidine (centre) or hydrophobicity (right). **n**, Schematic of DIESL in the ER membrane with the lumenal domain indicated (top) and depiction of DIESL constructs with or without the 49 amino acid-containing, N-terminal lumenal domain (bottom). **o**, Immunoblot of HAP1 ΔDIESL cells reconstituted with full-length DIESL (FL) or DIESL lacking the lumenal domain (50-324), each with (WT) or without (H130A) an intact active site. *indicates a non-specific band. **p**, Depiction of the DIESL catalytic pocket as modeled by Alphafold (left) with H130 in blue, depicting both the backbone (left) and the surface (right), as viewed from the surface of the membrane, as well as the DIESL catalytic dyad (right).

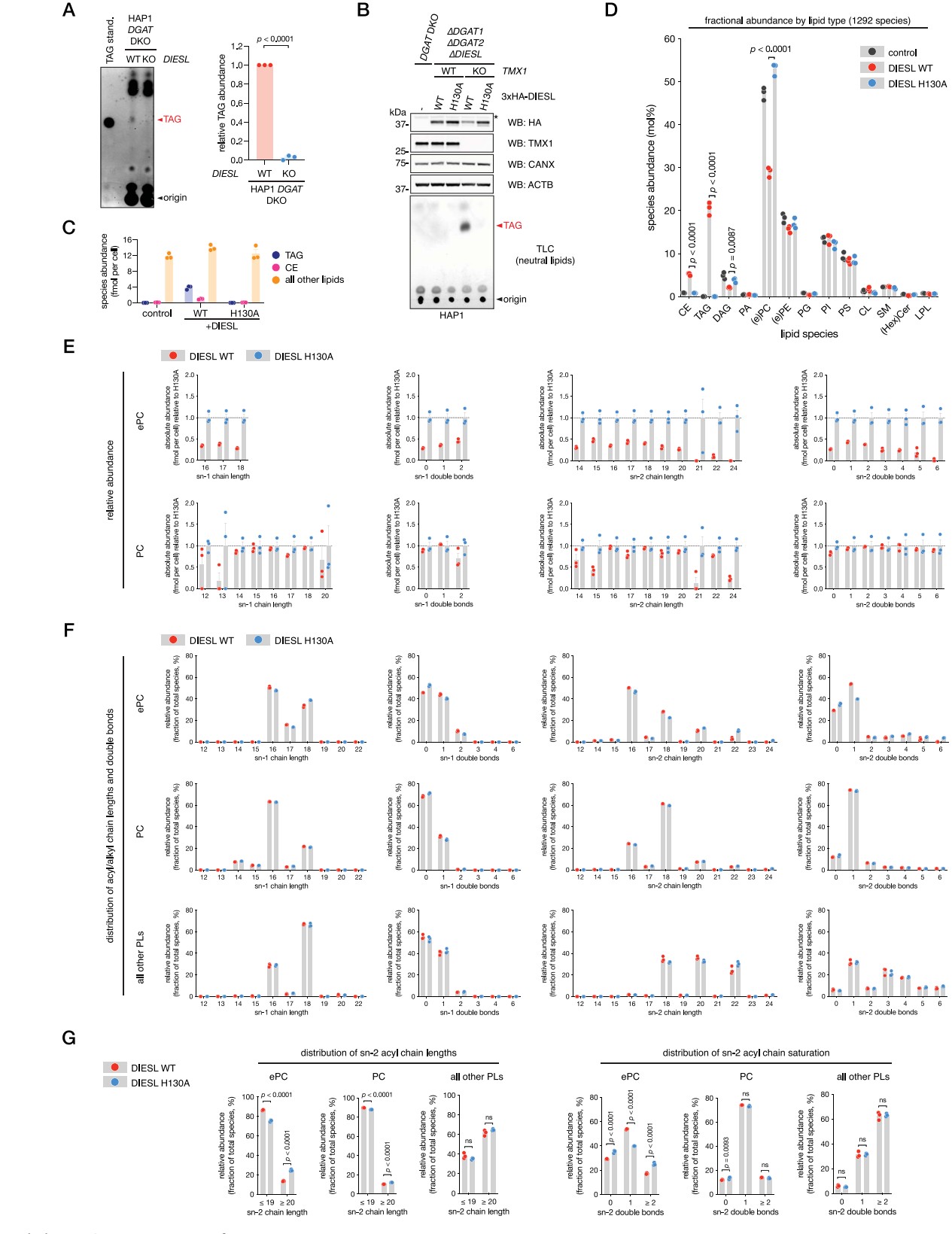

**Extended Data Fig. 5** | See next page for caption.

**Extended Data Fig. 5 | Perturbation of the lipidome by DIESL. a**, TLC analysis of neutral lipids (left) and TAG quantification (right) in HAP1 *DGAT* DKO and *DGAT DIESL* 3KO cells. Bars represent mean ± SEM of $n$ = 3 independent experiments (Student's $t$ test, two-tailed). **b**, TLC separation of neutral lipids and immunoblot analysis of HAP1 cell lines reconstituted with 3xHA-DIESL WT or H130A (* indicates a non-specific band). **c**, Absolute lipid abundance in HAP1 4KO cells *(ΔDGAT1ΔDGAT2ΔDIESLΔTMX1)* reconstituted with 3xHA-DIESL WT or H130A as determined by shotgun lipidomics. CE, cholesteryl ester; TAG, triacylglycerol. Bars represent mean ± SEM of $n$ = 3 independent samples. **d**, Lipidome of 4KO cells, either naïve (control) or expressing WT or H130A DIESL. Each species is represented as percent of the total lipid abundance in the respective samples, where 1292 unique lipid species were detected. Bars represent mean ± SEM of $n$ = 3 biologically-independent cell lines examined in a single experiment (two-way ANOVA, Bonferroni multiple-comparison correction; all significant differences between WT and H130A cells are indicated). CE, cholesteryl ester; TAG, triacylglycerol; DAG, diacylglycerol; PA, phosphatidic acid; (e)PC, (ether-linked) phosphatidylcholine;

(e)PE, (ether-linked) phosphatidylethanolamine; PG, phosphatidylglycerol; PI, phosphatidylinositol; PS, phosphatidylserine; CL, cardiolipin; SM, sphingomyelin; (Hex)Cer, (hexosyl)ceramide; LPL, lysophospholipid. **e**, Relative abundance of ePC and PC species (grouped by acyl chain length and saturation) in HAP1 4KO cells expressing WT and H130A DIESL, expressed as a fraction of the levels in H130A cells. Bars represent mean ± SEM of $n$ = 3 biologically-independent cell lines examined in a single experiment. **f**, Distribution of ePC, PC and all other phospholipid (PL) species (grouped by acyl chain length and saturation) in HAP1 4KO cells expressing WT and H130A DIESL, expressed as a percentage of the total amount of that species detected in that condition. Bars represent mean ± SEM of $n$ = 3 biologically-independent cell lines examined in a single experiment. **g**, Distribution of ePC, PC and all other phospholipid (PL) species, grouped by sn-2 chain length (left) and saturation (right), in HAP1 4KO cells expressing WT and H130A DIESL. Bars represent mean ± SEM of $n$ = 3 biologically-independent cell lines examined in a single experiment (two-way ANOVA, Bonferroni multiple-comparison correction; ns, not significant).

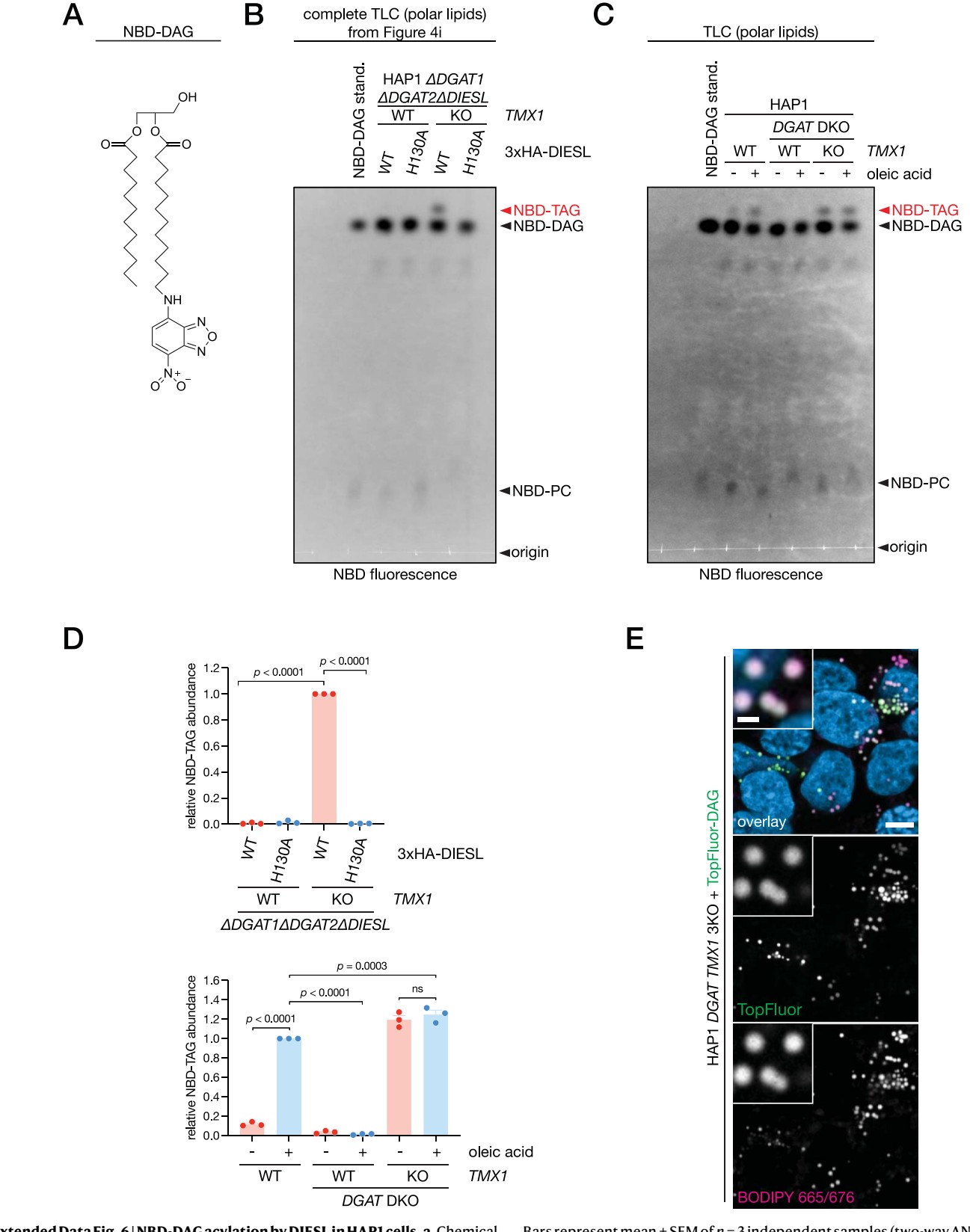

**Extended Data Fig. 6 | NBD-DAG acylation by DIESL in HAP1 cells. a**, Chemical structure of NBD-DAG. **b**, Complete TLC plate from Fig. 4i. **c**, Analysis of NBD-DAG acylation (by NBD fluorescence) by TLC separation of polar lipids. HAP1 WT, *DGAT* DKO and *DGAT DIESL* 3KO cells were incubated with 25 µM NBD-DAG in the presence or absence of 50 µM oleic acid for one hour prior to lipid extraction. **d**, Quantification of NBD-TAG in TLCs from **b** (top) and **c** (bottom).

Bars represent mean ± SEM of *n* = 3 independent samples (two-way ANOVA, Bonferroni correction; ns, not significant). **e**, Confocal imaging of HAP1 *DGAT TMX1* 3KO labeled with 50 µM TopFluor-DAG. After fixation, membranes were extracted with 0.1% TX-100 and lipid droplets were subsequently stained with BODIPY 665/676. Hoechst 33342, blue; scale bar, 5 and 1 microns.

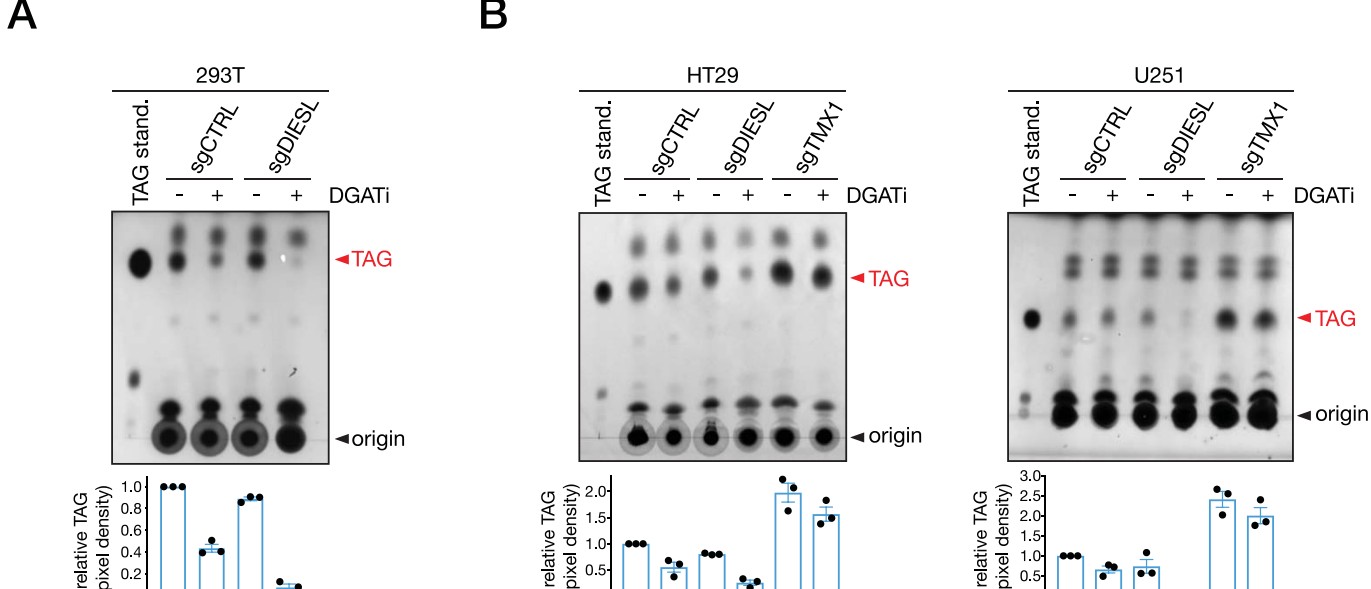

**Extended Data Fig. 7 | Steady-state DIESL activity. a**, TLC analysis of neutral lipids extracted from 293T cells treated with 10 μM (each) DGATi for 48 h. Cells were edited using CRISPR/Cas9 with a sgRNA targeting *DIESL* (sgDIESL) or a control sgRNA (sgCTRL). Bars represent mean ± SEM of TAG intensity from *n* = 3 independent experiments. **b**, TLC analysis of neutral lipids extracted from HT29 (left) and U251 (right) cells, treated with 10 μM (each) DGATi for 48 h. Cells were edited as in **a** with sgRNAs targeting *DIESL* (sgDIESL) or *TMX1* (sgTMX1), as well as a control sgRNA (sgCTRL). Bars represent mean ± SEM of TAG intensity from *n* = 3 independent experiments.

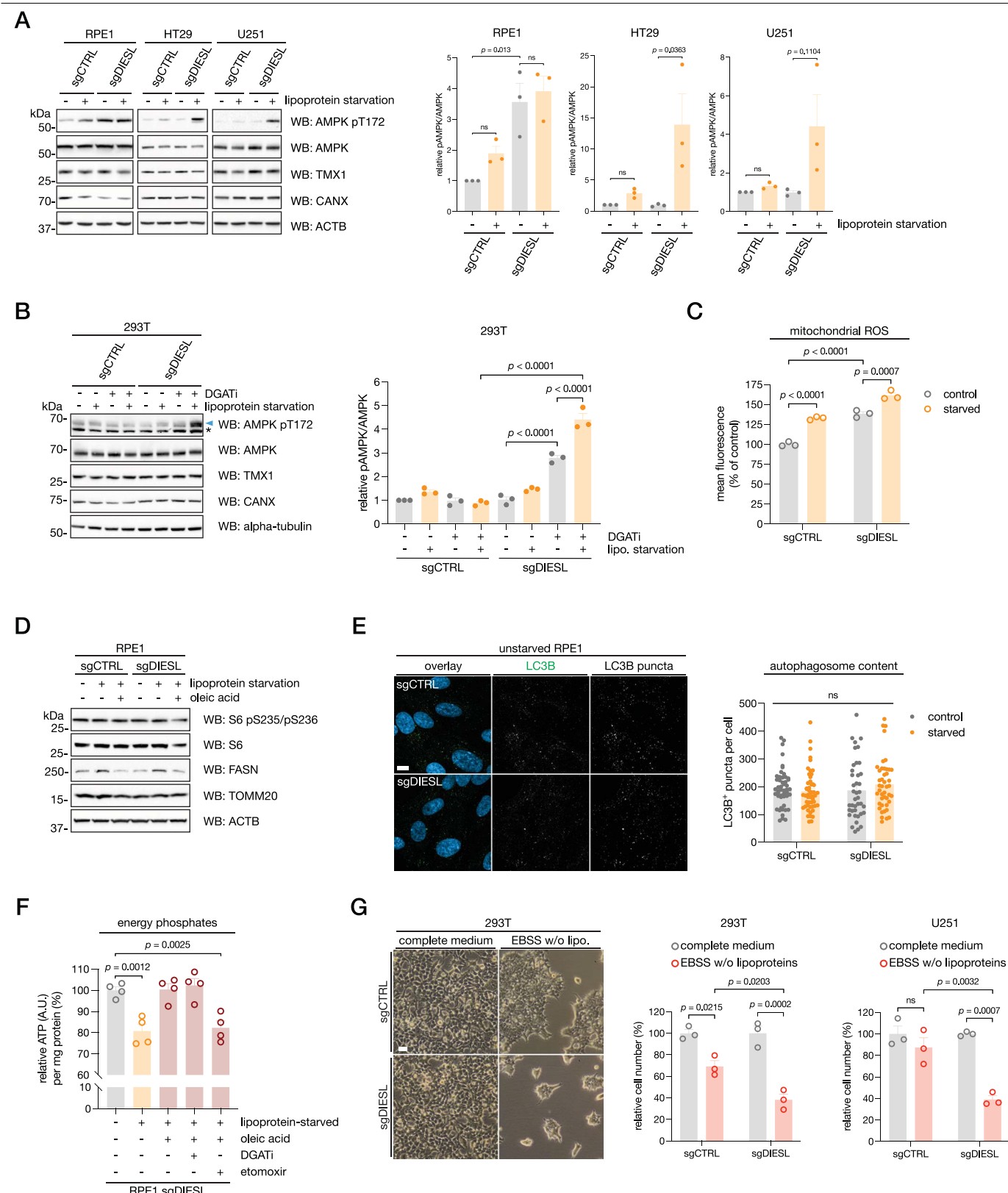

**Extended Data Fig. 8** | See next page for caption.

**Extended Data Fig. 8 | DIESL maintains mitochondrial function.**
**a**, Immunoblot analysis (left) and quantification (right) of phosphorylated AMPK in the indicated cell lines transduced with Cas9 and either a control sgRNA ("sgCTRL") or an sgRNA targeting *DIESL* ("sgDIESL"). Cells were cultured in complete medium (DMEM with lipoproteins) or in lipoprotein-depleted medium for 24 h (RPE1 cells) or 48 h (HT29 and U251 cells). Bars represent mean ± SEM of $n$ = 3 independent experiments (two-way ANOVA, Bonferroni correction; ns, not significant). **b**, Immunoblot analysis (left) and quantification (right) of AMPK phosphorylation in 293T cells, transduced with Cas9 and the indicated sgRNA, cultured as in a for 48 h, additionally treated with 10 μM (each) DGATi where indicated. * represents a non-specific band. Bars represent mean ± SEM of $n$ = 3 independent experiments (two-way ANOVA, Bonferroni correction). **c**, Mitochondrial ROS measurements in RPE1 cells cultured in lipoprotein-replete or -depleted conditions for 24 h. Cells were stained with MitoTracker Red CM-H$_2$XROS and fluorescence was analyzed by flow cytometry. Bars represent mean ± SEM of $n$ = 3 independent experiments (two-way ANOVA, Bonferroni correction). **d**, Immunoblot analysis of S6 phosphorylation in RPE1 cells cultured in complete medium or lipoprotein-depleted medium with or without 50 μM oleic acid for 24 h. FASN levels are a control for lipid starvation. **e**, Imaging (left) and quantification (right) of autophagosome content of RPE1 cells cultured in lipoprotein-replete or -depleted conditions for 24 h. Autophagosomes were identified by LC3B puncta (green), which were thresholded. Hoechst 33342, blue. Scale bar, 10 microns. Bars represent mean ± SEM of $n$ = 40 to 51 cells (two-way ANOVA, Bonferroni correction; ns, not significant). **f**, Quantification of ATP levels in RPE1 sgDIESL cells cultured for 24 h under the indicated conditions; 50 μM oleic acid, 10 μM (each) DGATi, 20 μM etomoxir. Bars represent mean ± SEM of $n$ = 4 independent experiments (one-way ANOVA, Bonferroni correction; ns, not significant). **g**, Brightfield images of 293T cells (left), and quantification of cell number of 293T cells (centre) and U251 cells (right) cultured under the indicated conditions for 24 h. Bars represent mean ± SEM of $n$ = 3 independent experiments (two-way ANOVA, Bonferroni correction; ns, not significant). Scale bar, 50 microns.

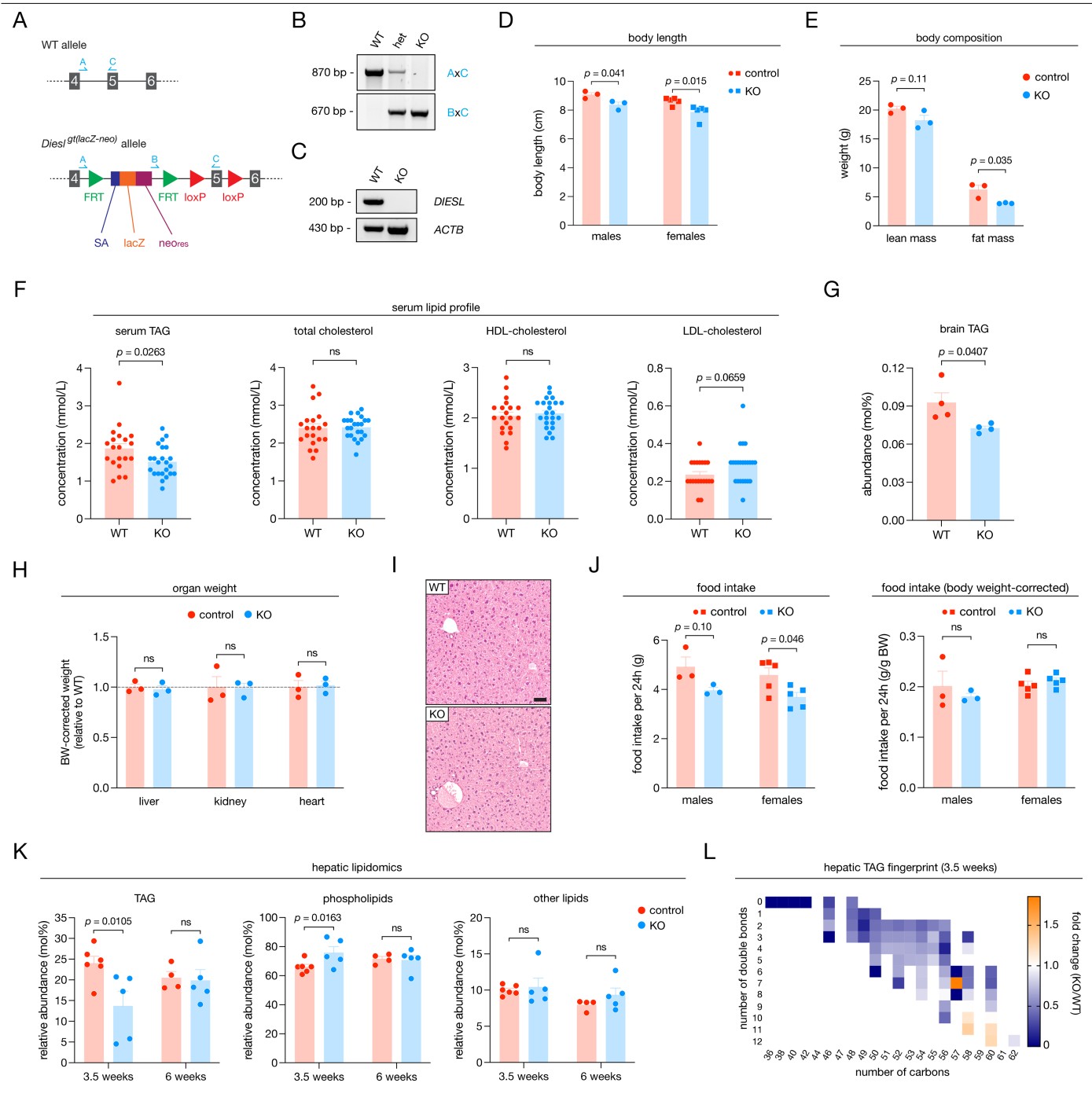

**Extended Data Fig. 9** | See next page for caption.

**Extended Data Fig. 9 | Characterization of *Diesl* KO mice. a**, Depiction of the loss-of-function allele harbored by *Diesl* KO mice. The disruptive gene-trap (SA, splice acceptor), lacZ and neomycin resistance cassette (neo$_{res}$) are integrated between exons 4 and 5 of the mouse *Diesl* gene. A, B and C represent primers used to map the integration of the cassette. **b**, PCR amplification of the *Diesl*/*Tmem68* genetic locus and integration of the gene-trap. PCR products were separated by agarose gel electrophoresis. Primer pairs indicated in the figure correspond to those denoted in **a**. **c**, *DIESL* mRNA was detected by RT-PCR using cDNA prepared from mouse liver, with primers spanning the exon junction of exons 4 and 5. *ACTB* mRNA was used as a loading control. PCR products were separated by agarose gel electrophoresis. **d**, Body length of adult male (circles) and female (squares) WT and *Diesl* KO mice. Bars represent mean ± SEM of *n* = 3 to 5 mice examined in a single experiment (two-tailed Student's *t* test). **e**, Body composition of adult WT and *Diesl* KO male mice as determined by MRI. Bars represent mean ± SEM of *n* = 3 male mice (two-way ANOVA, Bonferroni correction). **f**, Serum lipid profile of adult male mice of the indicated genotype, as determined by assay. Bars represent mean ± SEM of *n* = 20 to 24 mice examined in a single experiment (two-tailed Student's *t* test; ns, not significant). **g**, Whole-brain TAG content of 6-week-old male mice of the indicated genotype as measured by mass spectrometry. Bars represent mean ± SEM of *n* = 4 mice (two-tailed Student's *t* test). **h**, Body weight- (BW-) corrected weights of organs harvested from adult WT and *Diesl* KO male mice. Bars represent mean ± SEM of *n* = 3 mice (two-way ANOVA, Bonferroni correction; ns, not significant). **i**, Representative images of liver sections, collected from adult WT and *Diesl* KO male mice, stained with H&E. Scale bar, 50 microns. **j**, Food intake of adult male (circles) and female (squares) WT and *Diesl* KO mice, measured over a 24-hour period (left) and then corrected for body weight (BW, right). Bars represent mean ± SEM of *n* = 3 to 5 mice examined in a single experiment (two-way ANOVA, Bonferroni correction; ns, not significant). **k**, Categorized hepatic lipid levels in male mice (fed *ad libitum*), measured at 3.5 and 6 weeks using shotgun lipidomics, expressed as a percentage of total lipids measured in the sample. Bars represent mean ± SEM of *n* = 4 to 6 livers per condition, examined in a single experiment (two-way ANOVA performed on the entire dataset, Bonferroni correction; ns, not significant). **l**, TAG fingerprint in the liver of male mice (fed *ad libitum*) measured at 3.5 weeks of age, expressed as the fold-change of the average of the *Diesl* KO (*n* = 5 mice) over the average of WT mice (*n* = 6 mice).

# Reporting Summary

## Statistics

For all statistical analyses, confirm that the following items are present in the figure legend, table legend, main text, or Methods section.

| n/a | Confirmed | |
|---|---|---|
| ☐ | ☒ | The exact sample size (*n*) for each experimental group/condition, given as a discrete number and unit of measurement |
| ☐ | ☒ | A statement on whether measurements were taken from distinct samples or whether the same sample was measured repeatedly |
| ☐ | ☒ | The statistical test(s) used AND whether they are one- or two-sided<br>*Only common tests should be described solely by name; describe more complex techniques in the Methods section.* |
| ☐ | ☒ | A description of all covariates tested |
| ☐ | ☒ | A description of any assumptions or corrections, such as tests of normality and adjustment for multiple comparisons |
| ☐ | ☒ | A full description of the statistical parameters including central tendency (e.g. means) or other basic estimates (e.g. regression coefficient) AND variation (e.g. standard deviation) or associated estimates of uncertainty (e.g. confidence intervals) |
| ☐ | ☒ | For null hypothesis testing, the test statistic (e.g. *F*, *t*, *r*) with confidence intervals, effect sizes, degrees of freedom and *P* value noted<br>*Give P values as exact values whenever suitable.* |
| ☒ | ☐ | For Bayesian analysis, information on the choice of priors and Markov chain Monte Carlo settings |
| ☒ | ☐ | For hierarchical and complex designs, identification of the appropriate level for tests and full reporting of outcomes |
| ☒ | ☐ | Estimates of effect sizes (e.g. Cohen's *d*, Pearson's *r*), indicating how they were calculated |

*Our web collection on statistics for biologists contains articles on many of the points above.*

## Software and code

Policy information about availability of computer code

| | |
|---|---|
| Data collection | FACSDiva v.8.0.2 (BD Life Sciences) - flow cytometry<br>ProSort v.1.6 (BioRad) - cell sorting<br>LAS AF v.2.7.4 (Leica) - confocal microscopy<br>Typhoon FLA 9500 software v.1.1.0.187 (GE) - phosphorimaging |
| Data analysis | Analysis pipelines for haploid screens are available at https://github.com/BrummelkampResearch<br>ImageJ v.1.0 (NIH) - image quantification<br>FlowJo v.10.6.2 (BD Life Sciences) - flow cytometry analysis<br>PyMol v.2.5.4 (Schrodinger) - handling/representation of protein structures and models<br>Prism v.9.5.1 (GraphPad) - statistical analysis and plot construction<br>Photoshop v.24.4.1 (Adobe) - handling of microscopy images<br>RStudio v.2023.03.0+386 (Posit Software) - plot construction |

For manuscripts utilizing custom algorithms or software that are central to the research but not yet described in published literature, software must be made available to editors and reviewers. We strongly encourage code deposition in a community repository (e.g. GitHub). See the Nature Portfolio guidelines for submitting code & software for further information.

## Data

Policy information about availability of data

All manuscripts must include a data availability statement. This statement should provide the following information, where applicable:
- Accession codes, unique identifiers, or web links for publicly available datasets
- A description of any restrictions on data availability
- For clinical datasets or third party data, please ensure that the statement adheres to our policy

Sequencing data and screening data will be available, respectively, at the NCBI Sequence Read Archive (www.ncbi.nlm.nih.gov.sra; under the accession numbers SAMN35570720, SAMN35570721, and SAMN35570722) and an interactive screening database (https://phenosaurus.nki.nl/) upon publication.

## Human research participants

Policy information about studies involving human research participants and Sex and Gender in Research.

| | |
|---|---|
| Reporting on sex and gender | n/a |
| Population characteristics | n/a |
| Recruitment | n/a |
| Ethics oversight | n/a |

Note that full information on the approval of the study protocol must also be provided in the manuscript.

# Field-specific reporting

Please select the one below that is the best fit for your research. If you are not sure, read the appropriate sections before making your selection.

☒ Life sciences ☐ Behavioural & social sciences ☐ Ecological, evolutionary & environmental sciences

For a reference copy of the document with all sections, see nature.com/documents/nr-reporting-summary-flat.pdf

# Life sciences study design

All studies must disclose on these points even when the disclosure is negative.

| | |
|---|---|
| Sample size | Sample sizes were determined based on previous experience and norms in the field; no sample size calculations were performed. In instances requiring statistics, the sample size is described in the figure legends, and generally is three or more replicates. |
| Data exclusions | No data was excluded from analysis, except for sequencing reads that, when aligned to the human genome using Bowtie, carried more than one mismatch. |
| Replication | The number of replicates is indicated in the figure legends. All attempts at replication were successful. |
| Randomization | No randomization was performed, as phenotypes were assayed based on genotype and not treatment. |
| Blinding | In experiments where subjectivity could be introduced (i.e. the weighing of mice, quantification of microscopy images), the experimenter was blinded to the condition under which the experiment was completed. |

# Reporting for specific materials, systems and methods

We require information from authors about some types of materials, experimental systems and methods used in many studies. Here, indicate whether each material, system or method listed is relevant to your study. If you are not sure if a list item applies to your research, read the appropriate section before selecting a response.

## Materials & experimental systems

| n/a | Involved in the study |
|-----|----------------------|
| ☐ | ☒ Antibodies |
| ☐ | ☒ Eukaryotic cell lines |
| ☒ | ☐ Palaeontology and archaeology |
| ☐ | ☒ Animals and other organisms |
| ☒ | ☐ Clinical data |
| ☒ | ☐ Dual use research of concern |

## Methods

| n/a | Involved in the study |
|-----|----------------------|
| ☒ | ☐ ChIP-seq |
| ☐ | ☒ Flow cytometry |
| ☒ | ☐ MRI-based neuroimaging |

# Antibodies

**Antibodies used**

anti-ACTB (Abcam, ab6276) WB 1:10000
anti-alpha-tubulin (Santa Cruz Biotechnology, sc-32293) WB 1:1000
anti-AMPK (Cell Signaling Technology, 2532) WB 1:1000
anti-AMPK pT172 (Cell Signaling Technology, 2535) WB 1:1000
anti-CANX (Abcam, ab22595) IF 1:100, WB 1:1000
anti-CLTC (Thermo Fisher, PA5-17347) WB 1:1000
anti-EIF4G (Cell Signaling Technology, 2498) WB 1:1000
anti-FASN (Santa Cruz Biotechnology, sc-55580) WB 1:1000
anti-HA (Biolegend, 901503) IF 1:200, WB 1:1000
anti-HSPA5 (Cell Signaling Technology, 3177) WB 1:1000
anti-LAMP1 (Santa Cruz Biotechnology, sc-19992) WB 1:1000
anti-LC3B (Cell Signaling Technology, 2775) WB 1:1000
anti-LDHA (Cell Signaling Technology, 3582) WB 1:5000
anti-PDI (Abcam, ab2792) IF 1:500, WB 1:20000
anti-S6 (Cell Signaling Technology, 2317) WB 1:1000
anti-S6 pS235/pS236 (Cell Signaling Technology, 4856) WB 1:1000
anti-TMX1 (Atlas Antibodies, HPA003085) IF 1:100, WB 1:1000
anti-TMX1 (Origene, TA507042) WB 1:1000
anti-TOMM20 (Abcam, ab186735) WB 1:10000
anti-V5 (ThermoFisher, 14-6796-82) IF 1:500, WB 1:1000

**Validation**

anti-ACTB (Abcam, ab6276) detects ACTB by WB in human cells, as shown by the manufacturer using HAP1 ACTB KO cells (https://www.abcam.com/products/primary-antibodies/beta-actin-antibody-ac-15-ab6276.html)

anti-alpha-tubulin (Santa Cruz Biotechnology, sc-32293) was validated by WB using purified human protein in PMID 29146869

anti-AMPK (Cell Signaling Technology, 2532) was validated by WB in mouse cells in PMID 33596428 using AMPKa KO MEFs

anti-AMPK pT172 (Cell Signaling Technology, 2535) was validated by WB in mouse cells in PMID 33596428 using AMPKa KO MEFs

anti-CANX (Abcam, ab22595) was validated by IF as an ER marker in human cells and by WB as an ER stress-responsive protein in human cells (this paper)

anti-CLTC (Thermo Fisher, PA5-17347) was validated as a cytosol/membrane marker in human cells (this paper)

anti-EIF4G (Cell Signaling Technology, 2498) detects a band of the correct size in human cells, as shown by the manufacturer (https://www.cellsignal.com/products/primary-antibodies/eif4g-antibody/2498)

anti-FASN (Santa Cruz Biotechnology, sc-55580) detects FASN by WB in human cells, as shown in PMID 32111832 using HAP1 FASN KO cells

anti-HA (Biolegend, 901503) was validated for WB and IF in human cells by the manufacturer (https://www.biolegend.com/en-us/products/purified-anti-ha-11-epitope-tag-antibody-11374)

anti-HSPA5 (Cell Signaling Technology, 3177) was validated by WB as an ER stress-responsive protein in human cells (this paper)

anti-LAMP1 (Santa Cruz Biotechnology, sc-19992) detects LAMP1 by WB in human cells, as shown in PMID 24970085 using HAP1 LAMP1 KO cells

anti-LC3B (Cell Signaling Technology, 2775) detects LC3B by WB in human cells, as shown in PMID 31208283

anti-LDHA (Cell Signaling Technology, 3582) detects a band of the correct size in human cells, as shown by the manufacturer (https://www.cellsignal.com/products/primary-antibodies/ldha-c4b5-rabbit-mab/3582)

anti-PDI (Abcam, ab2792) detects PDI (P4HB) by IF in human cells (https://www.abcam.com/products/primary-antibodies/p4hb-antibody-rl90-ab2792.html)

anti-S6 (Cell Signaling Technology, 2317) detects a band of the correct size in human cells, as shown by the manufacturer (https://www.cellsignal.com/products/primary-antibodies/s6-ribosomal-protein-54d2-mouse-mab/2317)

anti-S6 pS235/pS236 (Cell Signaling Technology, 4856) was validated by WB by the manufacturer in stimulated human cells (https://www.cellsignal.com/products/primary-antibodies/phospho-s6-ribosomal-protein-ser235-236-2f9-rabbit-mab/4856)

anti-TMX1 (Atlas Antibodies, HPA003085) was validated by WB and IF in human cells using HAP1 TMX1 KO cells (this paper)

anti-TMX1 (Origene, TA507042) was validated by WB in human cells using HAP1 TMX1 KO cells (this paper)

anti-TOMM20 (Abcam, ab186735) detects a band of the correct size in human cells, as shown by the manufacturer (https://www.abcam.com/products/primary-antibodies/tomm20-antibody-epr15581-54-mitochondrial-marker-ab186735.html)

anti-V5 (ThermoFisher, 14-6796-82) was validated for WB and IF in human cells by the manufacturer (https://www.thermofisher.com/antibody/product/V5-Tag-Antibody-clone-TCM5-Monoclonal/14-6796-82)

## Eukaryotic cell lines

Policy information about cell lines and Sex and Gender in Research

| | |
|---|---|
| Cell line source(s) | HAP1 cells were subcloned and isolated in house; they are available from Horizon Discovery. HEK293T, A549, HeLa, HT-29, RPE-1 and U2OS and U251 cells were purchased from ATCC. |
| Authentication | The ploidy of HAP1 cells was routinely checked by DAPI content. No other cell lines were authenticated |
| Mycoplasma contamination | Cell lines were routinely tested for mycoplasma and the lines/clones used in this study were consistently negative. |
| Commonly misidentified lines (See ICLAC register) | HEK293T |

## Animals and other research organisms

Policy information about studies involving animals; ARRIVE guidelines recommended for reporting animal research, and Sex and Gender in Research

| | |
|---|---|
| Laboratory animals | This study used C57/BL6J and C57/BL6N mice, aged 2 to 28 weeks. Mice were maintained in a certified animal facility at 21°C and 55% humidity, in 12h light/dark cycles. |
| Wild animals | This study did not use wild animals. |
| Reporting on sex | Body weight measurements were carried out in both male and female mice, as this measurement is non-intrusive. Other measurements (TAG content of serum, lipidomics, immunoblots and morphological analysis) were performed in male mice as female mice were reserved for colony maintenance. |
| Field-collected samples | This study did not involve samples collected from the field. |
| Ethics oversight | National Ethics Committee for Animal Experiments of the Netherlands |

Note that full information on the approval of the study protocol must also be provided in the manuscript.

## Flow Cytometry

### Plots

Confirm that:

☒ The axis labels state the marker and fluorochrome used (e.g. CD4-FITC).

☒ The axis scales are clearly visible. Include numbers along axes only for bottom left plot of group (a 'group' is an analysis of identical markers).

☒ All plots are contour plots with outliers or pseudocolor plots.

☒ A numerical value for number of cells or percentage (with statistics) is provided.

### Methodology

| | |
|---|---|
| Sample preparation | Haploid screens:<br><br>2-3 x 10^9 gene-trapped HAP1 cells of the indicated genotype were harvested by trypsinization and fixed in Fix Buffer I (BD Biosciences) for 10 minutes at 37°C. For the oleic acid-loaded screen, cells were first cultured for 24h in complete medium supplemented with 200 μM oleic acid, and then chased in medium lacking oleic acid for another 24h prior to harvesting. Cells were treated with 1 mg/ml RNase A (Qiagen) diluted in FACS buffer (10% FBS in PBS) at 37°C for 30 minutes prior to staining with 1 μg/ml BODIPY 493/503 and 10 μg/ml propidium iodide (Sigma-Aldrich), diluted in FACS buffer, for one hour at room temperature. Cells were washed twice in FACS buffer before being passed through a 40 μm cell strainer. |

BODIPY 493/503 measurements in fixed HAP1 cells:

HAP1 cells, grown in 10cm plates, were collected by trypsinization and were fixed in Fix Buffer I (BD Biosciences) for 10 minutes at 37°C. Cells were pelleted, washed with FACS buffer (10% FBS in PBS), resuspended in FACS buffer and counted. 10 million cells were stained with 1 µg/ml BODIPY 493/503 and 5 µg/ml DAPI (Invitrogen), diluted in FACS buffer, for one hour at room temperature. Cells were washed once in FACS buffer, then passed through a 35 µm nylon mesh cell strainer into a FACS tube.

Mitochondrial measurements in live RPE1 cells:

RPE1 cells were treated as described and pulsed for 30 minutes with either 600 nM TMRM or 250 nM MitoTracker Red CM-H2XROS in medium depleted of lipoproteins. In TMRM experiments, cells were then incubated in 150 mM for an additional 30 minutes, using 20 µM CCCP as a positive control. Cells were collected by trypsinization and stored on ice in FACS buffer.

| Instrument | Cell sorting for screens was performed using an S3 Sorter (Bio-Rad) and analytical flow cytometry was performed on an LSR Fortessa (BD Biosciences). |
|---|---|
| Software | Haploid screens: sorting was performed using ProSort (Bio-Rad)<br>Analytical flow cytometry: flow cytometry was performed using FACSDiva (BD) and data was analyzed using FlowJo (BD) |
| Cell population abundance | For haploid screens, 10^7 cells were collected for both the lowest and highest 5% of BODIPY signal from haploid cells in G1. Prior to genomic DNA isolation, several thousand cells from each sorted population were re-analyzed to ensure purity of the sorted cells. |
| Gating strategy | FSC-A and SSC-A were gated to exclude cell debris. For haploid screens, PI (area) channel was gated for single haploid cells in G1 (DNA content = 1n). From this population, BODIPY 493/503 (area) was assessed using a 488 nm laser.<br><br>Intact RPE1 cells were identified using FSC-A/SSC-A as above, and then singlets were identified by SSC-A/SSC-W. From this population, mean fluorescence was assessed. |

☒ Tick this box to confirm that a figure exemplifying the gating strategy is provided in the Supplementary Information.

