## [Peer Review File · Nature]

Manuscript Title: Identification of an alternative triglyceride biosynthesis pathway

Reviewer Comments & Author Rebuttals

Reviewer Reports on the Initial Version:

Referees' comments:

Referee #1 (Remarks to the Author):

This manuscript reports identification of a novel pathway of triglyceride synthesis, involving DIESL, an acyltransferase of previously unknown function, and a negative regulator of DIESL, TMX1. The authors suggest that DIESL acts via acylation of DAG using a phospholipid or a phospholipid precursor as an acyl donor, and they propose that this alternate system is important to maintain energy homeostasis during periods of limited extracellular lipid availability, serving to provide membrane acyl chains for lipid catabolism by mitochondria. The findings are novel and intriguing but there are many unanswered questions.

Major concerns:

1. Changes in lipids with genetic manipulation of DIESL/TMX1 in mammalian and bacterial cells could be indirect, and the study fails to establish clear endogenous product to precursor relationships. Can the activity of DIESL/TMX1 pathway be reconstituted in vitro using stable isotopes to trace the progression from precursor to product? This is a particular concern given the co-incident alterations in cholesteryl esters in mammalian cells, which raise the possibility that with only a very brief period of serum starvation, the changes relate to differences in lipoprotein catabolism.
2. The authors cite references 17 and 18 in support of the existence of an alternate triglyceride synthesis pathway. If DIESL/TMX1 is responsible for the previous observations, then the pathway should be active in murine macrophages and hepatocytes, but not fibroblasts.
3. The authors speculate that the function of this pathway is to maintain mitochondrial energetics, presumably through oxidation of fatty acids. It is unclear why those lipids would first be channeled into triacylglycerol pools instead of directly to the mitochondria for beta oxidation. Regardless, the work would be strengthened by a demonstration of the physiological role of this new pathway.

Minor concern:

1. Better description is needed for mutational index.

Referee #2 (Remarks to the Author):

McLelland and colleagues report the identification of a triglyceride synthetic pathway independent of DGAT activity. A genetic screen was performed in haploid cells validated as deficient in DGAT1 and 2. TMX1 mutations increased lipid droplets that were large and consisted of aggregated smaller structures. There are other TMX family members, but only TMX1 could suppress droplet formation

in TMX1 KO cells.

Another genetic screen compared oleic acid loaded WT cells and TMX1 KO cells using lipid droplets as a readout. DGAT1 was identified in both cell types, and TMEM68 was identified in the TMX1 KO cells. TMEM68 was renamed DIESL. DIESL KO cells showed lipid droplets with oleic acid loading. TMX1 KO caused TAG accumulation that was lost when these cells were also subjected to DIESL KO.

Data are presented supporting the presence of a glycosylated DIESL facing the ER lumen. Crosslinking and co-IP studies were consistent with a DIESL-TMX1 interaction. Homology modeling identified a putative active site in DIESL and mutation of this site (H130A) prevented accumulation of lipid droplets.

HAP1 cells deficient in both DGATs and DIESL were generated. Active and inactive DIESL constructs were expressed in these cells and lipid profiling was performed. Two lipid species increased in the absence of DIESL, DAG and phosphatidylcholine. TAG production was lost in the absence of DIESL. Human DIESL was expressed in E Coli cultured in the absence of lipids and TAGs were synthesized.

When U251 cells were cultured under lipid deficiency, DIESL prevented mitochondrial disruption and AMPK activation. The authors conclude that DIESL synthesizes TAGs in part by using membrane phospholipids and this process preserves mitochondrial function under conditions of lipid deficiency.

1. This manuscript has value. The genetic screens and cell biology experiments are rigorous, and the finding that DIESL/TMX1 mediates TAG synthesis independent of DGATs is provocative.

2. However, these findings were generated in cultured cells, and not merely cultured cells but cancer/immortalized cells. HAP1 cells, in addition to being mostly haploid, were derived from a chronic myelogenous leukemia cell line. 293 T cells are human cells immortalized by an oncogene. U251 cells are derived from a malignant glioblastoma. All of these cell types have adapted to growth in the setting of constant nutrition (mostly glucose that perhaps fuels aerobic glycolysis) that make them strange models to pursue a pathway that may be important with lipid deprivation. Therefore, the burden is on the authors to show that this interesting pathway is relevant to normal physiology. At the least, data should be presented to demonstrate the presence of this pathway in cultured cells that model lipid handling and are not malignant. Is this pathway present in hepatocytes? Endogenous lipid synthesis is extremely important for neurons and glial cells. Is this pathway present in these cell types?

3. The notion that DIESL may maintain “energy homeostasis” is fascinating, but very little information is provided about this effect. For the experiments of Figure 4, DIESL knockdown activates AMPK and alters mitochondrial morphology. Is energy homeostasis disrupted? Are high energy phosphates low? Is respiration altered?

4. Does lipid starvation in the U251 cells activate autophagy pathways?

5. Some very nice experiments demonstrate the DIESL is an ER membrane glycoprotein, and the

glycosylation status of this protein is characterized. However, recombinant DIESEL was expressed in E coli (using a bacterial membrane targeting sequence) to demonstrate its TAG synthesis capacity. Since these E coli have presumably not been manipulated to be able to carry out glycosylation, what do these results say about the glycosylation of DIESEL in human cells?

6. The argument in lines 203 to 206 of the discussion about linkage between mouse liver DIESEL and the NNT gene is not strong. Variations at NNT were proposed to explain differences in metabolism observed between C57BL/6J mice and other strains. However, this does not appear to be true for insulin secretion, a major mediator of lipid metabolism (see Wong et al. *Endocrinology* 2010; 151:96-102). There is a Blog Post on the Jax website that summarizes the effects of NNT on the response to diet-induced obesity in mice, and the consensus is that variations at this locus are not required for a robust response to high fat feeding. So the authors are left with limited evidence for the physiologic relevance of DIESEL, even at the cellular level. A quick search of public databases suggests linkage between TMEM68 and height. Could this suggest a need for this pathway in normal growth?

Referee #3 (Remarks to the Author):

In this manuscript, the Brummelkamp group has utilized their elegant CRISPR screening strategy in haploid cells to pin down an alternative mechanism for triacylglycerol (TAG) biosynthesis in human cells. They report on the identification of TMEM68, an acyltransferase of previously unknown function, that they rename DIESEL as a TAG synthase, and TMX1, a PDI family oxidoreductase, as an interactor and negative regulator of DIESEL.

Overall, this study identifies a potentially critical new player in the control of neutral lipid metabolism and several of the findings made appear relatively robust. TMX1 was identified first in their study and considerable effort was devoted for instance to exclude a role for other TMX family members (TMX2-4). Instead, the characterization of DIESEL, its mechanism of action, regulation, and physiological significance remain less compelling, even though it represents the business end of the pathway. Moreover, considering that TMX1 and TMEM68/DIESEL are previously at least partially characterized proteins, the findings raise questions related to their earlier reported properties. Please see specific comments below.

Major issues:

1. Membrane topology and activity of DIESEL: The N-terminus is expected to be ER-luminal, if the N-glycosylation motif at N5-X-S/T is used, as reported in Extended Data Fig. 5. However, if expression of human DIESEL in E.coli yields a functional TAG synthase, this implies that this glycosylation may not be needed for human DIESEL to be active. This would be important to clarify. How is DIESEL anchored to the membrane and how is the putative active site expected to be positioned relative to the bilayer and substrate?

The interpretation of the BFA experiment (Extended Data Fig. 5e) does not seem to make sense. The authors claim that since BFA treatment results in an increased apparent MW of 3xHA-DIESEL suggestive of elongated glycan chains, this indicates that DIESEL faces the ER rather than the Golgi

lumen. Since BFA leads to redistribution of cis- and medial-Golgi components to the ER, this does not differentiate between ER and Golgi (cis-medial) luminal compartments.

2. DAG acylation by DIESL and E.coli data: The acylation of DAG by DIESL is demonstrated by using a fluorescent non-natural DAG analog (NBD-DAG) only (Fig. 4c, Extended Data Fig. 9b-c). These data should be strengthened by using more natural DAG as substrate.

The DIESL constructs showing activity in E.coli (Fig. 4d, Extended Data Fig. 9e,f) are not clear: does the triple HA-tag inhibit TAG generation in E. coli, since it is not shown in panel 9f?

The E.coli phospholipid data (Extended Data Fig. 9g) is too crude to be informative: based on this one cannot make the conclusion that DIESL-synthesized TAGs reflect the cellular lipidome in length, saturation and complexity, as claimed.

3. TMX1 inhibition of DIESL: TMX1 active site cysteines are needed for its disulfide-reducing activity. Is TMX1 activity needed for inhibiting DIESL and/or for stabilizing DIESL protein levels? How could the TMX1 inhibition of DIESL fit together with the finding that genetic disruption of TMX1 destabilizes the DIESL protein? TMX1 has been localized to mitochondria-associated ER membranes (MAMs). Is DIESL also localized to MAMs?

4. Effects on mitochondrial function and energy homeostasis: This part of the manuscript related to the cell physiological relevance of the findings appears premature. Mitochondrial morphological changes can be induced by multiple indirect cues and are therefore rather uninformative. Also, it seems that the mitochondrial morphological groups (fused, intermediated, aggregated; Fig. 4f; Extended Data Fig. 10b) were determined by subjective assessment. AMPK is activated under lipid starvation if DIESL is not functional (Fig. 4g). What does this mean, i.e. how is energy homeostasis supported by DIESL activity? Are cellular ATP levels decreased if DIESL is not functional under lipid starvation? AMPK can promote mitochondrial biogenesis but how is this related to the aggregated mitochondrial phenotype?

Additional comments:

5. The authors suggest that DIESL-dependent TAG synthesis somehow helps to maintain energy homeostasis in limited extracellular lipid availability. The exact delipidation/lipid starvation conditions used in each experiment should be clearly spelled out. Are other than lipid starvation cues relevant, does for instance amino acid starvation induce DIESL dependent TAG synthesis?

6. In the Discussion, the “two levels” at which DIESL-dependent TAG synthesis is linked to mitochondria/redox biology, is not clear; what are these levels?

7. Quantifications of individual lipid droplet sizes and numbers at the described light microscopic resolution are not accurate (Extended Data Fig. 2a). The area occupied by LDs per cell (middle panel) should suffice although it is not clear what the n's here indicate.

8. Strictly speaking, Extended Data Fig. 10a does not show DIESL inhibition (rather inhibition of DGATs in combination with DIESL CRISPR KO). The title of this figure should also be reformulated.

Author Rebuttals to Initial Comments:

Referee reports / Author response

Dear Dr Brummelkamp

Your manuscript entitled "Identification of an alternative triglyceride biosynthesis pathway" has now been seen by 3 referees, whose comments are attached below. While they find your work of potential interest, as do we, they have raised important concerns that in our view need to be addressed before we can consider publication in Nature.

The referees have expressed great interest in your findings. However, they raise a host of concerns, ranging from lack of sufficient biological contextualization of the role of this enzyme/pathway in cell biology *in vitro* and in cells/tissue, to technical concerns with the robustness of the biochemical data. We would be reluctant to consider a revised manuscript that does not fully address all of the points raised by the reviewers. Should further experimental data allow you to address these criticisms, we would be happy to consider a revised manuscript (unless something similar has been accepted at Nature or appeared elsewhere in the meantime).

We would like to thank the reviewers for their valuable feedback on our study. We used genetics to identify a new TAG-synthesizing complex in the ER, and were pleased that our work was received favourably. We found the questions raised by the referees very helpful, and we believe that in addressing these concerns we have greatly strengthened our study. Of the referees' comments, we do recognize that the request to provide a better insight into the (physiological) function of this pathway was of paramount importance to address.

This revised manuscript contains extensive new experimental data primarily aimed at understanding the function of DIESL in cells and its relevance *in vivo*, as well as further characterization of the DIESL enzyme. We provide here a brief summary of major new findings:

- 1) To demonstrate that DIESL is important in an organism, **we generate mice lacking *Diesl* and provide a first characterization of these animals**. Several tissues in these mice contain perturbed TAG levels, and these animals have a ~20% decrease in body mass. We additionally show that lack of *Diesl* triggers a stress response and leads to deregulated DAG levels in the liver.
- 2) We have extended our work on mitochondrial function, and now show that **DIESL localizes to mitochondria-ER membrane contact sites and its loss causes mitochondrial dysfunction**, as measured by both direct and indirect metrics; increased ROS and AMPK activation, as well as decreased membrane potential and ATP levels. This perturbed state is exacerbated when cells

are lipid-deprived, and this impacts cell fitness. We show that these phenotypes can be rescued by free fatty acids in a manner sensitive to mitochondrial lipid uptake. We propose a model in which DIESL maintains mitochondrial function by facilitating the uptake of fatty acids into mitochondria through its ability to synthesize TAG at mito-ER contacts.

- 3) **We reconstitute the activity of DIESL in a cell-free assay using radioactive DAG** (i.e. a natural substrate), demonstrating indeed that DIESL acylates DAG to form TAG. In a similar experiment to what was done during the characterization of DGAT1 (Cases et al. PNAS 1998), we reconstituted cell lysates lacking TAG synthesis capacity with DIESL and demonstrate DIESL-dependent TAG formation *in vitro*.
- 4) In greater detail, **we provide experiments examining DIESL in the ER**. We now provide more information on DIESL topology (including the DIESL AlphaFold model for reference), the role of its N-terminus on stability and/or import of the protein, and an expansion of our initial experiments related to glycosylation.

We feel that these new experiments provide a first insight into the physiological relevance and function of DIESL, and that the additional experiments have further strengthened the crux of our findings – that we have identified a new TAG-synthesizing unit in the ER. We have provided a detailed response to each point raised by the reviewers below.

Finally, we acknowledge that our understanding of DIESL function remains incomplete. The functions of enzymes with a similar enzymatic activity, such as DGAT enzymes, have been described over the years; the absorption of fats by the intestine, the synthesis and secretion of lipoproteins, mitochondrial metabolism, the detoxification of DAG accumulation, the protection of cells from ferroptosis, etc. In this study, we report the very first identification of the TMX1-DIESL TAG synthesis complex and show that it plays a role in normal physiology. We thus anticipate that future work on the regulation of DIESL by TMX1, the role of TMX1 *in vivo* and the effect of diet manipulation (caloric restriction and high-fat diets) on organisms lacking TMX1 and DIESL will be of high interest to the research communities of both metabolism and cell biology.

Referee expertise:

Referee #1: lipid biology/genetic screen

Referee #2: lipid metabolism

Referee #3: lipid metabolism/lipid droplet biology

Referees' comments:

Referee #1 (Remarks to the Author):

This manuscript reports identification of a novel pathway of triglyceride synthesis, involving DIESL, an acyltransferase of previously unknown function, and a negative regulator of DIESL, TMX1. The authors suggest that DIESL acts via acylation of DAG using a phospholipid or a phospholipid precursor as an acyl donor, and they propose that this alternate system is important to maintain energy homeostasis during periods of limited extracellular lipid availability, serving to provide membrane acyl chains for lipid catabolism by mitochondria. The findings are novel and intriguing but there are many unanswered questions.

Major concerns:

1. Changes in lipids with genetic manipulation of DIESL/TMX1 in mammalian and bacterial cells could be indirect, and the study fails to establish clear endogenous product to precursor relationships. Can the activity of DIESL/TMX1 pathway be reconstituted in vitro using stable isotopes to trace the progression from precursor to product?

This is a particular concern given the co-incident alterations in cholesteryl esters in mammalian cells, which raise the possibility that with only a very brief period of serum starvation, the changes relate to differences in lipoprotein catabolism.

We thank the reviewer for raising this crucial point. To address this, we reconstituted the DAG acyltransferase activity of DIESL in a cell-free assay, using radiolabeled DAG ([¹⁴C]DAG) as the DIESL substrate. We used reconstituted cell lysates as the purification of soluble, active DIESL proved to be highly challenging, likely due to its transmembrane structure. We thus harvested lysates from HAP1 4KO cells (lacking *DGAT1*, *DGAT2*, *DIESL* and *TMX1*) reconstituted with or without DIESL, incubated them with [¹⁴C]DAG, separated extracted lipids by TLC and measured radioactivity by phosphorimaging. This is very similar to what was done for the identification of DGAT1 (Cases *et al.* PNAS 1998), although in this latter case the human protein was expressed in *Drosophila* cells. **We show that the expression of DIESL stimulates the acylation of DAG to TAG in a manner that is sensitive to temperature and heat-inactivation** (new Fig. 4h). Although we measured robust TAG synthesis in extracts containing DIESL, we also did observe some background TAG synthesis in lysates lacking DIESL. This is not uncommon in this type of assay as, in the DGAT assay established by the Farese group, DGAT1 expression had a similar-fold stimulation of TAG synthesis in the initial experiment by Cases *et al.* (i.e. five-fold over background).

Here we provide the relevant figure, modified in red, from Cases *et al.*:

While we cannot explain the basal level of TAG synthesis in 4KO cell extracts incubated with a high concentration of DAG (which could be either physiological or artifactual, such as a TAG lipase functioning in reverse), we demonstrate that **expression of DIESL strongly stimulates the production of [¹⁴C]TAG from [¹⁴C]DAG**. This reconstituted assay, in combination with our data showing that DIESL confers *E.coli* with the ability to make TAG (which is normally undetectable) at the expense of DAG (new Fig. 4d-g), demonstrate that DIESL synthesizes TAG (rather than affects the turnover of TAG generated by other enzymes).

2. The authors cite references 17 and 18 in support of the existence of an alternate triglyceride synthesis pathway. If DIESL/TMX1 is responsible for the previous observations, then the pathway should be active in murine macrophages and hepatocytes, but not fibroblasts.

We thank the reviewer for bringing this to our attention. We found it difficult to assess if DIESL was responsible for those observations because addressing this thoroughly required the generation and analysis of embryos from double knockout mice (*Dgat* double knockout mice are embryonic lethal) vs *Dgat1/Dgat2/Diesl* triple knockout mice. We only used the aforementioned observations as a starting rationale in our introduction for examining TAG formation in DGAT-deficient cells. We cannot exclude that more DGAT- and DIESL-independent TAG synthesis pathways have yet to be discovered. Here, we would like to explicitly clarify that we are not implying to have resolved these previous observations. Accordingly, we have changed the text to refer to other organisms: *“Inspired by observations of alternative TAG synthesis mechanisms in algae, yeast and mice, we applied a haploid genetic approach to reveal an unexpected unit catalyzing alternative TAG synthesis in human cells”* (lines 54-56).

3. The authors speculate that the function of this pathway is to maintain mitochondrial energetics, presumably through oxidation of fatty acids. It is unclear why those lipids would first be channeled into triacylglycerol pools instead of directly to the mitochondria for beta oxidation. Regardless, the work would be strengthened by a demonstration of the physiological role of this new pathway.

We agree with the reviewer in that showing a physiological role for DIESL would strengthen the manuscript. To do this, **we have generated mice lacking *Diesl* and have provided an initial characterization of these animals**. These KO mice, while viable, have a marked body weight deficit. Both male and female KO mice are about ~20% lighter than WT and heterozygous animals (new Fig. 5a). As DIESL is an enzyme that synthesizes TAG, we measured this metabolite in these mice and detected TAG deficits (15-20%) in both the brain and serum (new Fig. 5b and Extended Data Fig. 7c). In the liver, we notably did not see a difference in stored TAG (new Fig. 5b); this was not so surprising, as DGAT2 has been shown to be the enzyme responsible for the bulk of stored hepatic TAG. However, we did observe a striking, 25% increase in the amount of the DIESL substrate DAG in this organ (new Fig. 5d) accompanied by a cellular stress response (new Fig. 5c) indicating a role for *Diesl* in this organ. Together, these *in vivo* data provide strong evidence that DIESL is important in normal physiology, as it regulates the homeostasis of lipids in several organs and affects body size.

Using perturbation experiments in cultured cells, we show that **DIESL is important for mitochondrial activity during periods of lipid starvation** (now expanded with more direct measurements for mitochondrial function, see new Fig. 5e-h and Extended Data Fig. 9a-f). Future studies applying perturbations on *Diesl* KO mice will thus be of great interest (i.e. crossing with DGAT knockouts, dietary changes, etc.).

In response to the question pertaining to why “lipids would first be channeled into triacylglycerol pools instead of directly to the mitochondria for beta oxidation”, we provide two explanations:

- (i) this process would provide optionality for the use of acyl chains for either storage or energy production, which would be of great importance when the free fatty acid pool is limited (i.e. starvation).
- (ii) this may better facilitate the correction of lipid imbalances within the ER that can lead to toxicity (i.e. toxic acyl chains within phospholipids, elevated DAG levels, etc.).

These different outcomes connect to the observed phenotypes in both cell culture and in mice (lipid droplet formation, ATP production and activation of CHOP) suggesting that DIESL could play more than one role.

Minor concern:

1. Better description is needed for mutational index.

We have included a short definition of the mutational index in the legend of Figure 1: “*The mutational index (MI) represents the ratio of inactivating gene-trap mutations per gene recovered*”

from each (HI and LO) population (see Methods for a complete description)", as well as the last sentence of the Mutagenesis Screening section of the Methods: "The mutational index (MI) represents the ratio of the occurrence of unique, disruptive (i.e. insertion [ins.] of gene-trap in the sense orientation) mutations in the body of a given gene (5'-UTR, exon and intron) in the HI compared to the LO population, normalized by the total other unique, disruptive mutations in each population". We additionally provide a calculation under this sentence in the same section.

Referee #2 (Remarks to the Author):

McLelland and colleagues report the identification of a triglyceride synthetic pathway independent of DGAT activity. A genetic screen was performed in haploid cells validated as deficient in DGAT1 and 2. TMX1 mutations increased lipid droplets that were large and consisted of aggregated smaller structures. There are other TMX family members, but only TMX1 could suppress droplet formation in TMX1 KO cells.

Another genetic screen compared oleic acid loaded WT cells and TMX1 KO cells using lipid droplets as a readout. DGAT1 was identified in both cell types, and TMEM68 was identified in the TMX1 KO cells. TMEM68 was renamed DIESL. DIESL KO cells showed lipid droplets with oleic acid loading. TMX1 KO caused TAG accumulation that was lost when these cells were also subjected to DIESL KO.

Data are presented supporting the presence of a glycosylated DIESL facing the ER lumen. Crosslinking and co-IP studies were consistent with a DIESL-TMX1 interaction. Homology modeling identified a putative active site in DIESL and mutation of this site (H130A) prevented accumulation of lipid droplets.

HAP1 cells deficient in both DGATs and DIESL were generated. Active and inactive DIESL constructs were expressed in these cells and lipid profiling was performed. Two lipid species increased in the absence of DIESL, DAG and phosphatidylcholine. TAG production was lost in the absence of DIESL. Human DIESL was expressed in E Coli cultured in the absence of lipids and TAGs were synthesized.

When U251 cells were cultured under lipid deficiency, DIESL prevented mitochondrial disruption and AMPK activation. The authors conclude that DIESL synthesizes TAGs in part by using membrane phospholipids and this process preserves mitochondrial function under conditions of lipid deficiency.

1. This manuscript has value. The genetic screens and cell biology experiments are rigorous, and the finding that DIESL/TMX1 mediates TAG synthesis independent of DGATs is provocative.

We thank the reviewer for their enthusiasm regarding our work on this new metabolic route.

2. However, these findings were generated in cultured cells, and not merely cultured cells but cancer/immortalized cells. HAP1 cells, in addition to being mostly haploid, were derived from a chronic myelogenous leukemia cell line. 293 T cells are human cells immortalized by an oncogene. U251 cells are derived from a malignant glioblastoma. All of these cell types have adapted to growth in the setting of constant nutrition (mostly glucose that perhaps fuels aerobic glycolysis) that make them strange models to pursue a pathway that may be important with lipid deprivation. Therefore, the burden is on the authors to show that this interesting pathway is relevant to normal physiology. At the least, data should be presented to demonstrate the presence of this pathway in cultured cells that model lipid handling and are not malignant. Is this pathway present in hepatocytes? Endogenous lipid synthesis is extremely important for neurons and glial cells. Is this pathway present in these cell types?

We appreciate the reviewer's suggestion that this manuscript would be strengthened with additional data that provide better insight into the role of this pathway in normal physiology. We chose to address this by generating *Diesl* knockout mice, and we report several measurements in new Fig. 5a-d as well as new Extended Data Fig. 7 pertaining to the reviewer's concerns:

1. *Diesl* KO mice have a 15-20% reduction in body mass (new Fig. 5a), demonstrating that DIESL plays a physiologically-relevant role *in vivo*.
2. These mice also have perturbations in TAG levels in particular organs, such as the circulatory system (Fig. 5b) and the brain (new Extended Data Fig. 7c; culturing neurons and glia on their own presents a technical challenge thus we measured whole-brain TAG by mass spectroscopy).
3. In the liver (a complex metabolic organ), we detect no perturbations in stored TAG levels (new Fig. 5b). However, we observe two phenotypes that indicate to us that DIESL plays a role in this tissue; i) we observe an increase in the stress-responsive transcription factor CHOP (Fig. 5c) and ii) we measure a ~25% increase in the abundance of the DIESL substrate DAG (Fig. 5d). We consider the lack of change in hepatic TAG rather unsurprising as DGAT2 is known to be by far the biggest contributor to steady-state levels of TAG in the liver (Cases *et al.* JBC 2001; Stone *et al.* JBC 2004). DIESL-derived TAGs may also be transient, in that they are rapidly metabolized by mitochondria in this tissue (as suggested by our cell culture experiments) or secreted (we do measure a decline in serum TAGs in the *Diesl* KO). Notably, the activation of CHOP mirrors a similar phenomenon observed in mouse adipose tissue lacking DGAT1 (Chitraju *et al.* JLR 2019) and is likely caused by lipid imbalance in the ER.

We additionally agree with the reviewer in that we looked at DIESL function primarily in transformed cell lines. To address this, now provide data showing that DIESL maintains mitochondrial function in untransformed RPE1 cells. These experiments are elaborated upon in point 3.

3. The notion that DIESL may maintain “energy homeostasis” is fascinating, but very little information is provided about this effect. For the experiments of Figure 4, DIESL knockdown activates AMPK and alters mitochondrial morphology. Is energy homeostasis disrupted? Are high energy phosphates low? Is respiration altered?

We agree that our initial characterization of the effect on DIESL loss on what we referred to broadly as “energy homeostasis” was perhaps preliminary. We now have included several new experiments (mostly in untransformed RPE1 cells) showing that DIESL **maintains mitochondrial function in a manner relating to fatty acid availability**. RPE1 cells lacking DIESL have mitochondrial dysfunction when starved of lipoproteins; their mitochondria are hypopolarized and energy phosphates are low, a situation that results in impaired cell fitness (new Fig. 5f-g). Importantly, these phenotypes can be rescued by the addition of fatty acids in a manner sensitive to inhibition of mitochondrial fatty acid uptake (new Extended Data Fig. 9f). These cells also have perturbed AMPK and mitochondrial ROS levels in the basal state (new Extended Data Fig. 9a and c). Indeed, the TMX1-DIESL complex is enriched in mitochondria-ER contact sites (new Fig. 5e, “MAM” fraction). Thus, RPE1 cells lacking DIESL have a mitochondrial phenotype that is worsened (to the point of affecting cell fitness) when RPE1 cells are starved of lipids.

We have measured AMPK across several cell lines (RPE1, U251, HT29 and 293T) and would like to note that, while the AMPK response varies between lines, the direction is consistent. **In all cell lines, loss of DIESL sensitizes cells to activation of AMPK**. However, we have found that the threshold needed to activate this kinase is different between these cell lines; AMPK is already active in RPE1 DIESL KOs at the steady-state, while in U251 and HT29 this requires lipoprotein starvation as a stimulus, whereas 293Ts need to be lipoprotein-starved and treated with DGAT inhibitors. These different thresholds for activation could be derived from differences in extracellular lipid uptake, mitochondrial activity, lipid storage or other expected variations between these normal and cancer cells.

We have not, so far, detected strong variations in AMPK in unperturbed mice. In our hands, the strongest activator of AMPK in untreated is aging. Below is an immunoblot of liver lysates collected from 6- and 24-week-old *Diesl* WT and KO mice:

Because AMPK responds to various stressors, it would be very interesting to monitor AMPK activation in *Diesl* KO mice that are exposed to different perturbations.

4. Does lipid starvation in the U251 cells activate autophagy pathways?

We observe no induction of autophagy in RPE1 cells, as determined by activation of mTORC1 (using pS6 as a surrogate, new Extended Data Fig. 9d) or by quantification of autophagosome number (new Extended Data Fig. 9e).

5. Some very nice experiments demonstrate the DIESL is an ER membrane glycoprotein, and the glycosylation status of this protein is characterized. However, recombinant DIESL was expressed in *E. coli* (using a bacterial membrane targeting sequence) to demonstrate its TAG synthesis capacity. Since these *E. coli* have presumably not been manipulated to be able to carry out glycosylation, what do these results say about the glycosylation of DIESL in human cells?

Our data indicate that glycosylation of DIESL is dispensable for its function, at least in the context of activation by TMX1 deletion. In our original manuscript, this was implied by the data in *E. coli*, which as the reviewer notes is an N-glycosylation-deficient organism. We now provide more evidence with the N5Q glycosylation mutant in (human) cells lacking TMX1, showing that this mutant is not deficient in TAG synthesis (new Extended Data Fig. 4j).

The information on glycosylation is of course relevant for this manuscript because it provides strong evidence for its membrane topology and cellular localization. We consider it possible that glycosylation may play a more subtle role in DIESL regulation (of its activity, or localization, etc.). However, in our experiments, the major determinants of DIESL activity appear to be its active site and TMX1.

6. The argument in lines 203 to 206 of the discussion about linkage between mouse liver DIESL and the NNT gene is not strong. Variations at NNT were proposed to explain differences in metabolism observed between C57BL/6J mice and other strains. However, this does not appear to be true for insulin secretion, a major mediator of lipid metabolism (see Wong et al. *Endocrinology* 2010; 151:96-102). There is a Blog Post on the Jax website that summarizes the effects of NNT on the response to diet-induced obesity in mice, and the consensus is that variations at this locus are not required for a robust response to high fat feeding. So the authors are left with limited evidence for the physiologic relevance of DIESL, even at the cellular level. A quick search of public databases suggests linkage between TMEM68 and height. Could this suggest a need for this pathway in normal growth?

We thank the referee for pointing this out. We were excited by the proteomics study linking NNT to DIESL and we used this to hint at potential physiological roles. We have now removed this from the discussion and believe that the strengthened cell culture data as well as the addition of the KO mouse and the phenotypes described here (listed in point 2) address the worry that DIESL may be irrelevant physiologically. Indeed, we do observe a link between mouse *Diesl* and body weight, as have been noted in GWAS data from other organisms, as well as other perturbations in TAG levels in certain tissues and disruption of liver lipid and stress homeostasis.

Referee #3 (Remarks to the Author):

In this manuscript, the Brummelkamp group has utilized their elegant CRISPR screening strategy in haploid cells to pin down an alternative mechanism for triacylglycerol (TAG) biosynthesis in human cells. They report on the identification of TMEM68, an acyltransferase of previously unknown function, that they rename DIESL as a TAG synthase, and TMX1, a PDI family oxidoreductase, as an interactor and negative regulator of DIESL.

Overall, this study identifies a potentially critical new player in the control of neutral lipid metabolism and several of the findings made appear relatively robust. TMX1 was identified first in their study and considerable effort was devoted for instance to exclude a role for other TMX family members (TMX2-4). Instead, the characterization of DIESL, its mechanism of action, regulation, and physiological significance remain less compelling, even though it represents the business end of the pathway. Moreover, considering that TMX1 and TMEM68/DIESL are previously at least partially characterized proteins, the findings raise questions related to their earlier reported properties. Please see specific comments below.

Major issues:

1. Membrane topology and activity of DIESL: The N-terminus is expected to be ER-luminal, if the N-glycosylation motif at N5-X-S/T is used, as reported in Extended Data Fig. 5. However, if expression of human DIESL in *E. coli* yields a functional TAG synthase, this implies that this glycosylation may not be needed for human DIESL to be active. This would be important to clarify. How is DIESL anchored to the membrane and how is the putative active site expected to be positioned relative to the bilayer and substrate?

We thank the reviewer for raising the point that the relevance of the DIESL glycosylation site and its topology requires some clarification. We initially included this data because (as the reviewer mentions) it indicates that the DIESL N-terminus is localized in the ER lumen. We now include data

from HAP1 cells showing that the glycosylation-deficient mutant (N5Q) can still synthesize TAG at levels similar to WT (new Extended Data Fig. 4j).

We have included additional data that pertains to the topology, localization and orientation of DIESL in an effort to further characterize this important “business end” of the pathway as noted by the reviewer:

1. Localization: While we previously showed that DIESL localized to the endoplasmic reticulum (ER) by microscopy (new Extended Data Fig. 4a), we additionally provide a subcellular fractionation experiment demonstrating enrichment of DIESL (and TMX1) at the mitochondria-ER contact site (new Fig. 5e).
2. Topology: we show that the AlphaFold model of DIESL demonstrates it to be a transmembrane protein (new Extended Data Fig. 4k). Combined with our knowledge that the N-terminus resides in the ER lumen, this indicates that the acyltransferase domain localizes to the cytosolic side of the ER. The model positions the acyltransferase domain in proximity of the membrane, with the active site pocket oriented towards the membrane (new Extended Data Fig. 4n).
3. Import/Stability: based on the AlphaFold model (new Extended Data Fig. 4k), we deleted the luminal domain of DIESL (the first 49 amino acids, new Extended Data Fig. 4l), hypothesizing that this may be a region important for its import into the ER and/or its stability. We show that truncated DIESL is no longer detectable in HAP1 cells (new Extended Data Fig. 4m), and thus this region may represent a cryptic import sequence.

Taken together, these new data demonstrate that DIESL is a type III transmembrane glycoprotein likely containing a cryptic import sequence and resides in the ER (at contact sites with mitochondria) with its active site available to lipid substrates in the membrane.

The interpretation of the BFA experiment (Extended Data Fig. 5e) does not seem to make sense. The authors claim that since BFA treatment results in an increased apparent MW of 3xHA-DIESL suggestive of elongated glycan chains, this indicates that DIESL faces the ER rather than the Golgi lumen. Since BFA leads to redistribution of cis- and medial-Golgi components to the ER, this does not differentiate between ER and Golgi (cis-medial) luminal compartments.

We apologize to the reviewer for not being clearer with our wording regarding this experiment. The goal of the BFA treatment was to merge Golgi glycosylases with resident ER glycoproteins, resulting in ectopic elongation of these shorter ER glycans (initially demonstrated by Lippincott-Schwartz et al. Cell 1989).

We have re-worded the text to read “these glycans could be elongated by treatment of cells with brefeldin A, which allows Golgi glycosylases access to ER resident proteins by merging both organelles” (lines 134-136).

The BFA experiment is now complementary to our microscopy data localizing DIESL to the ER (new Extended Data Fig. 4a) and thus the early secretory pathway. If the reviewer stills finds this set of experiments problematic we would also consider removing these data as the microscopy data is itself informative.

2.DAG acylation by DIESL and E.coli data: The acylation of DAG by DIESL is demonstrated by using a fluorescent non-natural DAG analog (NBD-DAG) only (Fig. 4c, Extended Data Fig. 9b-c). These data should be strengthened by using more natural DAG as substrate.

This is a critical point raised by the reviewer, which we have addressed using radioactive [¹⁴C]DAG in a cell-free assay. In this experiment (new Fig. 4h) demonstrates acylation of 'natural' DAG by DIESL, forming TAG in a manner sensitive both temperature and heat-inactivation.

The DIESL constructs showing activity in E.coli (Fig. 4d, Extended Data Fig. 9e,f) are not clear: does the triple HA-tag inhibit TAG generation in E. coli, since it is not shown in panel 9f?

There is no large difference in the activity of bacterial expression constructs with or without the HA tag as assessed by TLC (new Fig. 4e) – both variants produce similar amounts of TAG. For budgetary reasons we believed it would be redundant to carry out lipidomic analysis on both tagged and untagged DIESL and thus only one DIESL-expressing sample (in triplicate) was analyzed and compared to bacteria expressing an empty vector.

The E.coli phospholipid data (Extended Data Fig. 9g) is too crude to be informative: based on this one cannot make the conclusion that DIESL-synthesized TAGs reflect the cellular lipidome in length, saturation and complexity, as claimed.

We understand the concern and have thus removed this data.

3.TMX1 inhibition of DIESL: TMX1 active site cysteines are needed for its disulfide-reducing activity. Is TMX1 activity needed for inhibiting DIESL and/or for stabilizing DIESL protein levels?

This is also a very interesting point raised by the reviewer. To address this, we generated single and double cysteine-to-alanine mutants of TMX1 and expressed them in HAP1 cells. Both mutants were relatively unstable and therefore could not rescue lipid droplet formation (new Extended

Data Fig. 2e; the TMX1 redox cycle is depicted in new Extended Data Fig. 2d). While we find the concept of TMX1 oxidation as a regulator of DIESL activity compelling (especially with regards to the role of DIESL in regulating mitochondrial function), the data now generated using mutants only indicate that the activity of TMX1 is required for DIESL inhibition most likely in an indirect way via the regulation of its own stability.

How could the TMX1 inhibition of DIESL fit together with the finding that genetic disruption of TMX1 destabilizes the DIESL protein?

In the absence of TMX1, DIESL has no binding partner and becomes less stable. In this setting, DIESL would be turned over by ERAD-related processes. We also cannot exclude the possibility that TMX1 loss may influence DIESL levels indirectly (i.e. reduced mito-ER contacts and therefore a reduction in DIESL protein levels at these locations). Destabilization of DIESL would obviously oppose its activity, however this effect is not absolute as a significant amount of the protein still remains. Thus, the increase in DIESL activity is of such a magnitude that the reduction of activity stemming from destabilization of the protein is counteracted.

We believe that most likely the TMX1-DIESL complex becomes “active” via a mechanism that we have yet to understand (i.e. post-translational modification, changes in membrane composition, an activating binding partner, local changes in redox, etc.) and that TMX1 may act as the regulatory subunit of the complex. We look forward to elucidating the mechanism of DIESL activation in the future.

TMX1 has been localized to mitochondria-associated ER membranes (MAMs). Is DIESL also localized to MAMs?

As discussed in point 1, we have fractionated cells and found that DIESL and TMX1 are both enriched in the MAM fraction (new Fig. 5e).

4. Effects on mitochondrial function and energy homeostasis: This part of the manuscript related to the cell physiological relevance of the findings appears premature. Mitochondrial morphological changes can be induced by multiple indirect cues and are therefore rather uninformative. Also, it seems that the mitochondrial morphological groups (fused, intermediated, aggregated; Fig. 4f; Extended Data Fig. 10b) were determined by subjective assessment.

We agree with the reviewer in that our initial characterization of the mitochondrial phenotype of DIESL KO cells was incomplete. As for the subjectivity of mitochondrial measurements, assessing

mitochondrial morphology in this way is typical for the field but we have decided to leave out this data in the revision as we now have assessed mitochondrial health more directly in untransformed RPE1 epithelial cells. These new experiments are detailed below:

- 1) We measure mitochondrial membrane potential ($\Delta\Psi$) by TMRM and ATP in RPE1 cells, showing that DIESL KO cells are depolarized compared to their WT counterparts when lipoprotein-starved, resulting in deficits in ATP levels (new Fig. 5f). Importantly, this deficit is bypassed by the addition of fatty acids in a manner sensitive to fatty acid uptake by mitochondria (new Extended Data Fig. 9f).
- 2) We additionally measure mitochondrial reactive oxygen species (ROS) in RPE1 cells lacking DIESL, demonstrating that the KOs have increased ROS that is exacerbated by lipoprotein depletion (new Extended Data Fig. 9c).
- 3) We also show that RPE1 cells have activated (phosphorylated) AMPK when DIESL is lost (new Extended Data Fig. 9a). This is in alignment with our earlier experiments in U251 cells, now replicated in HT29 as well, demonstrating increased AMPK phosphorylation in DIESL KO cells, although these (transformed) cells require further lipoprotein starvation in order to observe this phenotype (also shown in Extended Data Fig. 9a). We additionally show a similar trend in 293T cells lacking DIESL, although this required DGAT inhibition in addition to lipoprotein starvation (Extended Data Fig. 9b). Taken together, these data indicate that AMPK in DIESL KO cells is more sensitive to activation, although the threshold for activation appears to vary with each cell line. These different thresholds for activation could be derived from differences in extracellular lipid uptake, mitochondrial activity, lipid storage or other expected variations between these normal and cancer cells.
- 4) Finally, we challenge various cell lines with reduced nutrient availability (glucose, amino acids and lipoproteins) and show that DIESL KO cells have decreased fitness compared to their WT counterparts (new Fig. 5g-h).

Taken together, these new data demonstrate that DIESL plays a role in maintaining mitochondrial function, particularly under conditions of lipid starvation. As i) this deficit can be rescued by the addition of free fatty acids, and ii) we have now localized DIESL to the MAM (new Fig. 5e), the mitochondrial dysfunction observed in the DIESL KO likely stems from a lack of fatty acids available to this organelle.

AMPK is activated under lipid starvation if DIESL is not functional (Fig. 4g). What does this mean, i.e. how is energy homeostasis supported by DIESL activity? Are cellular ATP levels decreased if DIESL is not functional under lipid starvation?

As mentioned above, ATP levels are perturbed in RPE1 cells lacking DIESL (both under lipoprotein-replete and depleted conditions, new Fig. 5f).

AMPK can promote mitochondrial biogenesis but how is this related to the aggregated mitochondrial phenotype?

The reviewer is correct in that AMPK promotes mitochondrial biogenesis in addition to its other, related roles in activation of the mitochondrial fission pathway and the mitophagy pathway. The “aggregated” mitochondrial phenotype scored in the first submission (now removed) is the result of unopposed mitochondrial fission (the mitochondrial fragments ultimately aggregate around the nucleus).

AMPK is known to respond to mitochondrial stress and acts directly on the fission pathway to stimulate mitochondrial division. At the same time, AMPK activates mitophagy to degrade dysfunctional mitochondrial fragments, thus “purifying” the mitochondrial reticulum. Finally, AMPK stimulates mitochondrial biogenesis in order to replace the dysfunctional organelles that have been degraded. This is illustrated in the below figure from a review from the Shaw group (Herzig & Shaw, NRMCB 2017):

Nature Reviews | Molecular Cell Biology

In the revised manuscript, we provide more direct assays of mitochondrial health and thus now no longer rely solely on AMPK activation and mitochondrial morphology, which we used previously as surrogates for ATP levels and mitochondrial health, respectively.

Additional comments:

5.The authors suggest that DIESTL-dependent TAG synthesis somehow helps to maintain energy homeostasis in limited extracellular lipid availability. The exact delipidation/lipid starvation conditions used

in each experiment should be clearly spelled out. Are other than lipid starvation cues relevant, does for instance amino acid starvation induce DIESL dependent TAG synthesis?

We used the same delipidated conditions in almost every experiment; we have adjusted the figures and text to read “lipoprotein-starved”. In some cases, “starved” is simply used due to space constraints but the exact nature of the starvation (almost exclusively, culturing in the absence of lipoproteins) is explained in the figure legend. At the very end of the manuscript (new Fig. 5h), we use nutrient-depleted medium (EBSS) and are thus more explicit in our labeling.

6. In the Discussion, the “two levels” at which DIESL-dependent TAG synthesis is linked to mitochondria/redox biology, is not clear; what are these levels?

These levels were TMX1 (an oxidoreductase localized to mito-ER contacts and regulator of DIESL activity in cells) and NNT (a redox-related mitochondrial protein and regulator of DIESL protein levels in the liver [Chick *et al.* Nature 2016]). The current discussion no longer mentions NNT.

7. Quantifications of individual lipid droplet sizes and numbers at the described light microscopic resolution are not accurate (Extended Data Fig. 2a). The area occupied by LDs per cell (middle panel) should suffice although it is not clear what the n’s here indicate.

The n’s in this panel indicated the number of lipid droplets that were counted in the analysis. We have removed this data in its entirety, including the area occupied by lipid droplets as this specific panel is redundant with the FACS measurements of BODIPY fluorescence shown in new Fig. 1e.

8. Strictly speaking, Extended Data Fig. 10a does not show DIESL inhibition (rather inhibition of DGATs in combination with DIESL CRISPR KO). The title of this figure should also be reformulated.

We thank the referee for pointing this out and we have changed the title of this figure (now new Extended Data Fig. 8) into: “Steady-state DIESL activity”.

Reviewer Reports on the First Revision:

Referee expertise:

Referees' comments:

Referee #1 (Remarks to the Author):

This manuscript reports genetic studies that uncover a novel function for the protein, TMEM68, as DIESL, an enzyme that synthesizes TAG from DAG, and its regulation by the interacting protein TMX1. The authors have responded in detail to the issues raised in the initial review and provided substantial new data. The data provide strong evidence that DIESL and TMX1 together control a novel pathway for TAG synthesis in cultured cells and that DIESL functions as a TAG synthetase enzyme that acylates DAG.

Major concerns:

1. The authors now include data from a mouse model with germline DIESL knockout. However, characterization of this model is too preliminary to provide a clear and compelling picture of the physiological role of this ubiquitously expressed enzyme. It is unclear what tissues are contributing to the observations, or whether there are developmental effects. Interpretation of difference in weights of adult animals would be aided by information on impact on growth, body composition, diet, food intake, nutrient absorption, and energy expenditure. Effect on steady state serum TAG, shown for males and females together, is modest (appears to be driven by one outlier animal). Is this driven by differences in lipoprotein secretion or metabolism? All metabolic parameters should be reported separately for males and females. Conditions under which measures are made or tissues are harvested (hours of fast vs fed) should also be specified, as these could have a major effect on outcomes. The marked increase in CHOP and increase in PA and DAG in the liver are intriguing, but more data would be needed to understand the significance of these findings for liver physiology.

2. Analysis of a variety of immortalized, dispersed, rapidly growing cultured cell lines with CRISPR deletion of DIESL suggest differences in AMPK signaling, mitochondrial function and cell growth under lipoprotein deficient conditions. However, this does not elucidate the physiological significance of this enzyme. It is unclear how these findings link to altered PA/DAG in liver or TAG in the brain, as these are non-transformed primary somatic tissues.

Minor Comments:

3. Throughout the manuscript, thin layer chromatography, immunoprecipitation, and immunoblot data are presented as an image of one TLC plate or membrane. In addition to these representative images, figures should include quantification and legends should include statement of the number of independent experiments for which quantification is provided.

4. In cellular studies with targeted knockout of DIESL or TMX1 using CRISPR, it should be stated how

many independent clonal lines were studied for each manipulation. Given the possibility of off-target effects, demonstration that phenotypes can be rescued by re-expression of the targeted genes would strengthen the observations.

5. In experiments with N-terminal truncations of DIESL or disruption of TMX1, decreased abundance of the mutated or wild type DIESL are interpreted as showing changes in protein stability. Controls to show absence of change in mRNA and metabolic labeling of proteins would be necessary to establish that these findings resulted from increased protein degradation.

6. The Methods report that the primary screen was performed by gene-trap mutagenesis. Please specify whether the suppressor screen was performed with this method as well and provide data on how many gene disruptions per cell.

Referee #2 (Remarks to the Author):

The authors have responded with impressive new data to satisfy my concerns.

Specifically, they developed Diesl knockout mice and these animals have a phenotype, strongly suggesting physiological relevance for this alternative TAG biosynthesis pathway. They have also shown that Diesl deficiency affects mitochondrial function, did not find induction of autophagy, addressed my concerns about glycosylation, and removed NNT information.

Figure 5A convincingly shows that both male and female Diesl knockout mice weigh less than controls. These animals have decreased serum triglycerides and brain triglycerides. Given the significance of findings in this paper, the work would be substantially enhanced with two pieces of data. First, please provide genetic confirmation in the knockout and control mice showing that the loss of function allele depicted schematically in Extended Data 7A is present in the knockouts and absent in the controls. This might be done by PCR mapping of exons 4 and 5 of the mouse Diesl gene, or perhaps by showing that Diesl mRNA or protein is lost in the knockouts. Second, please provide body composition analyses for the knockout and control mice, i.e. is there a decrease in adiposity in the knockout animals? Do the animals have the same nasal-anal length? I am not asking for a physiologic explanation of the decrease in body weight. However, the body composition and size data will be important for future work.

As a minor point, on line 186, "mi" should be "mice".

Referee #3 (Remarks to the Author):

Overall, the authors have done a good job in responding to the concerns raised and the extensive additional data included during the revision further strengthen this remarkable manuscript.

Yet, introduction of the DIESL knockout mouse and the role of DIESL in mitochondrial functions in the revised manuscript beg some clarifications as follows:

1. The levels of ether PCs (ePC) should be included in Fig. 5D: This figure shows that there is no change in the abundance of total PC between WT and DIESL KO mouse livers, but ePC might actually be more affected considering that the major effects of DIESL activity in cells are observed in the abundance of ePC rather than PC (Extended Data Fig. 5).
2. In Fig. 5F (and e.g. Extended Data Fig. 9F) the addition of exogenous oleic acid rescues the phenotypes in sgDIESL cells. How does fit together with the authors' proposal that DIESL uses endogenous fatty acids as substrate? There is probably a reduction in mito membrane potential and energy phosphates in sgDIESL compared to sgCTRL already in the lipoprotein-starved conditions, although this is not indicated.
3. Extended Data Fig. 7C: The brain is apparently one of the organs where reduced TAGs are observed in DIESL KO mice. The data are as abundance mol% (a value of roughly 0.09 for control) - but of what?
4. The reduced body weight of the DIESL KO animals is reported at 22-28 weeks and most of the other parameters, such as serum TAG and liver CHOP levels, at 6 weeks. Does this imply that the weight reduction is a late phenotype?
5. The markers used in WB of Fig. 5E should be explained.
6. The expression levels of DIESL (and TMX) in different murine tissues should be commented on, especially considering that DIESL was originally identified as a brain-specific protein.

Author Rebuttals to First Revision:

Referees' comments:

Referee #1 (Remarks to the Author):

This manuscript reports genetic studies that uncover a novel function for the protein, TMEM68, as DIESL, an enzyme that synthesizes TAG from DAG, and its regulation by the interacting protein TMX1. The authors have responded in detail to the issues raised in the initial review and provided substantial new data. The data provide strong evidence that DIESL and TMX1 together control a novel pathway for TAG synthesis in cultured cells and that DIESL functions as a TAG synthetase enzyme that acylates DAG.

We appreciate the reviewer's assessment that we have identified a new TAG synthesis pathway.

Major concerns:

The authors now include data from a mouse model with germline DIESL knockout. However, characterization of this model is too preliminary to provide a clear and compelling picture of the physiological role of this ubiquitously expressed enzyme.

While our initial characterization of the mouse was perhaps preliminary, we included it to demonstrate the importance of DIESL in organismal physiology. In this new revision, we have taken the reviewer's valuable suggestions on the mouse studies to heart. We now characterize the metabolic phenotype of adult KO mice in detail using the analyses outlined by the reviewer, and additionally report a role for DIESL during postnatal growth in younger mice. These data are all described in the sections below and are included in the manuscript.

To summarize the phenotype in the adult mice, *Diesl* KOs display a reduction in both body weight (as reported in the previous submission) as well as body length. Organ weights in the KO are not specifically smaller, as they scale with body weight. We additionally observe a reduction in fat mass ($p = 0.035$), as well as a smaller reduction in lean mass ($p = 0.11$). Using calorimetric measurements (VCO_2/VO_2), we now show that adult KO mice have a perturbed response to fasting, characterized by an impaired switch from carbohydrate to lipid utilization. These new metabolic data demonstrate that DIESL is a crucial metabolic enzyme that governs energy utilization, and are consistent with our genetic and biochemical data demonstrating that DIESL is a TAG synthase.

The new, more precise analyses of the mice have given us a very important insight into the development of the phenotype of the *Diesl* KO. Strikingly, DIESL plays an integral role in organismal growth at a specific postnatal developmental stage; *Diesl* KOs fail to put on as much weight as WT mice during a critical 2-week period encompassing 2 to 4 weeks after birth. This period coincides with a reduction in lipid content of the diet, as mice switch from milk to solid chow.

Thus, we can conclude that DIESL is certainly important for organismal growth, and we report two conditions

when *Diesl*-deficient mice present a metabolic phenotype: (i) during a period of rapid organismal growth that is associated with the timing of a diet switch from milk (containing 20% fat [Yajima *et al.*, *Exp. Anim.* 2006]) to chow (containing 4.2 % fat), and (ii) during fasting that is associated with an impaired switch from carbohydrate to lipid utilization.

It is unclear what tissues are contributing to the observations, or whether there are developmental effects.

This question turned out to be most important. To address this, we followed KO mice from birth to adulthood. *Diesl*/KO mice are born at expected Mendelian ratios and at equal weights, and are accordingly indistinguishable from WT mice. The weight phenotype manifests between postnatal weeks two and four, as outlined above. This is precisely when mice shift their diet from milk (20% fat) to solid food (4% fat). After week 4, there is no growth delay and the weight difference remains constant throughout life. At week 3 (but not a week 6) we measure a 40% reduction in liver triglycerides in *DIESL* KO mice, indicative of a metabolic defect relating to energy availability.

Interpretation of difference in weights of adult animals would be aided by information on impact on growth, body composition, diet, food intake, nutrient absorption, and energy expenditure.

At the request of the reviewer, we have now performed in-depth characterization of the metabolic phenotype of these mice. To this end, we now include data on body composition and length, food intake, respiratory exchange ratio, energy expenditure, and other calorimetric parameters under both the basal steady-state condition and under metabolic challenges that stimulate lipid oxidation.

We now report that *Diesl* KO mice are smaller in length in addition to being lighter in weight, and see no difference in food intake or energy expenditure. We additionally provide detailed growth curves in the first weeks of life.

Although it was not specifically asked, we also provide calorimetric data (RER) demonstrating a preferred usage of carbohydrates rather than lipids by *Diesl* KO mice during fasting.

Effect on steady state serum TAG, shown for males and females together, is modest (appears to be driven by one outlier animal). Is this driven by differences in lipoprotein secretion or metabolism?

We agree that this difference was modest. However, it turned out not to be caused by the outlier animal because the data remains significant when this outlier is removed. We have refocused our new mouse data on the emergence of the phenotype in young mice (two to four weeks old) and in fasted adult mice. Thus, we have moved the small difference in circulating TAGs in adults to the supplemental data (Extended Fig. 9F).

All metabolic parameters should be reported separately for males and females. Conditions under which measures are made or tissues are harvested (hours of fast vs fed) should also be specified, as these could have a major effect on outcomes.

We appreciate this request by the referee. Focusing on the body weight/size phenotype, we have measured body weight after birth weekly into adulthood in both male and female littermates in addition to the previously-reported weight differences in adult mice. We noticed a similar phenotype in males and females: no weight difference at birth, a decline in growth during weeks 2-4 resulting in lighter/smaller mice in adulthood. All these measurements were done with sufficient animals for both sexes and genotypes to reach highly significant conclusions.

However, obtaining age-matched littermates for these studies of both genotypes required the generation of more than 124 mice for the revision which was the maximum that we could achieve (per litter of ~6 mice we obtain only 0.75 male homozygous KO). Given that females were also needed for breeding, we have restricted the experiments for which mice needed to be sacrificed to males. We hope that having measured the beginning (weight after 1 week) and the end point (adult weight) and having determined the same onset of weight decline identical in both sexes, the additional mechanistic characterization in male mice is likely representative for both sexes although this will require additional future studies.

The marked increase in CHOP and increase in PA and DAG in the liver are intriguing, but more data would be needed to understand the significance of these findings for liver physiology.

During this revision, we found that the most obvious phenotype (reduced bodyweight) arose at 2-4 weeks after birth. To further test the relation of CHOP activation to the body weight phenotype, we have now looked at different time-points and tissues in mice; this yielded two new insights. First, as shown (referee figure R1A, next page) the CHOP response is not limited to liver tissue, as it could be detected in lung and kidney. Furthermore, we confirmed our previous observation that CHOP was elevated at week 6 after birth, but this elevation was not yet present at week 3.5 (referee figure R1B). In addition, CHOP levels became comparable between WT and KO mice at week 10 (referee figure R1B). After week 2-4 there is no growth delay and the weight difference remains identical. Thus, CHOP elevation at week 6 was likely a response to DIESL loss that was not causally involved in the weight/body size phenotype. We therefore suggest to remove this data from our manuscript.

A

B

Referee Figure R1. Kinetics of the CHOP response in *Diesl* KO mice. **a**, Immunoblot analysis (left) and quantification (right) of CHOP in lysates prepared from liver, kidney and lung harvested from 6-week-old WT and *Diesl* KO mice ($n = 3$ mice per genotype; two-tailed Student's t test). **b**, Immunoblot analysis of CHOP in liver lysates collected from 3.5-, 6- and 10-week-old mice ($n = 4$ mice per genotype).

2. Analysis of a variety of immortalized, dispersed, rapidly growing cultured cell lines with CRISPR deletion of *DIESL* suggest differences in AMPK signaling, mitochondrial function and cell growth under lipoprotein deficient conditions. However, this does not elucidate the physiological significance of this enzyme. It is unclear how these findings link to altered PA/DAG in liver or TAG in the brain, as these are non-transformed primary somatic tissues.

Knockout mice and cancer cells cultured in a dish represent two different systems with which to study *DIESL* loss. In the current study, the cancer cell line data and the mouse analyses serve two different purposes.

Our cell line data enabled us to study the *DIESL* pathway in isolation from other triglyceride sources such as extracellular lipids and the DGATs; we could selectively remove nutrients and/or affect the enzymatic activity of known TAG synthases and determine the outcome of loss of *DIESL* function. While *DIESL* does indeed make TAG, these experiments demonstrated that the *DIESL*-made TAGs can be used by mitochondria in a context that was especially notable when external lipid sources became limiting. Thus, the experiments with manipulated cells provide important insight on what *DIESL* can do although they do not necessarily provide insight into its physiological function.

In mice, this reductive experimentation would be very challenging; in addition to *DIESL*, mice have two DGAT enzymes that, when deleted, result in lethality and the analysis is further complicated by dietary lipids and lipid distribution across tissues. Our purpose for including the mouse was to demonstrate physiological relevance for *DIESL* function, as this was requested by the reviewers. We thus generated the KO mouse during the first revision to show beyond a doubt that *DIESL* is important for normal physiology in the organism.

The more detailed analysis of *DIESL*-deficient mice that we provide now gives further insight into its physiological role. While mice are born at equal weights, KO mice fail to grow at the rate of WT mice between the ages of two and four weeks. This is the postnatal stage at which pups switch from the milk of their mother (20% fat) to solid chow (4.2% fat). This reduction in growth is associated with energy shortage, as liver TAGs are reduced by 40% at this age. We also analyzed the effect of *DIESL* loss in adult mice in response to nutrient starvation. Adult mice respond to fasting by a switch from glucose to lipid oxidation. *DIESL*-deficient mice however retain increased carbohydrate metabolism during fasting. Thus, in cultured cells we observe that *DIESL* becomes most critical when the cells are starved for externally provided lipids and *DIESL*-deficient mice show a growth delay that is associated with the timing

of a switch from a high fat diet (milk) to a normal diet (chow) and show decreased lipid oxidation upon fasting.

After ~4 weeks of age *DIESL*-deficient mice grow at the same rate as WT mice. As this is not associated with changes in food uptake or energy expenditure, the most plausible explanation is that this results from metabolic rewiring which may involve lower TAG synthesis in the brain and less fat storage in general (Extended Data Figure 9E-G). However, with the presentation of our more detailed analysis we chose to de-emphasized the liver PA/DAG and brain TAG data in older mice.

Minor Comments:

3. Throughout the manuscript, thin layer chromatography, immunoprecipitation, and immunoblot data are presented as an image of one TLC plate or membrane. In addition to these representative images, figures should include quantification and legends should include statement of the number of independent experiments for which quantification is provided.

We now show quantification and/or describe replication for all TLCs, IPs and immunoblots. We have inserted histograms into the figures and explicitly mention independent experiments in the figure legends.

4. In cellular studies with targeted knockout of *DIESL* or *TMX1* using CRISPR, it should be stated how many independent clonal lines were studied for each manipulation. Given the possibility of off-target effects, demonstration that phenotypes can be rescued by re-expression of the targeted genes would strengthen the observations.

For clonal knockouts, we have replicated results in at least two independent cell clones; the number of clones and the sequence of the indicated edited gene are shown in Supplementary Table 3. For our mitochondrial studies, we used pools of KO cells to avoid clonal artifacts.

We have indeed rescued phenotypes of both *TMX1* and *DIESL* KO in cells:

- rescue of the lipid droplet and TAG phenotypes in *TMX1* KO: Extended Data Fig. 2J and K
- rescue of *DIESL*-dependent TAG synthesis in *TMX1* *DIESL* DKO: Fig. 2H, Fig. 3E, Extended Data Fig. 4L
- rescue of *DIESL*-dependent TAG synthesis in 4KO cells (including lipidomics): Fig. 4B and C, Extended Data Fig. 5B-G
- rescue of *DIESL*-dependent DAG acylation in 4KO cells: Extended Data Fig. 6B and D

5. In experiments with N-terminal truncations of *DIESL* or disruption of *TMX1*, decreased abundance of the mutated or wild type *DIESL* are interpreted as showing changes in protein stability. Controls to show absence of change in mRNA and metabolic labeling of proteins would be necessary to establish that these findings resulted from increased protein degradation.

We agree with the reviewer that our interpretations of perturbed DIESL proteins levels in the absence of TMX1 were perhaps too strong. We have now performed cycloheximide chase experiments to show that the protein half-life is diminished in the absence of TMX1. In the text, we have rephrased our conclusion and state '*When protein synthesis was inhibited, deficiency of TMX1 led to an increased turnover of DIESL protein (Extended Data Fig. 4c) which could indicate the existence of a stable TMX1-DIESL complex*'.

6. The Methods report that the primary screen was performed by gene-trap mutagenesis. Please specify whether the suppressor screen was performed with this method as well and provide data on how many gene disruptions per cell.

All three screens presented in this manuscript (*DGATDKO*, WT with oleic acid, *TMX1 KO*) were performed in HAP1 cells using gene-trap mutagenesis to generate mutant libraries. This is now explicitly described in the Methods, under the Mutagenesis Screening subheading: "*All genetic screens reported here employ gene-trap mutagenesis in haploid (HAP1) cells...*". The number of unique mutations recovered in each population has also been indicated in Supplementary Table 4. Also, all sequencing files related to the screens will be deposited in the NCBI Short Read Sequence Archive and the analyzed screen data will be available at <https://phenosaurus.nki.nl>. In addition, analyzed data corresponding to all presented screen graphs will be provided as supplementary excel tables.

Referee #2 (Remarks to the Author):

The authors have responded with impressive new data to satisfy my concerns.

Specifically, they developed Diesl knockout mice and these animals have a phenotype, strongly suggesting physiological relevance for this alternative TAG biosynthesis pathway. They have also shown that Diesl deficiency affects mitochondrial function, did not find induction of autophagy, addressed my concerns about glycosylation, and removed NNT information.

Figure 5A convincingly shows that both male and female Diesl knockout mice weigh less than controls. These animals have decreased serum triglycerides and brain triglycerides. Given the significance of findings in this paper, the work would be substantially enhanced with two pieces of data. First, please provide genetic confirmation in the knockout and control mice showing that the loss of function allele depicted schematically in Extended Data 7A is present in the knockouts and absent in the controls. This might be done by PCR mapping of exons 4 and 5 of the mouse Diesl gene, or perhaps by showing that Diesl mRNA or protein is lost in the knockouts.

We thank the reviewer for these excellent suggestions. We have included both a PCR amplifying either the transgene or intact locus, as well as an RT-PCR showing that *DIESL* mRNA was undetectable in the KO (Extended Data Fig. 9b and c).

Second, please provide body composition analyses for the knockout and control mice, i.e. is there a decrease in adiposity in the knockout animals? Do the animals have the same nasal-anal length? I am not asking for a physiologic explanation of the decrease in body weight. However, the body composition and size data will be important for future work.

We now provide detailed analyses of both body length and composition. Body length is decreased in both male and female mice, and we observe additionally a reduction in fat mass in adult mice (Extended Data Fig. 9d and e).

As a minor point, on line 186, “mi” should be “mice”.

We have made the requested change.

Referee #3 (Remarks to the Author):

Overall, the authors have done a good job in responding to the concerns raised and the extensive additional data included during the revision further strengthen this remarkable manuscript.

Yet, introduction of the DIESEL knockout mouse and the role of DIESEL in mitochondrial functions in the revised manuscript beg some clarifications as follows:

1. The levels of ether PCs (ePC) should be included in Fig. 5D: This figure shows that there is no change in the abundance of total PC between WT and DIESEL KO mouse livers, but ePC might actually be more affected considering that the major effects of DIESEL activity in cells are observed in the abundance of ePC rather than PC (Extended Data Fig. 5).

We thank the reviewer for this suggestion. We have now performed deeper lipidomic analysis of the livers of KO and control mice, at two time points, and have not robustly detected ether-linked PCs in the liver. Because we cannot quantify ePCs in a reliable manner in the livers of control mice we consider two possibilities that are not mutually-exclusive:

1. That HAP1 cells have a high amount of ePCs that is not reflective of what is present in the mouse liver.

2. That liver tissue has a low amount of ePCs. Perhaps, we should have analyzed another tissue (eg brain) to detect these more complex lipids.

Although we believe that this observation is interesting, it is not critical to our conclusions and we will therefore not mention this in the main text (although all lipidomic data will remain accessible to the readers and presented in supplemental figures).

2. In Fig. 5F (and e.g. Extended Data Fig. 9F) the addition of exogenous oleic acid rescues the phenotypes in sgDIESL cells. How does it fit together with the authors' proposal that DIESL uses endogenous fatty acids as substrate?

Under conditions of lipid starvation (and thus depletion of the DGAT substrate), cells require DIESL activity to maintain viability. We believe we can rescue the fitness defect of *DIESL* KO cells with the addition of exogenous oleic acid because it is a way to provide fatty acids to mitochondria (the rescue is sensitive to the CPT1A inhibitor etomoxir [Extended Data Fig. 8F]).

There is probably a reduction in mito membrane potential and energy phosphates in sgDIESL compared to sgCTRL already in the lipoprotein-starved conditions, although this is not indicated.

We thank the reviewer for pointing this out and have now indicated it in the figure, however this difference is not significant ($p > 0.05$).

3. Extended Data Fig. 7C: The brain is apparently one of the organs where reduced TAGs are observed in DIESL KO mice. The data are as abundance mol% (a value of roughly 0.09 for control) - but of what?

This is the molar percentage of all the lipids detected; this is now indicated in the Methods, at the very end of the Shotgun Lipidomics / Sample Preparation subheading: "*Lipid abundancies are expressed either in weight (where indicated, corrected for the amount of protein in the sample) or as a molar percentage of the total lipid species in the sample (mol%).*"

4. The reduced body weight of the DIESL KO animals is reported at 22-28 weeks and most of the other parameters, such as serum TAG and liver CHOP levels, at 6 weeks. Does this imply that the weight reduction is a late phenotype?

The reviewer is correct to point out that we have taken different measurements at different stages of life. This was done because we did not know what to expect in advance. In this current revision, we have performed extensive mouse analyses and show that the weight reduction stems from an early postnatal phase (2-4 weeks) and is thus maintained throughout life. Our new lipidomic analyses were thus performed at 3.5 weeks and (again) at 6 weeks.

5. The markers used in WB of Fig. 5E should be explained.

We now explain these markers in the figure legend.

6. The expression levels of DIESL (and TMX) in different murine tissues should be commented on, especially considering that DIESL was originally identified as a brain-specific protein.

We thank the reviewer for this concern. As shown in Extended Data Fig. 9C, we can readily detect *DIESL* mRNA in the liver. In agreement with this, TMEM68 protein has been detected in the mouse liver by mass spectrometry (Chick *et al.*, *Nature* 2016) as well as in most other tissues (<https://www.ebi.ac.uk/gxa/experiments/E-PROT-13/Results?specific=true&geneQuery=%255B%257B%2522value%2522%253A%2522TMEM68%2522%252C%2522category%2522%253A%2522symbol%2522%257D%255D&filterFactors=%257B%257D&cutoff=%257B%2522value%2522%253A0.000001%257D>).

Human RNA sequencing data from the Human Protein Atlas dataset demonstrates that both *DIESL* and *TMX1* are expressed across all tissues tested ($n = 43$), and their expression correlates to a certain extent (Pearson's $r = 0.2413$, $p = 0.12$). Thus, analysis on RNA as well as protein level in humans and mice show expression beyond brain.

At the request of the reviewer, we have included the following statement in the Methods, at the end of the section describing RT-PCR: “*While DIESL has previously been reported to be a brain-specific protein, we readily detected DIESL expression in the mouse liver, in agreement with proteomic data. Additionally, expression data from the Human Protein Atlas indicates that DIESL expression could be detected in all tissues that were tested.*”.

Reviewer Reports on the Second Revision:

Referees' comments:

Referee #1 (Remarks to the Author):

Using carefully designed genetic screens, the authors identify a novel pathway for triacylglycerol (TAG) synthesis in human cells, involving DIESL and its negative regulator, TMX1. In detailed cell biological, biochemical and physiological analyses, the work demonstrates that this pathway functions to support TAG synthesis and maintain mitochondrial function under periods of lipid starvation.

The authors have carefully and constructively addressed the points raised in the previous review. The responses and substantial new data provided clarify and extend the prior submission. In its current form, the manuscript provides convincing data in support of a novel, alternative TAG biosynthesis pathway and insights into its physiological role. This represents an important advance in the field.

Minor points:

1. Please add to the methods the breeding scheme used to generate mice for study, specifically indicating whether these animals were derived from cross of heterozygote breeders and whether analyses used littermate KO and controls (WT).
2. In figures Extended Data Fig.2E (mutation of TMX1 redox cysteines) and Extended Data Fig 4O (mutation of DIESL lacking luminal domain), the authors interpret lower steady state mutant protein abundance as an indication of protein instability. Unless the authors can provide additional data showing equivalent mRNA levels and metabolic labeling for protein half-life, these statements in the text should be changed to indicate that these mutants are simply expressed at lower levels.
3. In figure Extended Data Fig. 2I-K, expression of TMX2 is much lower than the other TMXs assayed. While conclusions for TMX3 and TMX4 are well supported by the data, it is not possible to rule out that failure of TMX2 to suppress TAG and lipid droplet accumulation is simply a reflection of lower expression. Wording should reflect this limitation.
4. In Extended Data Fig. 5, please replace the yellow bars with a higher contrast color.
5. In Extended Data Fig. 5 or in the Methods, please indicate whether quantification of lipid species in lipidomics analyses were corrected for multiple comparisons.
6. In Extended Fig. 9 legend, please either make a broad statement that all mice analyzed were male, or add indication of sex to legends for panels J and K. Please add the fed or fasted (number of hours) conditions under which livers were harvested to legends for panels I and m.

Referee #2 (Remarks to the Author):

1. This revised version addresses my previous concerns about genetic confirmation of the knockout and characterization of body composition/body length.
2. It is potentially important that growth in the knockout mice is impaired during a specific postnatal time frame associated with a substantial nutritional transition.
3. The indirect calorimetry data in this revised version should be analyzed using regression approaches. Line 560 states that values are corrected for lean mass of each individual mice [sic], which may be incorrect. It is more appropriate to use an open source application like CalR (go to calrapp.org). These analyses should be applied to data in Extended Data Figure 9K and Figure 5G.
4. The food intake data of Extended data Figure 9J should be shown as food intake and not corrected for body weight.
5. The sentence spanning lines 49 to 52 is misleading. High levels of TAGs do not lead to obesity and metabolic syndrome. There is no evidence that low levels of TAGs contribute to cachexia. For both of these clinical conditions, TAGs may be associated, but the physiology is complex. For example, cachexia can be associated with very high levels of TAGs due to inhibition of lipoprotein lipase.
6. The phrase on line 184 referring to Figure 4H is confusing. Do the authors mean “sensitive to heat-inactivation and LOW temperature”?

Referee #3 (Remarks to the Author):

The authors have responded satisfactorily to my concerns.

The more thorough analysis of the DIESL knockout mice helps to better understand the physiological context in which this enzyme acts. The new data on liver lipidomics failed to detect ether PCs – possibly because ePCs are not very abundant in the liver. However, a reduction in liver TAGs was observed transiently at 3.5 weeks in male mice. This is reassuring. Yet, the statement of “a severe reduction (40%) in liver TAG”, which is essentially based on 2 animals, should be tuned down, or additional supporting data should be provided.

Author Rebuttals to Second Revision:

Referees' comments:

Referee #1 (Remarks to the Author):

Using carefully designed genetic screens, the authors identify a novel pathway for triacylglycerol (TAG) synthesis in human cells, involving DIESL and its negative regulator, TMX1. In detailed cell biological, biochemical and physiological analyses, the work demonstrates that this pathway functions to support TAG synthesis and maintain mitochondrial function under periods of lipid starvation.

The authors have carefully and constructively addressed the points raised in the previous review. The responses and substantial new data provided clarify and extend the prior submission. In its current form, the manuscript provides convincing data in support of a novel, alternative TAG biosynthesis pathway and insights into its physiological role. This represents an important advance in the field.

Minor points:

1. Please add to the methods the breeding scheme used to generate mice for study, specifically indicating whether these animals were derived from cross of heterozygote breeders and whether analyses used littermate KO and controls (WT).

We have added the following sentence in the **Methods** concerning **Mice**, within the subsection **Generation and genotyping** (lines 532-533): *"To generate mice for analysis, mice heterozygous for the mutant Diesl allele were intercrossed to generate knockout and control mice."*

2. In figures Extended Data Fig.2E (mutation of TMX1 redox cysteines) and Extended Data Fig 4O (mutation of DIESL lacking luminal domain), the authors interpret lower steady state mutant protein abundance as an indication of protein instability. Unless the authors can provide additional data showing equivalent mRNA levels and metabolic labeling for protein half-life, these statements in the text should be changed to indicate that these mutants are simply expressed at lower levels.

The reviewer is correct, and we have made the appropriate changes in the text:

Extended Data Fig. 2E (lines 86-87): *"Mutation of these redox cysteines correlated with a severe reduction in TMX1 protein abundance"*

Extended Data Fig. 4O (lines 148-150): *“the expression of truncated DIESL lacking this N-terminal luminal domain was undetectable by immunoblot”*

3. In figure Extended Data Fig. 2I-K, expression of TMX2 is much lower than the other TMXs assayed. While conclusions for TMX3 and TMX4 are well supported by the data, it is not possible to rule out that failure of TMX2 to suppress TAG and lipid droplet accumulation is simply a reflection of lower expression. Wording should reflect this limitation.

We have changed the wording to focus on TMX3 and TMX4 (lines 90-92): *“Unlike TMX1, TMX3 and the evolutionarily-similar TMX4 failed to suppress TAG and lipid droplet accumulation in these cells (Extended Data Fig. 2i-k; TMX2 was expressed at lower levels).”*

4. In Extended Data Fig. 5, please replace the yellow bars with a higher contrast color.

We have now replaced the blue/yellow palette with a red/blue one.

5. In Extended Data Fig. 5 or in the Methods, please indicate whether quantification of lipid species in lipidomics analyses were corrected for multiple comparisons.

We indeed corrected for multiple comparisons using a Bonferroni *post-hoc* analysis. In Extended Data Fig. 5, this is now explicitly stated: *“two-way ANOVA, Bonferroni multiple-comparison correction”*

6. In Extended Fig. 9 legend, please either make a broad statement that all mice analyzed were male, or add indication of sex to legends for panels J and K. Please add the fed or fasted (number of hours) conditions under which livers were harvested to legends for panels I and m.

We have explicitly indicated the sex of the mice in the figure legend. We have added that the mice were fed ad libitum in the legend for old panels L and M (new panels K and L), and have additionally modified the following statement under the **Sample Preparation** subheading in the **Shotgun Lipidomics** section of the **Methods** (lines 579-580) to read: *“For mouse tissue samples, tissues were collected from mice fed ad libitum and stored at -80°C.”*

Referee #2 (Remarks to the Author):

1. This revised version addresses my previous concerns about genetic confirmation of the knockout and characterization of body composition/body length.

2. It is potentially important that growth in the knockout mice is impaired during a specific postnatal time frame associated with a substantial nutritional transition.

3. The indirect calorimetry data in this revised version should be analyzed using regression approaches. Line 560 states that values are corrected for lean mass of each individual mice [sic], which may be incorrect. It is more appropriate to use an open source application like CalR (go to calrapp.org). These analyses should be applied to data in Extended Data Figure 9K and Figure 5G.

We thank the referee for this suggestion, and we have taken the following approach to address this:

1. We have re-analyzed the data using the CalR application as was suggested. Regression analysis indicated that the that the differences in the calorimetry data remained significant. However, this application gives a warning for the use of regression analysis with the small number of samples contained within the experiment (in agreement with common literature).
2. As such, we have left Figure 5G (RER) in the manuscript and have clearly indicated the caveat of the low statistical power in the main text (ANOVA analysis for such experiments, however, is commonly used in the literature).
3. We have removed the energy expenditure experiment (old Extended Data Figure 9K) because it reported no significant difference, whereas energy expenditure measurements are indeed typically analyzed using regression analysis.
4. Given the above three points, we adjusted our conclusions to reflect the limitation in statistical power in the experiment:
 - a. The abstract now reads: *"In mice, Diesel deficiency affects rapid postnatal growth and energy homeostasis coinciding with changes in nutrient availability"*. This is toned-down from a specific reference to fasting.
 - b. The end of the results section now reads: *"We used indirect calorimetry in fasted adult mice to measure energy homeostasis under conditions of nutrient shortage. We observed an increased respiratory exchange ratio (RER) in response to fasting in DIESEL-deficient mice (significant by ANOVA, although n was too low for reliable regression analysis; Fig. 5g), suggesting an attenuated switch from carbohydrate to lipid oxidation compared to wild-type mice. Taken together, TMX1-DIESEL is essential to support rapid organismal growth of young animals and regulates a response to dynamic changes in nutrient availability."* This statement acknowledges the limitations of the experiment.
 - c. Lines 268-269 in the Discussion summarize these data in a toned-down manner: *"Additionally, fasting of adult mice lacking Diesel suggests an impaired switch from carbohydrate-to-lipids for nutrient oxidation."*

We think that presenting the data (accompanied with careful conclusions and caveats) will be of interest for the community. However, if the referee and editor prefer, we would also consider leaving it out of the manuscript in its current form.

4. The food intake data of Extended data Figure 9J should be shown as food intake and not corrected for body weight.

We now show both uncorrected and body weight-corrected food intake (Extended Data Fig. 9J). The text now reads (lines 235-237): *“KO mice displayed a small decrease in food intake that was absent when this was adjusted for body weight (Extended Data Fig. 9j).”*

5. The sentence spanning lines 49 to 52 is misleading. High levels of TAGs do not lead to obesity and metabolic syndrome. There is no evidence that low levels of TAGs contribute to cachexia. For both of these clinical conditions, TAGs may be associated, but the physiology is complex. For example, cachexia can be associated with very high levels of TAGs due to inhibition of lipoprotein lipase.

We agree with the reviewer, and have modified these statements to read:

“In humans, most (if not all) cell types are able to synthesize triglycerides. High levels of TAGs (hypertriglyceridemia) are associated with obesity and metabolic syndrome, and the mobilization of TAGs from adipose tissue can contribute to cachexia, a multiorgan wasting disorder.”

We have additionally cited here the 2011 study by *Das et al.* demonstrating that disruption of TAG lipolysis rescues cancer-associated cachexia in mice.

6. The phrase on line 184 referring to Figure 4H is confusing. Do the authors mean “sensitive to heat-inactivation and LOW temperature”?

This is indeed what we meant to say and we have now clarified this.

Referee #3 (Remarks to the Author):

The authors have responded satisfactorily to my concerns.

The more thorough analysis of the DIESL knockout mice helps to better understand the physiological context in which this enzyme acts. The new data on liver lipidomics failed to detect ether PCs – possibly because ePCs are not very abundant in the liver. However, a reduction in liver TAGs was observed transiently at 3.5 weeks in male mice. This is reassuring. Yet, the statement of “a severe reduction (40%) in liver TAG”, which is essentially based on 2 animals, should be tuned down, or additional supporting data should be provided.

At the request of the reviewer, we have removed the word “severe” from the text.

Reviewer Reports on the Third Revision:

Referees' comments:

Referee #2 (Remarks to the Author):

Please change the text on line 28 (part of the Summary). Replace "lead to" with "are associated with".

The current version has appropriately addressed my concerns, especially with respect to metabolic characterization of the DIESL deficient mice.

Author Rebuttals to Third Revision:

Referees' comments:

Referee #2 (Remarks to the Author):

Please change the text on line 28 (part of the Summary). Replace "lead to" with "are associated with".

We have made the requested change.

The current version has appropriately addressed my concerns, especially with respect to metabolic characterization of the DIESL deficient mice.